# Don't Forget the Critic: Value-Based Data Rehearsal for Multi-Cyclic Continual Reinforcement Learning

## Abstract

Data rehearsal has emerged as a leading approach for mitigating catastrophic forgetting in Continual Reinforcement Learning (CRL). However, existing work remains confined to policy gradient frameworks, regularizing only actors due to the performance degradation incurred by critic regularization. This actor-centric approach overlooks the potential of data rehearsal for value function approximation. Moreover, existing evaluations in CRL rarely consider multi-cyclic environments where task sequences repeat, a critical real-world scenario that exacerbates forgetting and plasticity. We investigate data rehearsal for Deep Q-Networks using Q-value regularization in multi-cyclic settings and propose Qreg+NWLU which introduces two simple modifications: (1) continuous data rehearsal that dynamically collects and updates stored Q-values throughout training, and (2) "No-Wait" regularization that applies immediately rather than after the first task. Together, these modifications yield improvements in learning efficiency, forgetting mitigation, and knowledge transfer over Qreg and conventional CRL methods within value function approximation settings.

## 1 Introduction

Continual learning (CL) develops systems capable of lifelong learning by leveraging knowledge from prior experiences to address new problems while preserving performance on previously learned tasks. Like traditional CL, continual reinforcement learning (CRL) applies these principles to RL algorithms, enabling learning across multiple tasks. However, both face significant challenges, including catastrophic forgetting (Kirkpatrick et al., 2017) and plasticity loss (Dohare et al., 2024).

Recent data rehearsal CRL research has primarily adapted supervised CL methods to policy gradient frameworks, particularly actor-critic methods (Tomilin et al., 2023; Wołczyk et al., 2022; 2021; Rolnick et al., 2019; Ahn et al., 2025). These studies apply data rehearsal via behavioral cloning regularization exclusively to the actor, as extending regularization to the critic has been found to harm performance (Tomilin et al., 2023; Wołczyk et al., 2022; 2021). This has done little to encourage the exploration of value-based methods (Mnih et al., 2015; van Hasselt et al., 2016; Hessel et al., 2018; Wang et al., 2016) in CRL, leaving a significant gap in understanding how data rehearsal with regularization can enable effective value function learning in continual settings.

Moreover, existing catastrophic forgetting CRL research rarely evaluates methods in multi-cyclic settings, where the sequence of tasks repeats across multiple cycles. This setting is particularly important for real-world applications, where tasks and environments naturally reoccur over time. Multi-cyclic settings have already proven informative in plasticity loss research, where repeated task exposure has been shown to progressively degrade an agent's ability to adapt and learn (Abbas et al., 2023; Hemati et al., 2025; Dohare et al., 2024). Given that plasticity and forgetting are closely related phenomena, multi-cyclic evaluation is equally valuable for catastrophic forgetting analysis, as repeated task cycles can amplify forgetting effects or reveal resistance to forgetting that remains obscured in single-cycle settings (Houyon et al., 2023; Powers et al., 2022; Rolnick et al., 2019). Together, the limited study of value-based data rehearsal and the lack of multi-cyclic evaluation motivate our central question: how can data rehearsal with regularization be effectively adapted for Deep Q-Nework (DQN) (Mnih et al., 2015), a well-established and fundamental

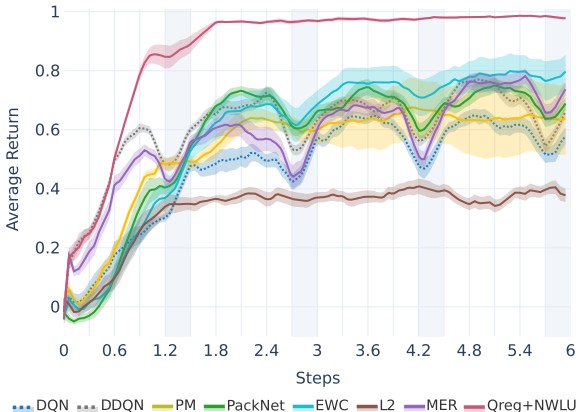

Figure 1: Conventional value-based CRL methods struggle to adapt or suffer catastrophic forgetting in multi-cyclic scenarios like the Minihack Room Ultimate task shown here. Our proposed Qreg+NWLU technique mitigates forgetting while enhancing knowledge transfer. The gray shaded regions indicate the training periods of the Ultimate task over four cycles.

value-based algorithm, to mitigate catastrophic forgetting in value-based algorithms, and how does such an approach compare to standard CRL methods under multi-cyclic settings?

We begin by evaluating data rehearsal with *Q-value regularization* (Qreg)[1], the value-based counterpart to behavioral cloning (Rolnick et al., 2019; Wołczyk et al., 2022; Tomilin et al., 2023). Our extensive experiments in multi-cyclic CRL settings reveal that Qreg exhibits instability and inconsistency, while standard CRL methods demonstrate subpar performance. To address these limitations in Qreg, we introduce *No-Wait Qreg with Live-Updates* (Qreg+NWLU), a modified approach that incorporates two key strategies: (1) *live-updates*, continuous data collection and updating throughout training, and (2) *No-Wait regularization*, applying Qreg immediately upon rehearsal sample availability rather than waiting until after the first task. Our results demonstrate that these strategies help increase training efficiency, decrease forgetting, and boost knowledge transfer across all evaluated environments (Figure 1). Code will be available at `https://anonymous.4open.science/r/Qreg-0F9A` upon publication. Our contributions are summarized as follows:

- We investigate value-based data rehearsal via regularization in multi-cyclic CRL setting.
- We identify limitations of Qreg and propose Qreg+NWLU with two simple modifications.
- We show Qreg+NWLU consistently outperforms standard CRL baselines.

## 2 Related Work

Similarly to the broader supervised CL literature, CRL works can be categorized into the following: weight regularization (Kirkpatrick et al., 2017), data rehearsal (Chaudhry et al., 2019), architectural regularization (e.g., parameter isolation) (Mallya & Lazebnik, 2018), and meta-learning (Riemer et al., 2019) methods (Tomilin et al., 2023; Wołczyk et al., 2022; Powers et al., 2022).

Weight regularization methods constrain weight updates based on prior task weights to prevent forgetting. Examples include L2 regularization (Kirkpatrick et al., 2017), which penalizes deviation from prior task weights, and Elastic Weight Consolidation (EWC) (Kirkpatrick et al., 2017), which uses the Fisher information matrix to weight constraints by each parameter's importance to prior tasks. Data rehearsal methods store samples from prior tasks for experience replay, regularizing the actor via behavioral cloning and the critic via L2 loss (Wołczyk et al., 2022; Rolnick et al., 2019). Architectural regularization preserves task-specific knowledge through network structure modifications, such as PackNet (Mallya & Lazebnik, 2018),

---

[1]General value function regularization has also been referred to as value-cloning and critic regularization in prior work.

which partitions network weights across tasks and updates only the relevant subset during training. Finally, meta-learning approaches adapt meta-learning frameworks to CL settings, such as Meta-Experience Replay (MER) (Riemer et al., 2019), which leverages optimization-based meta-learning to mitigate catastrophic forgetting.

Data rehearsal approaches in CRL have been primarily studied within the actor-critic framework (Tomilin et al., 2023; Wołczyk et al., 2022; Rolnick et al., 2019; Ahn et al., 2025). Traditionally, studies show that data rehearsal with behavioral cloning outperforms other CRL methods. Notably, these works apply CRL techniques predominantly to the actor network, as including the critic has been shown to harm performance (Tomilin et al., 2023; Wołczyk et al., 2022; 2021). Although some studies have explored value function approximation algorithms (e.g., DQN) in continual settings (Kirkpatrick et al., 2017; Riemer et al., 2019; Abbas et al., 2023), data rehearsal for value-based methods remains significantly underexplored compared to policy gradient approaches in CRL.

The plasticity literature has increasingly adopted multi-cyclic evaluation, where agents revisit the same task sequence multiple times, as a more realistic and informative benchmark. Tasks and environments naturally reoccur over time, and this setting reveals issues that single-cycle evaluations cannot surface, such as the progressive degradation of an agent's ability to adapt and learn (Abbas et al., 2023; Hemati et al., 2025; Dohare et al., 2024). For example, Abbas et al. (2023) demonstrates that plasticity deteriorates over successive cycles, even in state-of-the-art methods that perform well in single-cycle settings (Dohare et al., 2024). The same reasoning applies to catastrophic forgetting research, where multi-cyclic evaluation is both more realistic and informative, as repeated task cycles may amplify forgetting effects or reveal long-term robustness that shorter evaluations obscure (Houyon et al., 2023; Powers et al., 2022; Rolnick et al., 2019). Despite this, multi-cyclic evaluation remains underexplored in catastrophic forgetting research, with many studies either omitting it entirely (Riemer et al., 2019; Kirkpatrick et al., 2017; Ahn et al., 2025) or limiting evaluation to at most two cycles (Tomilin et al., 2023; Wołczyk et al., 2022)

## 3 Preliminaries

**Value Function Approximation** Value-function approximation uses parameterized functions to estimate the state-action value, which represents the expected long-term reward for taking a specific action in a given state. The action-value function $Q(s, a)$ is learned by satisfying the Bellman optimality equation:

$$Q(s, a) = E[r + \gamma \max_{a'} Q(s', a') | s, a] \tag{1}$$

where $r$ is the reward received after taking action $a$ in the state $s$, $\gamma \in [0, 1]$ is the discount factor determining the importance of future rewards, $s'$ is the resulting next state, and the expectation accounts for stochasticity in state transitions. The term $\max_{a'} Q(s', a')$ represents the maximum value achievable from the next state, ensuring the function captures optimal future behavior. The optimal policy $\pi$ is derived by selecting actions that maximize the Q-function: $\pi = \arg \max_a Q(s, a)$.

**Multi-cyclic Continual Learning** The goal of the CRL setting in this paper is to learn a policy $\pi_\theta$ by training the agent on a sequence of $N$ tasks $\mathcal{T} = \{\mathcal{T}_1, ..., \mathcal{T}_N\}$. Each task $\mathcal{T}_i$ is trained for a fixed number of iterations before the next task $\mathcal{T}_{i+1}$ is trained. Each sequence is cycled through $C$ times, meaning that once the end of a sequence has been reached, the cycle has concluded and a new sequence and cycle begin. To better reflect realistic continual learning conditions, it is assumed that there is no resetting of the weights, replay buffer, or optimizer variables between tasks or cycles, unless otherwise stated.

## 4 Data Rehearsal with Value Function Regularization

Data rehearsal with regularization has demonstrated strong performance in both supervised CL (Isele & Cosgun, 2018) and CRL (Rolnick et al., 2019; Wołczyk et al., 2022; Tomilin et al., 2023). However, as noted earlier, it has been predominantly applied to policy gradient methods while avoiding critic regularization

due to observed performance issues (Tomilin et al., 2023; Wołczyk et al., 2022; 2021). This leaves open the question of whether data rehearsal through value function regularization is effective when applied directly to value-based methods. In this section, we investigate Q-value regularization and explore additional strategies to enhance its effectiveness.

### 4.1 Q-Value Regularization

Data rehearsal in continual reinforcement learning (CRL) involves storing samples in a dedicated rehearsal replay buffer (RRB) which is separate from the standard replay buffer (Wołczyk et al., 2022; Rolnick et al., 2019). This RRB can be implemented with either fixed memory capacity (Rolnick et al., 2019) or unlimited memory (Wołczyk et al., 2022). However, as task sequences grow longer, fixed-capacity implementations become more practical.

In CRL, replay buffers are typically small relative to both the number of training steps per task and the total steps across all tasks and cycles (Riemer et al., 2019; Tomilin et al., 2023; Wołczyk et al., 2022; 2021). This size constraint limits their ability to retain diverse experiences. Consequently, the RRB and replay buffer serve complementary memory functions: the RRB acts as long-term, cross-task memory by storing samples from multiple tasks, while the replay buffer serves as short-term, task-specific memory by retaining only samples from the current task. This functional distinction is particularly pronounced when training steps per task substantially exceed the replay buffer size, or when the replay buffer is reset between tasks. Rehearsal samples are typically selected randomly from the replay buffer at the end of each task for storage in the RRB (Wołczyk et al., 2022).

For policy gradient regularization, the RRB is typically combined with behavioral cloning to regularize the actor policy (Wołczyk et al., 2022; Rolnick et al., 2019). When applying the RRB to value function approximation, Q-value regularization (Qreg) minimizes the squared L2 distance between current and stored Q-values:

$$\text{Qreg} = \frac{\lambda}{N_{\text{RBS}}} \sum_{i=1}^{N_{\text{RBS}}} (Q_\theta(s^{(i)}, a^{(i)}) - Q_{\text{RRB}}^{(i)})^2, \tag{2}$$

where $N_{\text{RBS}}$ is the regularization batch size, $\lambda$ is the regularization coefficient, $Q_\theta(s^{(i)}, a^{(i)})$ is the current Q-value computed from stored state-action pairs, and $Q_{\text{RRB}}^{(i)}$ is the corresponding stored Q-value. The RRB maintains tuples $(s, a, Q_{\text{RRB}})$ with samples drawn randomly for computing Qreg. Regularization is applied only after the first task completes and RRB samples are available (Wołczyk et al., 2022). Such sampling may introduce inconsistencies in Qreg's performance, as drawing from inadequately learned tasks risks storing suboptimal or inaccurate Q-values. This in turn can produce effects analogous to primacy bias (Nikishin et al., 2022), potentially hindering or outright preventing learning for the affected task and related tasks.

### 4.2 Live-Updates and No-Wait Regularization

To improve the consistency of the learning process for Qreg, we propose two simple yet crucial strategies for rehearsal sample management and regularization, collectively referred to as "No-Wait Qreg with Live-Updates" (Qreg+NWLU). These enhancements enable more timely and adaptive regularization, leading to substantial gains in consistency, stability, and knowledge retention across tasks in multi-cyclic CRL. Pseudocode for the Qreg+NWLU framework is provided in Appendix B.

First, we introduce "Live", a strategy in which rehearsal samples are incrementally added to the replay buffer at a fixed frequency throughout training, rather than sampling numerous transitions upon task completion as in prior work (Wołczyk et al., 2022). By distributing sample selection over time, Live reduces selection bias toward transitions occurring near the end of training and promotes greater sample diversity, particularly beneficial when the replay buffer cannot contain the entire task's training data. The severity of this bias depends on the disproportion between replay buffer size and training steps per task. In standard Qreg, a very small replay buffer means only samples near the end of training are selected for rehearsal, severely restricting diversity. This becomes critical when learning is incomplete or when performance degrades toward the end of the task, as rehearsal sample selection is confined to a narrow temporal window that may fail to capture the most informative experiences.

One limitation of both standard Qreg and Qreg with Live is that Q-values stored in the replay buffer are never updated. Inaccurate Q-values, whether due to insufficient training or obsolescence from knowledge gained in subsequent tasks, can persist throughout learning, potentially degrading performance by regularizing toward stale targets. To address this, we propose "Updates", where Q-values for rehearsal samples corresponding to the current task are periodically refreshed during training. For example, when training on Task 1, all Task 1 samples in the RRB have their Q-values updated at a preset frequency. This requires storing task identifiers alongside each sample. The combination of Live and Updates is particularly desirable, as samples are both frequently added and kept current. However, for standard Qreg with Updates (without Live), updates can only occur once samples for the current task have been added, meaning updating cannot begin until Cycle 2, potentially hindering early learning.

Second, we introduce "No-Wait" regularization, a mechanism enabled by the Live strategy. Rather than deferring regularization until task completion, No-Wait applies regularization immediately once rehearsal samples become available[2]. This in-time regularization aims to stabilize initial task learning, especially in settings where baseline DQN lacks a warm-up or exploration phase. By introducing regularization earlier in the training process, No-Wait helps mitigate instability that may arise at the beginning of learning, essential in continual learning where each task's knowledge must be efficiently leveraged for subsequent tasks.

## 5 Experiment Setup

### 5.1 Environments and Task Sequences

All tasks are chosen in accordance with prior works, with an emphasis on environment computational efficiency to enable thorough exploration into data rehearsal problems. All environment sequences contain "soft" transitions between tasks, where each new task in the sequence is defined by introducing new minor mechanics to the original environment or small perturbations to the environment dynamics. This contrasts with a "hard" transition, where a new task differs significantly or completely from previous tasks.

All experiments use the PyGame Learning Environment (PLE) (Tasfi, 2016) and Minihack (Samvelyan et al., 2021) frameworks. PLE offers Atari-like environments that are simpler than classical Atari but computationally efficient. Following Riemer et al. (2019), we use the PLE environments Flappy Bird (navigating a bird between pipes) and Catcher (catching falling fruit with a paddle). Minihack provides fast, scalable, procedurally generated tasks ideal for harder continual learning scenarios where tasks share goals but introduce new mechanics and observation restrictions. Following Powers et al. (2022), we adopt Minihack's Room environment, a modified Gridworld where the agent must reach the descending staircase while facing progressively more difficult modifiers.

We define two five-task sequences for Flappy Bird and Catcher similar to Riemer et al. (2019). In Flappy Bird, the pipe gap decreases from 100 by 5 per task, making task 5 the hardest. In Catcher, pellet velocity increases from 0.608 by 0.03 per task. Adapting Powers et al. (2022), the Minihack Room sequence features harder transitions than PLE due to the introduction of observation space changes. Starting with Room Random (basic navigation with random start and end positions), subsequent tasks introduce new challenges: Room Dark adds partial observability, Random Monsters adds lethal enemies, Random Trap adds teleportation traps, and Room Ultimate combines all modifiers. For each task sequence, tasks follow a fixed order that is repeated each cycle.

### 5.2 Metrics

We adapt the forgetting and forward transfer metrics from Powers et al. (2022) to create a general transfer metric that quantifies how much knowledge one task transfers to another. We do not quantitatively distinguish between forward and backward transfer, as these directional concepts become unclear with multiple training cycles. Let $R_{i,j}^t$ denote the total return for evaluation task $\mathcal{T}_i$ while training on task $\mathcal{T}_j$ at timestep $t$. We define *Final Transfer* as the change in return between the end of training on task $\mathcal{T}_{j-1}$ (i.e., just before

---

[2]This Qreg loss is computed even if the number of samples is less than the regularization batch size, in which case all available samples are used.

the current task) and the end of training on task $\mathcal{T}_j$, capturing the net effect of training on $\mathcal{T}_j$ with respect to performance on $\mathcal{T}_i$:

$$\mathcal{F}_{i,j} = \frac{R^{\text{end}}_{i,j} - R^{\text{end}}_{i,j-1}}{|\max_j R_{i,j}|}, \tag{3}$$

where $R^{\text{end}}_{i,j-1}$ is the return for evaluation task $\mathcal{T}_i$ immediately before training on task $\mathcal{T}_j$, and $\max_j R_{i,j}$ is the maximum return for evaluation task $\mathcal{T}_i$ across all training tasks. Values of $\mathcal{F}_{i,j} > 0$ indicate positive transfer, while $\mathcal{F}_{i,j} < 0$ indicates negative transfer. Forgetting is captured when $i < j$ and $\mathcal{F}_{i,j} < 0$, while forward transfer is captured when $i > j$ and $\mathcal{F}_{i,j} > 0$.

One downside to the Final Transfer metric is that it only compares performance based on the end of training, failing to capture negative transfer (i.e., forgetting) that occurs during training. If a task suffers significant degradation throughout training but recovers by the end, $\mathcal{F}$ will not detect this negative transfer. Thus, we define *Worst Transfer* as the difference between the return just before training on task $\mathcal{T}_{j-1}$ and the lowest return observed while training on task $\mathcal{T}_j$:

$$\mathcal{W}_{i,j} = \frac{R^{\text{min}}_{i,j} - R^{\text{end}}_{i,j-1}}{|\max_j R_{i,j}|}, \tag{4}$$

where $R^{\text{min}}_{i,j}$ is the minimum return achieved for task $\mathcal{T}_i$ while training on task $\mathcal{T}_j$.

For each evaluation period, we compute the performance using average return $\mathcal{G}$ and transfer metrics $\mathcal{F}/\mathcal{W}$ across all evaluation episodes and seeds. To further summarize these metrics we compute grand averages across all tasks and cycles. The grand average return is computed as

$$\overline{\mathcal{G}}_i = \frac{1}{CN} \sum_{c=1}^{C} \sum_{j=1}^{N} \mathcal{G}^c_{i,j}, \tag{5}$$

where $\overline{\mathcal{G}}_i$ summarizes the average return for the evaluation task $\mathcal{T}_i$ across all training tasks and cycles. The grand averages for transfer are computed for each training task $\mathcal{T}_j$ as

$$\overline{\mathcal{F}}_j = \frac{1}{CN} \sum_{c=1}^{C} \sum_{i=1}^{N} \mathcal{F}^c_{i,j}, \quad \overline{\mathcal{W}}_j = \frac{1}{CN} \sum_{c=1}^{C} \sum_{i=1}^{N} \mathcal{W}^c_{i,j}, \tag{6}$$

where each grand average summarizes the transfer from training on task $\mathcal{T}_j$ to all evaluation tasks by averaging transfer scores across evaluation tasks and cycles. Following Powers et al. (2022), all transfer metrics are scaled by 10 for readability.

### 5.3 Implementation Details

All algorithms employ DQN as the base framework to isolate the contribution of CRL-specific components. Strong baselines are established by selecting CRL algorithms according to the benchmark results of Tomilin et al. (2023). Specifically, PackNet, EWC, and L2 are included on the grounds that each outperforms competing methods within its respective category. Notably, the weight regularization approaches L2 and EWC match or exceed their counterparts (Nguyen et al., 2018; Aljundi et al., 2018). MER is additionally incorporated to ensure representation of the meta-learning family, which is otherwise absent from the benchmark. As conventional baselines, we employ vanilla DQN alongside a variant whose replay buffer capacity spans one full cycle of samples (1.5 million transitions), referred to hereafter as Perfect Memory (PM). We also include Double-DQN (DDQN) as a baseline variant which aims at address Q-value overestimation bias found in the original DQN (Van Hasselt et al., 2016).

All algorithms follow a consistent implementation protocol unless noted otherwise. Each algorithm is trained for four cycles or 6 million steps (300K per task) using 10 different random seeds. Evaluation occurs every 60K steps, with each task in the sequence evaluated over 10 episodes.[3] Consistent with prior work (Riemer

---

[3]For Catcher, only five evaluation episodes are used to reduce total runtime, with little impact on performance variance.

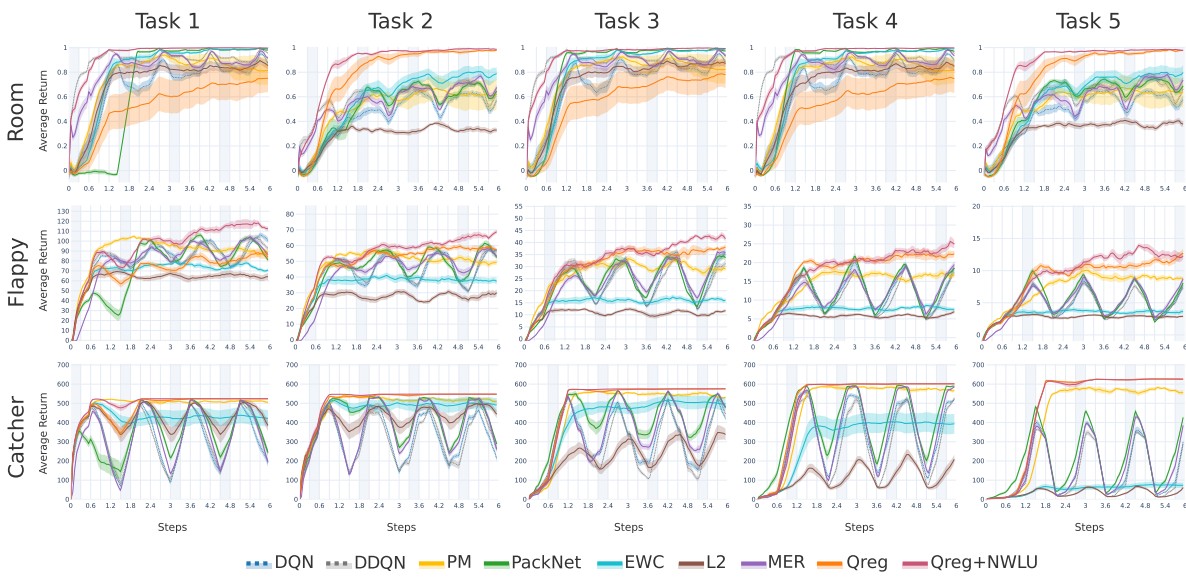

Figure 2: Learning curves for the total return $\mathcal{G}$ averaged over 10 seeds and smoothed with a moving average. The color shaded regions for each line report the standard error. The rows correspond to the task sequence, while the columns correspond to the individual tasks. The gray-shaded areas indicate the training periods for a particular task.

Table 1: Final transfer grand average $\bar{\mathcal{F}}$ with standard error for Room. Highlight indicates forgetting.

| Algorithm | Task 1 | Task 2 | Task 3 | Task 4 | Task 5 |
|---|---|---|---|---|---|
| DQN | -0.08 ± 0.13 | 0.45 ± 0.11 | 0.38 ± 0.18 | 0.78 ± 0.16 | 0.61 ± 0.09 |
| DDQN | 1.57 ± 0.06 | 0.29 ± 0.05 | 0.25 ± 0.04 | -0.06 ± 0.05 | 0.11 ± 0.04 |
| PM | 0.03 ± 0.18 | 0.56 ± 0.11 | 0.77 ± 0.07 | 0.48 ± 0.08 | 0.10 ± 0.09 |
| PackNet | 0.45 ± 0.04 | 0.81 ± 0.04 | 0.66 ± 0.06 | 0.30 ± 0.05 | 0.04 ± 0.04 |
| EWC | 0.30 ± 0.07 | 0.36 ± 0.09 | 0.68 ± 0.10 | 0.64 ± 0.10 | 0.37 ± 0.12 |
| L2 | 0.04 ± 0.07 | 0.48 ± 0.08 | 0.78 ± 0.08 | 0.65 ± 0.08 | 0.05 ± 0.09 |
| MER | 0.92 ± 0.14 | 0.45 ± 0.06 | 0.25 ± 0.12 | 0.19 ± 0.10 | 0.44 ± 0.11 |
| Qreg | 0.14 ± 0.05 | 0.40 ± 0.08 | 0.80 ± 0.08 | 0.63 ± 0.11 | 0.23 ± 0.04 |
| Qreg+NWLU | 1.21 ± 0.08 | 0.45 ± 0.04 | 0.63 ± 0.04 | 0.24 ± 0.05 | 0.04 ± 0.01 |

et al., 2019; Tomilin et al., 2023), all methods except for PM use a tiny replay buffer with 50K transition capacity, insufficient to store a full task. This constraint increases forgetting likelihood and emphasizes CRL algorithm effectiveness. Unlike traditional DQN, learning begins once the buffer reaches batch size, without warm-up or dedicated exploration. Following Riemer et al. (2019), we use a flat 5% epsilon-greedy rate for exploration throughout training. All algorithms use the standard DQN architecture (Mnih et al., 2015), except PackNet, which uses a Q-value head per Tomilin et al. (2023). The replay buffer, weights, and optimizer are never reset between tasks, and no task-specific exploration period is provided. PackNet is the exception and resets the buffer and optimizer between tasks to ensure task-specific training and avoid optimizer carry over degradation. Additional implementation and hyperparameter details can be found in Appendix D and Appendix F.

## 6 Results

Experiments are conducted on the Room, Flappy, and Catcher sequences using baselines including MER, L2, EWC, PackNet, PM, and DQN. Figure 2 depicts the learning curves for average return across all three tasks.

Table 2: Worst transfer grand average $\bar{\mathcal{W}}$ with standard error for Room. Highlight indicates notable forgetting ($< -0.1$).

| Algorithm | Task 1 | Task 2 | Task 3 | Task 4 | Task 5 |
|---|---|---|---|---|---|
| DQN | -0.50 ± 0.07 | -0.06 ± 0.07 | -0.22 ± 0.08 | -0.27 ± 0.08 | -0.24 ± 0.05 |
| DDQN | -0.01 ± 0.02 | -0.11 ± 0.03 | -0.04 ± 0.02 | -0.49 ± 0.03 | -0.37 ± 0.03 |
| PM | -0.39 ± 0.14 | -0.11 ± 0.09 | 0.03 ± 0.01 | 0.03 ± 0.03 | -0.24 ± 0.07 |
| PackNet | -0.07 ± 0.03 | 0.06 ± 0.02 | -0.05 ± 0.04 | -0.17 ± 0.03 | -0.31 ± 0.03 |
| EWC | -0.08 ± 0.03 | -0.01 ± 0.04 | 0.05 ± 0.03 | -0.03 ± 0.04 | -0.08 ± 0.06 |
| L2 | -0.42 ± 0.04 | -0.11 ± 0.04 | -0.11 ± 0.04 | -0.03 ± 0.04 | -0.23 ± 0.05 |
| MER | -0.12 ± 0.05 | -0.05 ± 0.03 | -0.20 ± 0.07 | -0.32 ± 0.05 | -0.29 ± 0.03 |
| Qreg | -0.13 ± 0.03 | 0.01 ± 0.03 | 0.07 ± 0.03 | 0.07 ± 0.04 | 0.01 ± 0.02 |
| Qreg+NWLU | -0.01 ± 0.01 | 0.08 ± 0.02 | 0.20 ± 0.01 | 0.05 ± 0.02 | -0.03 ± 0.01 |

Table 3: Return grand average $\bar{\mathcal{G}}$ with standard error for Room. Near-zero values are indicated by $^*$.

| Algorithm | Task 1 | Task 2 | Task 3 | Task 4 | Task 5 |
|---|---|---|---|---|---|
| DQN | 0.69 ± 0.02 | 0.45 ± 0.01 | 0.70 ± 0.02 | 0.69 ± 0.02 | 0.46 ± 0.01 |
| DDQN | 0.89 ± 0.01 | 0.58 ± 0.01 | 0.89 ± 0.01 | 0.89 ± 0.01 | 0.60 ± 0.01 |
| PM | 0.76 ± 0.03 | 0.52 ± 0.05 | 0.76 ± 0.03 | 0.76 ± 0.04 | 0.53 ± 0.05 |
| PackNet | 0.69 ± 0.00$^*$ | 0.51 ± 0.02 | 0.83 ± 0.00$^*$ | 0.83 ± 0.00$^*$ | 0.56 ± 0.01 |
| EWC | 0.81 ± 0.01 | 0.58 ± 0.04 | 0.81 ± 0.01 | 0.81 ± 0.01 | 0.59 ± 0.04 |
| L2 | 0.71 ± 0.04 | 0.29 ± 0.01 | 0.72 ± 0.04 | 0.70 ± 0.04 | 0.32 ± 0.01 |
| MER | 0.83 ± 0.01 | 0.56 ± 0.02 | 0.83 ± 0.01 | 0.83 ± 0.01 | 0.57 ± 0.02 |
| Qreg | 0.53 ± 0.11 | 0.78 ± 0.03 | 0.57 ± 0.11 | 0.54 ± 0.11 | 0.77 ± 0.03 |
| Qreg+NWLU | **0.94 ± 0.00$^*$** | **0.87 ± 0.00$^*$** | **0.93 ± 0.00$^*$** | **0.93 ± 0.00$^*$** | **0.87 ± 0.00$^*$** |

Grand average metrics for Flappy and Catcher are provided in Appendix G.1, while detailed per-task $\mathcal{F}$ and $\mathcal{W}$ metrics identifying which evaluation tasks are most affected are provided in Appendix G.3. Additionally, Appendix F.1 provides hyperparameter search results for Qreg+NWLU.

**Room** The first row in Figure 2 depicts the task learning curves for the Room task sequence. The grand averages for final transfer $\overline{\mathcal{F}}$, worst transfer $\overline{\mathcal{W}}$, and average return $\overline{\mathcal{G}}$ are reported in Tables 1, 2, and 3, respectively. Several baselines (DQN, DDQN, MER, and EWC) exhibit offset cyclic forgetting, alternating forgetting and recovery across cycles. This oscillation often occurs with a delay, as forgetting happens when training on the evaluation task itself. Take for instance the DQN, where Task 1 exhibits self-forgetting during training in Cycle 2 (1.5m steps). This likely occurs because the replay buffer is not reset, causing training to begin with Task 5 samples, and the significant disparity between Task 1 and 5 likely amplifies this forgetting. This phenomenon is reflected in Task 1's negative transfer metrics of $\overline{\mathcal{F}} = -0.08$ and $\overline{\mathcal{W}} = -0.50$, which are the worst among all algorithms.

For Task 1, 3, and 4, DDQN, PackNet, EWC, and MER achieve good performance by Cycle 2, with DDQN, Packnet and EWC approaching maximum performance by Cycle 3 (3m steps). DDQN performs exceptionally well, likely because its overestimation reduction prevents the collapse seen in other methods, consistent with primacy bias. Even so, it does have cyclic-forgetting throughout training. PackNet performs poorly on Task 1 during the first cycle because it evaluates using only the weights trained on Task 1, lacking accumulated knowledge from previous tasks until Cycle 2. PM performs well initially but starts to forget by Cycle 4 (4.5m steps), while L2 maintains mediocre performance with less pronounced cyclic forgetting than DQN. For Tasks 2 and 5, all baselines show forgetting and perform comparably to or only slightly better than DQN, with none reaching maximum performance. EWC performs best on these tasks, while L2 performs relatively poorly compared to other tasks.

Qreg exhibits high standard error on Tasks 1, 3, and 4, driven by a stark dichotomy between runs that achieve strong performance and those that fail entirely. This reflects an effect akin to primacy bias, where failure to initially learn Task 1 inhibits learning in subsequent cycles and on structurally similar tasks (e.g., Task 3 and 4). However, Qreg is still able to learn Tasks 2 and 5, likely due to differences in their observation spaces. Notably, Qreg exhibits minimal worst-case negative transfer across all tasks except Task 1, avoiding the performance degradation seen in other methods. Qreg+NWLU outperforms all algorithms, achieving near-optimal performance by Cycle 2 with notably low standard error. By mitigating Qreg's volatility, these strategies simultaneously achieve positive $\overline{\mathcal{F}}$ scores, near-zero $\overline{\mathcal{W}}$ scores, and the highest $\overline{\mathcal{G}}$ across all tasks, demonstrating strong robustness against cyclic forgetting while preserving overall performance.

**Flappy** The Flappy results are highly variable due to random pipe gap placement. As task difficulty increases, some gaps become nearly impossible to traverse due to gap size and placement, causing significant return decreases from Task 1 to Task 5. Consequently, small improvements in later tasks are as meaningful as large improvements in earlier tasks. The second row in Figure 2 depicts the learning curves for the Flappy sequence. DQN, DDQN, MER, and PackNet exhibit increasing cyclic forgetting as difficulty rises, with large negative $\overline{\mathcal{F}}$ and $\overline{\mathcal{W}}$ scores for Tasks 1 and 2. L2 and EWC show fewer signs of cyclic forgetting but achieve worse overall performance. PM starts strong but gradually begins forgetting by Cycles 2 or 3, leading to decreased performance for most tasks.

Qreg shows reduced cyclic forgetting as task difficulty increases. While Qreg performs worse than DQN on Task 1 throughout all cycles, it shows improved performance on other tasks, reflected in positive $\overline{\mathcal{F}}$ and small negative $\overline{\mathcal{W}}$ scores. Qreg+NWLU follows a similar trend to Qreg but outperforms DQN on Task 1, with consistent improvement over time across all tasks. While its $\overline{\mathcal{F}}$ and $\overline{\mathcal{W}}$ scores remain comparable to Qreg, it achieves the highest $\overline{\mathcal{G}}$ across all tasks.

**Catcher** The Catcher results are more straightforward than their Flappy counterparts. The third row in Figure 2 shows the learning curves for the Catcher sequence. DQN, DDQN, MER, PackNet, and L2 all exhibit cyclic forgetting with a noticeable drop in peak performance by Task 5. While these methods achieve positive $\overline{\mathcal{F}}$ scores, they have significantly large negative $\overline{\mathcal{W}}$ scores. L2 performs reasonably well and forgets less than the other aforementioned algorithms on early tasks but struggles significantly on Tasks 3, 4, and 5. MER and PackNet perform on par with or slightly better than DQN, depending on the task and cycle. EWC shows minimal forgetting at the cost of lower peak performance and higher variance. Among the baselines, PM is the only method that maintains both low forgetting and good performance.

Qreg performs well and reaches near-maximum performance by Cycle 2 for all tasks. However, Qreg exhibits some forgetting except during the first cycle for Task 1. Similarly, Qreg+NWLU reaches near-optimal performance by Cycle 2 and greatly mitigates the forgetting observed in Qreg. Although Qreg+NWLU shows a small amount of forgetting for Task 5 at the end of Cycle 2, it quickly recovers to achieve maximum performance. Overall, Qreg and Qreg+NWLU perform similarly across all metrics, with Qreg+NWLU achieving marginally higher scores across most tasks.

Table 4: Ablation final transfer grand average $\overline{\mathcal{F}}$ with standard error for Room. Highlight indicates forgetting.

| Algorithm | Task 1 | Task 2 | Task 3 | Task 4 | Task 5 |
|---|---|---|---|---|---|
| DQN | -0.08 ± 0.13 | 0.45 ± 0.11 | 0.38 ± 0.18 | 0.78 ± 0.16 | 0.61 ± 0.09 |
| DDQN | 1.57 ± 0.06 | 0.29 ± 0.05 | 0.25 ± 0.04 | -0.06 ± 0.05 | 0.11 ± 0.04 |
| Qreg | 0.14 ± 0.05 | 0.40 ± 0.08 | 0.80 ± 0.08 | 0.63 ± 0.11 | 0.23 ± 0.04 |
| U | 0.26 ± 0.15 | 0.48 ± 0.07 | 0.86 ± 0.05 | 0.67 ± 0.11 | 0.20 ± 0.04 |
| L | 0.32 ± 0.09 | 0.35 ± 0.08 | 0.41 ± 0.07 | 0.09 ± 0.06 | 0.13 ± 0.04 |
| LU | 0.19 ± 0.07 | 0.58 ± 0.06 | 0.93 ± 0.09 | 0.61 ± 0.10 | 0.16 ± 0.04 |
| NWL | 1.50 ± 0.05 | 0.39 ± 0.01 | 0.44 ± 0.03 | 0.08 ± 0.02 | 0.06 ± 0.01 |
| NWLU | 1.21 ± 0.08 | 0.45 ± 0.04 | 0.63 ± 0.04 | 0.24 ± 0.05 | 0.04 ± 0.01 |

Table 5: Ablation worst transfer grand average $\bar{\mathcal{W}}$ with standard error for Room. Highlight indicates notable forgetting $(< -0.1)$.

| Algorithm | Task 1 | Task 2 | Task 3 | Task 4 | Task 5 |
|-----------|--------|--------|--------|--------|--------|
| DQN | -0.50 ± 0.07 | -0.06 ± 0.07 | -0.22 ± 0.08 | -0.27 ± 0.08 | -0.24 ± 0.05 |
| DDQN | -0.01 ± 0.02 | -0.11 ± 0.03 | -0.04 ± 0.02 | -0.49 ± 0.03 | -0.37 ± 0.03 |
| Qreg | -0.13 ± 0.03 | 0.01 ± 0.03 | 0.07 ± 0.03 | 0.07 ± 0.04 | 0.01 ± 0.02 |
| U | -0.13 ± 0.03 | 0.04 ± 0.03 | 0.12 ± 0.03 | 0.14 ± 0.03 | 0.03 ± 0.02 |
| L | -0.12 ± 0.02 | -0.06 ± 0.04 | -0.11 ± 0.02 | -0.26 ± 0.02 | -0.28 ± 0.02 |
| LU | -0.07 ± 0.02 | 0.10 ± 0.03 | 0.12 ± 0.03 | 0.13 ± 0.03 | 0.06 ± 0.01 |
| NWL | -0.04 ± 0.02 | 0.01 ± 0.01 | 0.15 ± 0.01 | -0.08 ± 0.02 | -0.11 ± 0.02 |
| NWLU | -0.01 ± 0.01 | 0.08 ± 0.02 | 0.20 ± 0.01 | 0.05 ± 0.02 | -0.03 ± 0.01 |

Table 6: Ablation return grand average $\bar{\mathcal{G}}$ with standard error for Room. $^*$ indicates near-zero values.

| Algorithm | Task 1 | Task 2 | Task 3 | Task 4 | Task 5 |
|-----------|--------|--------|--------|--------|--------|
| DQN | 0.69 ± 0.02 | 0.45 ± 0.01 | 0.70 ± 0.02 | 0.69 ± 0.02 | 0.46 ± 0.01 |
| DDQN | 0.89 ± 0.01 | 0.58 ± 0.01 | 0.89 ± 0.01 | 0.89 ± 0.01 | 0.60 ± 0.01 |
| Qreg | 0.53 ± 0.11 | 0.78 ± 0.03 | 0.57 ± 0.11 | 0.54 ± 0.11 | 0.77 ± 0.03 |
| U | 0.73 ± 0.08 | 0.82 ± 0.02 | 0.75 ± 0.07 | 0.72 ± 0.08 | 0.82 ± 0.02 |
| L | 0.17 ± 0.05 | 0.30 ± 0.07 | 0.23 ± 0.06 | 0.18 ± 0.05 | 0.31 ± 0.07 |
| LU | 0.75 ± 0.09 | 0.84 ± 0.02 | 0.75 ± 0.09 | 0.75 ± 0.09 | 0.84 ± 0.02 |
| NWL | 0.92 ± 0.01 | 0.79 ± 0.02 | **0.93 ± 0.00**$^*$ | 0.91 ± 0.01 | 0.77 ± 0.01 |
| NWLU | **0.94 ± 0.00**$^*$ | **0.87 ± 0.00**$^*$ | **0.93 ± 0.00**$^*$ | **0.93 ± 0.00**$^*$ | **0.87 ± 0.00**$^*$ |

### 6.1 Ablations

We conduct an ablation study to investigate the importance of each strategy for Qreg+NWLU. We evaluate Qreg+U (Updates only), Qreg+L (Live only), Qreg+LU (Live-Updates only), and Qreg+NWL (No-Wait and Live only), comparing them against DQN, Qreg, and Qreg+NWLU baselines. Figure 3 depicts the task learning curves for all task sequences. $\overline{\mathcal{F}}$, $\overline{\mathcal{W}}$, and $\overline{\mathcal{G}}$ for Room are reported in Tables 4, 5, and 6.

**Room** The Room ablation results show that removing No-Wait regularization leads to extremely high standard error and variance, as previously observed with Qreg. While Qreg+U and Qreg+L improve over Qreg across all tasks, they underperform DQN on Tasks 1, 3, and 4. In contrast, Qreg+NWL and Qreg+NWLU perform best in all tasks, demonstrating significantly improved first-task learning with increased peak returns, accelerated learning speed, and large positive transfer with high $\overline{\mathcal{F}}$ scores. Notably, No-Wait seems to alleviate the primacy bias-like affect observed in Qreg.

**Flappy** For the Flappy ablation, all variations perform similarly across Tasks 2-5, except Qreg+L and Qreg+NWL, which underperform both Qreg and DQN, demonstrating that Updates are critical for initial task learning and positive transfer. For Task 1, Qreg+NWLU and Qreg+LU perform best with positive $\overline{\mathcal{F}}$ scores and small negative $\overline{\mathcal{W}}$ scores, indicating temporary forgetting that recovers by the end. Qreg+NWLU performs slightly better, experiencing a smaller performance drop at the beginning of Cycle 2 than Qreg+LU, demonstrating that NWLU improves initial task performance while also reducing forgetting.

**Catcher** The Catcher ablation results shows similar results to Flappy, as both involve softer task transitions. All variations perform comparably across Tasks 2-5 except Qreg+L and Qreg+NWL. For Task 1, Qreg+NWLU and Qreg+LU are the top performers, though Qreg alone performs well, with all three achieving positive $\overline{\mathcal{F}}$ and $\overline{\mathcal{W}}$ scores. Qreg+NWLU again outperforms Qreg+LU due to a smaller performance drop at the beginning of Cycle 2. However, Qreg+NWLU experiences slight forgetting around Task 3 in Cycle 2, whereas Qreg+LU does not.

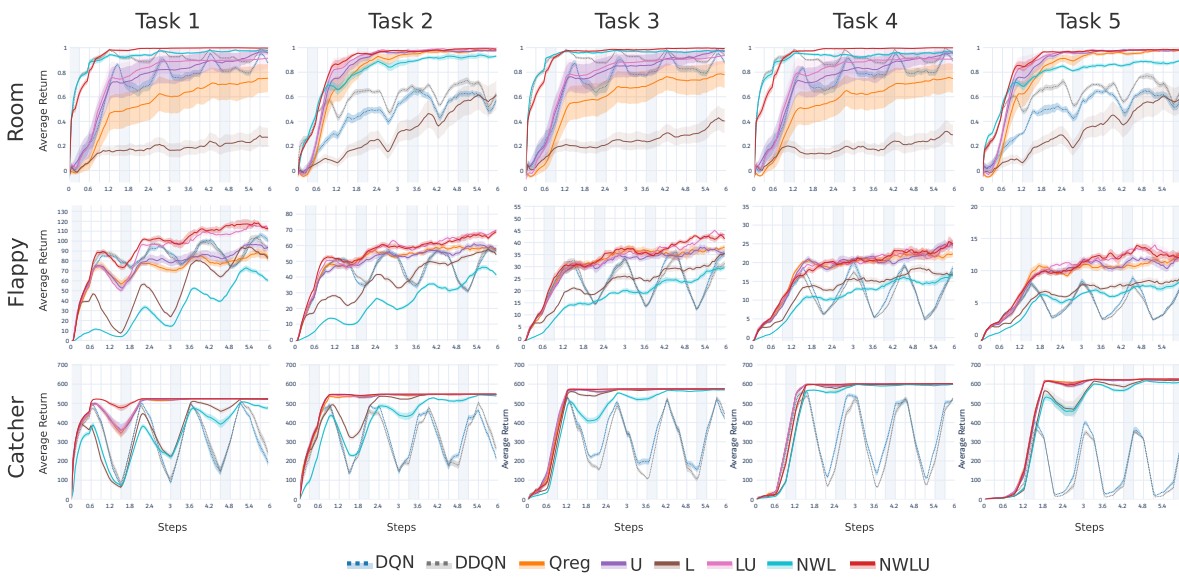

Figure 3: Ablation learning curves for the total return $\mathcal{G}$ averaged over 10 seeds smoothed with a moving average.

## 7  Discussion

These results demonstrate that data rehearsal with Qreg is promising for preventing forgetting and encouraging positive transfer. Qreg's major drawback is its inconsistency across environments, as evidenced by the Room sequence results which seem to suffer from a primacy bias-like affect. When further analyzing Qreg's high variability in the Room sequence, we found that the poor runs are characterized by unstable Q-value norms $\|Q\|_2$ that fluctuate greatly upon task transitions. As shown in the Room row of Figure 4, Qreg exhibits large jumps in Q-norms between tasks, while Qreg+NWLU produces smoother, more stable transitions. Yet, Qreg shows good baseline performance for the Flappy and Catcher task sequences, despite exhibiting less smooth Q-norms (Figure 4, rows 2–3). This could be due to these tasks having softer transitions that naturally progress in difficulty, whereas Room has harder transitions with similar tasks spread out rather than following one after another. Nevertheless, for these tasks, Qreg+NWLU still facilitates smoother transitions in Q-norms, while also reducing the range of Q-norm values over each cycle.

When examining the individual strategies of Qreg+NWLU, a discrepancy arises for Qreg+NWL and Qreg+LU between Room and Catcher/Flappy. In Room, Qreg+NWL outperforms Qreg+LU, performing on par with Qreg+NWLU, even though it produces higher variance in the Q-norms but with smooth value transitions. Additionally it seems that Qreg+NWL, particularly No-Wait, accounts for the stable and fast learning for Task 1 in Room, replicating results simar to that of the DDQN. This could suggest benefits to regularizing Q-values immediately rather than waiting for the 2nd task, achieving similar overestimation and primacy bias prevention. Comparatively, in Flappy and Catcher, Qreg+LU outperforms Qreg+NWL. While both display smoothing of Q-norms, Qreg+LU additionally clamps the Q-norm value range over each cycle. The addition of No-Wait (i.e., Qreg+NWLU) mainly appears to help mitigate forgetting of the first task. These findings suggest that all strategies help promote smoother Q-norms between task transitions, but Live and Updates help to also clamp the range of Q-norm values over time. Collectively, these strategies enhance the robustness of Qreg even if individual strategies are not equally effective in all environments.

## 8  Limitations

This work demonstrates that Qreg is a robust and effective approach for CRL when using DQN, a purely value-based algorithm. At the same time, several limitations point to valuable directions for future research. First, the evaluation excludes randomized task sequences and hard transitions involving sequences of entirely

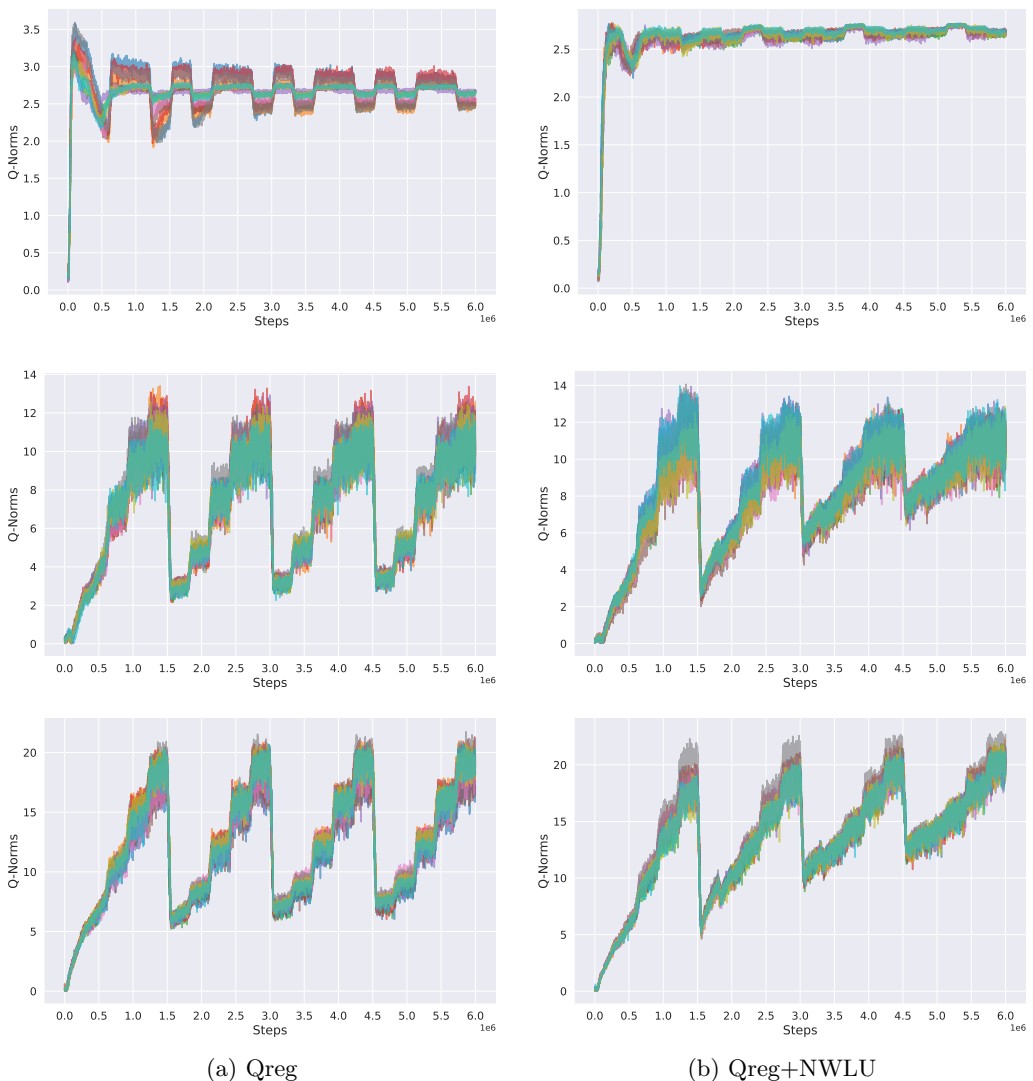

(a) Qreg                    (b) Qreg+NWLU

Figure 4: Q-value norm comparisons between Qreg (left) and Qreg+NWLU (right) across all three tasks (Room, Flappy, Catcher, top to bottom). For Qreg+NWLU, the Room task shows consistently smooth transitions, while the Flappy and Catcher tasks show a progressively shrinking Q-norm range over time. Color indicates each unique run.

unique tasks. Second, the Updates strategy relies on task labels and typically requires frequent updates (task dependent, see Appendix F.1). Third, this value-focused approach does not account for value networks in actor-critic frameworks, where more intricate value-policy interactions could present additional challenges. Finally, our analysis focus exclusively on Qreg and does not explore alternative methods for utilizing RRB, such as knowledge distillation methods (Ahn et al., 2025).

# 9 Conclusion

In conclusion, incorporating data rehearsal into value function approximation via Qreg significantly enhances learning efficiency, transfer capabilities, and resilience to multi-cyclical forgetting. However, achieving this requires the proper framework. Our results show that Qreg+NWLU, combining No-Wait regularization, continual sampling of rehearsal data from the replay buffer, and periodic updating of the rehearsal samples, provides a robust framework for achieving them. We believe this work is a first step towards broader application of value-based data rehearsal regularization in CRL.

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

## A Notation

Table 7 displays the notation used throughout the paper and appendix.

Table 7: Notation

| Notation | Definition |
|---|---|
| $T_{\text{steps}}$ | Number of timesteps when training a task |
| $t$ | The current timestep |
| $N$ | Number of tasks |
| $\mathcal{T}$ | Task sequence |
| $\mathcal{T}_i$ | The $i$th task in $\mathcal{T}$ |
| $C$ | Number of cycles |
| $\theta$ | Network parameters |
| $\hat{\theta}$ | Target network parameters |
| $Q_\theta$ | Q-values computed using the network parameters $\theta$ |
| $D_{\text{RB}}$ | The replay buffer |
| $N_{\text{RB}}$ | Size of the replay buffer memory |
| $N_{\text{BS}}$ | Batch size or the batch size for the replay buffer |
| $F_{\text{Train}}$ | Training frequency or the rate (in steps) between training intervals |
| $F_{\text{TNU}}$ | Target network update frequency or rate (in steps) at which the target network is updated |
| $D_{\text{RRB}}$ | The Rehearsal Replay Buffer (RRB) or the buffer that contains the data rehearsal samples |
| $N_{\text{RRB}}$ | Size of the RRB memory |
| $F_{\text{RAF}}$ | Regularization add frequency or the rate (in steps) at which samples are added to the RRB |
| $F_{\text{RUF}}$ | Regularization update frequency or the rate (in steps) at which RRB samples are updated |
| $N_{\text{RBS}}$ | Regularized batch size or batch size for the RRB |
| $N_{\text{RASS}}$ | Regularization add sample size or the number of rehearsal samples selected from the replay buffer to appended to the RRB |
| $N_{\text{RAH}}$ | Regularization add history or the number of latest samples from the replay buffer from which rehearsal samples equal to RASS are drawn from |
| $Q_{\text{RRB}}$ | Q-values stored in the RRB |
| $\lambda$ | Scaling factor for Q-value regularization loss |

## B  Pseudocode

Algorithm 1 depicts the psuedo-code for the Qreg+NWLU framework using notation covered in Table 7.

---

**Algorithm 1** Qreg+NWLU Framework

---

1: Initialize value function $Q$ with random weights $\theta$
2: Initialize target value function $\hat{Q}$ with weights $\hat{\theta}$
3: Initialize replay buffer $D_{\mathrm{RB}}$ to size $N_{RB}$
4: Initialize regularization replay buffer $D_{\mathrm{RRB}}$ to size $N_{RRB}$
5: global_step $\leftarrow 0$
6: **for** $c = 1$ **to** $C$ **do**
7:     **for** $i = 1$ **to** $\mathcal{T}$ **do**
8:         **for** $t = 1$ **to** $T_{\mathrm{steps}}$ **do**
9:             $s \leftarrow$ get current state from $\mathcal{T}_i$
10:            $a \leftarrow$ select action
11:            $s', r \leftarrow$ step $\mathcal{T}_i$ using $a$
12:            Store $(s, a, r, s', i)$ in $D_{\mathrm{RB}}$                                  ▷ Store transition and task ID $i$

13:            Every $F_{\mathrm{TNU}}$ global_step:
14:                Update $\hat{\theta}$ with $\theta$

15:            Every $F_{\mathrm{RUF}}$ global_step:                                  ▷ Update RRB Samples
16:                Update rehearsal samples in RRB with task ID $i$

17:            Every $F_{\mathrm{RAF}}$ global_step:                                  ▷ Live sample rehearsal samples
18:                $D_{\mathrm{RB}}^{\mathrm{latest}} \leftarrow$ Sample most recent $N_{\mathrm{RAH}}$ samples from $D_{\mathrm{RB}}$
19:                $s_b \leftarrow$ Randomly select $N_{\mathrm{RASS}}$ states from $D_{\mathrm{RB}}^{\mathrm{latest}}$
20:                $Q(s_b, \cdot) \leftarrow$ Compute all Q-values for rehearsal samples
21:                Store $(s_b, Q(s_b, \cdot))$ in $D_{\mathrm{RRB}}$

22:            Every $F_{\mathrm{Train}}$ global_step:
23:                Sample batch of size $N_{\mathrm{BS}}$ from $D_{\mathrm{RB}}$                ▷ Value function update
24:                Compute $\nabla_Q$ using value-function loss
25:                Sample batch of size $N_{\mathrm{RBS}}$ from $D_{\mathrm{RRB}}$            ▷ No-Wait regularization update
26:                Compute $\nabla_{\mathrm{Qreg}}$ using Equation 2
27:                $\theta \leftarrow \nabla_Q + \nabla_{\mathrm{Qreg}}$

28:                global_step $\leftarrow$ global_step $+ 1$
29:         **end for**
30:     **end for**
31: **end for**

---

## C  Environment Details

Figure 5 depicts the Minihack Room tasks. Room actions consist of the 8 cardinal directions for movement (N, NE, E, SE, S, SW, W, NW). A reward of $-1e^{-3}$ is given per each step to encourage finding the shortest path. Once the goal is reached, the game ends and a sparse reward of +1 is provided. Different modifiers such as enemies can result in the game ending without any reward given.

Figure 6 depicts the PLE tasks. Flappy actions consist of either flying up or no action where gravity pulls the bird down. Colliding with the ceiling, floor, or the pipes terminates the game eliciting a reward of -1 while a reward of +1 is given for successfully flying between the two pipes. Catcher actions consist of moving either left or right along the x-axis. Each time a fruit is missed, a life is lost and a reward of -1 is given while each time a fruit is caught, a reward of +1 is given. There are a total of 3 lives and the game ends when there are no more lives remaining.

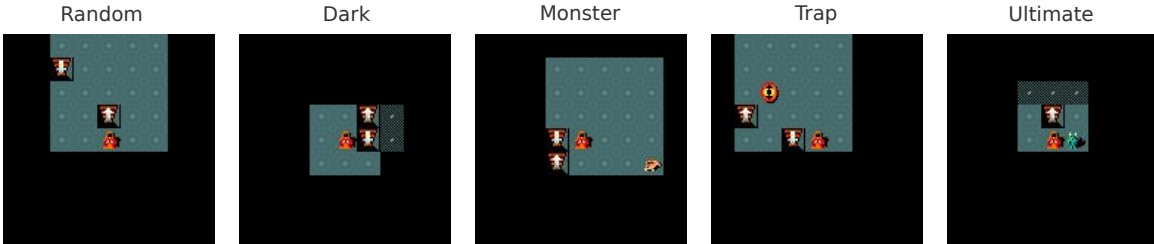

Figure 5: Example of environments from Minihack Room task sequence.

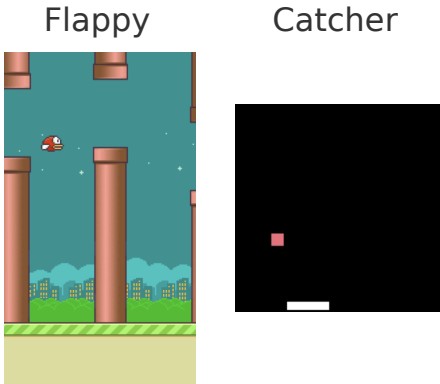

Figure 6: Example of PLE environments, Flappy and Catcher.

## D  Implementation Details

All algorithms use the standard DQN architecture as defined according to (Mnih et al., 2015). The encoder, responsible for outputting the latent embeddings, consists of a three-layer convolutional neural network followed by a fully connected layer. The first layer has 32 filters and an 8×8 kernel. The second layer has 64 filters and a 4×4 kernel. The third layer has 64 filters and a $3 \times 3$ kernel. The final fully connected layer consists of 256 hidden units. Each layer is preceded by a ReLU activation. The Q-value head takes the encoder's output and output the Q-values corresponding to the environment's action space.

The following baselines have had minor changes made to their algorithm, architecture, or CRL setup, typically done to improve performance. For example, L2 regularization is applied only for the encoder, as regularizing the Q-value head resulted in extremely degraded performance. MER uses batch learning for computing the DQN loss instead of online learning, which otherwise led to extremely poor results. PackNet resets the buffer to ensure the selected weights for a given task are trained exclusively on that task. Additionally, PackNet

resets the optimizer between tasks, as retraining it cause performance degradation. Finally, PackNet's Q-value head includes layer normalization to match the architecture used in (Tomilin et al., 2023).

Data is preprocessing for images is given as follows. For all environments (PLE and Minihack) images resized to (84,84), and image normalization is performed, before being passed to the DQN. For PLE tasks, images are stacked so that the latest four images stacked together and are converted to gray-scale. For Minihack, there is no frame stacking and colored images are used.

## E    Hardware

Experiments were conducted in a cluster environment across four different types of GPU nodes: NVIDIA A40 with an Intel Xeon Gold 6326 CPU, NVIDIA A100 with an Intel Xeon Gold 6326 CPU, NVIDIA A100 with an AMD EPYC 7502 CPU, NVIDIA L40s with an Intel Xeon Gold 6338 CPU or Intel Xeon Platinum 8362 CPU, and NVIDIA H200 with an AMD EPYC 9555 CPU. All nodes used 64 GB of RAM.

# F   Hyperparameters

This section presents the hyperparameters used for all algorithms, along with learning curve plots for various Qreg+NWLU hyperparameter values. Table 8 lists the hyperparameters for each algorithm.

Baseline hyperparameters were set based on prior work. Extensive hyperparameter search for baselines was only conducted when performance showed little to no learning. All hyperparameter searches used the Minihack Room as the testing environment. For EWC, we searched over regularization coefficient values of $[25, 100, 10^3, 10^4, 10^5]$. For L2, we searched over regularization coefficient values of $[0.01, 1, 10, 100, 10^3]$. For MER, we searched over batch sizes of $[16, 32]$ and online/batch sampling strategies.

For Qreg and its variations, with $N_{\mathrm{RRN}} = 100k$ and $N_{\mathrm{RASS}} = 10k$, the RRB holds 20k samples per task (two cycles of samples before overwriting). A large regularization batch size such as $N_{\mathrm{RBS}} = 256$ helps ensure (though does not guarantee) that rehearsal samples from all seen tasks are included in any given Qreg update. For Qreg+NWLU, with $F_{\mathrm{RAF}} = 2k$, $N_{\mathrm{RAH}} = 2k$, and $N_{\mathrm{RASS}} = 64$, approximately 9.6k rehearsal samples are added to the RRB by the end of training on each task (compared to 10k samples with standard Qreg). Typically, $F_{\mathrm{RAF}}$ and $N_{\mathrm{RAH}}$ should be equal to avoid skipping transition samples. If $N_{\mathrm{RAH}} < F_{\mathrm{RAF}}$, some samples will not be considered for selection as rehearsal samples. Finally, with $F_{\mathrm{RUF}} = F_{\mathrm{RAF}}$, samples are updated before adding new samples.

| Hyperparameter | Value |
|---|---|
| *Shared* | |
| optimizer | Adam |
| learning rate | $10^{-4}$ |
| exploration rate | 0.05 |
| discount factor | 0.99 |
| target update frequency | 10k |
| buffer size | 50k |
| frame stack | 4 |
| frame skip | 4 |
| timestep per collection (train frequency) | 4 |
| reward clipping | (-1, 1) |
| *Qreg* | |
| $\lambda$ | 1 |
| $F_{\mathrm{RAF}}$ | 300k |
| $N_{\mathrm{RASS}}$ | 10k |
| $N_{\mathrm{RBS}}$ | 256 |
| *Qreg+NWLU* | |
| $\lambda$ | 1 |
| $F_{\mathrm{RAF}}$ | 2k |
| $N_{\mathrm{RASS}}$ | 64 |
| $F_{\mathrm{RUF}}$ | 2k |
| $N_{\mathrm{RBS}}$ | 256 |
| *Qreg+NWL* | |
| $\lambda$ | 1 |
| $F_{\mathrm{RAF}}$ | 2k |
| $N_{\mathrm{RASS}}$ | 64 |
| $N_{\mathrm{RBS}}$ | 256 |
| *Qreg+U* | |
| $\lambda$ | 1 |
| $F_{\mathrm{RAF}}$ | 300k |
| $N_{\mathrm{RASS}}$ | 10k |
| $F_{\mathrm{RUF}}$ | 300k |
| $N_{\mathrm{RBS}}$ | 256 |
| *Qreg+L* | |
| $\lambda$ | 1 |
| $F_{\mathrm{RAF}}$ | 2k |
| $N_{\mathrm{RASS}}$ | 64 |
| $N_{\mathrm{RBS}}$ | 256 |
| *EWC* | |
| regularization coefficient | 100k |
| *L2* | |
| regularization coefficient | 100 |
| *PackNet* | |
| retrain steps | $10^4$ |
| *MER* | |
| Within $\beta$ | 1 |
| Across $\lambda$ | 0.3 |

Table 8: Hyperparameters used for all algorithms.

### F.1 Qreg+NWLU Hyperparameter Search Results

Below are various learning curve plots for Qreg+NWLU hyperparameter investigations. Figure 7 depicts various $\lambda$ values for scaling, where values around $\lambda = 1$ tend to perform best across all tasks. Figure 8 depicts various $N_{\mathrm{RRB}}$ values (size of RRB memory), where performance can be maintained as $N_{\mathrm{RRB}}$ shrinks. Here, 100k allows for two cycles worth of samples to be stored, with each task having roughly the same number of samples. Likewise, 50k allows for one cycle worth of samples to be stored, and 25k allows for half a cycle of samples to be stored. Figure 9 depicts various $F_{\mathrm{RUF}}$ values (rate at which samples in RRB are updated), where performance tends to drop as $F_{\mathrm{RUF}}$ increases. It should be noted that the frequency at which samples are added to the RRB remains constant at $F_{\mathrm{RAF}} = 2\mathrm{k}$. Increasing $F_{\mathrm{RUF}}$ alongside $F_{\mathrm{RAF}}$ could improve performance, as this would reduce the number of samples with potentially outdated Q-values added early in training.

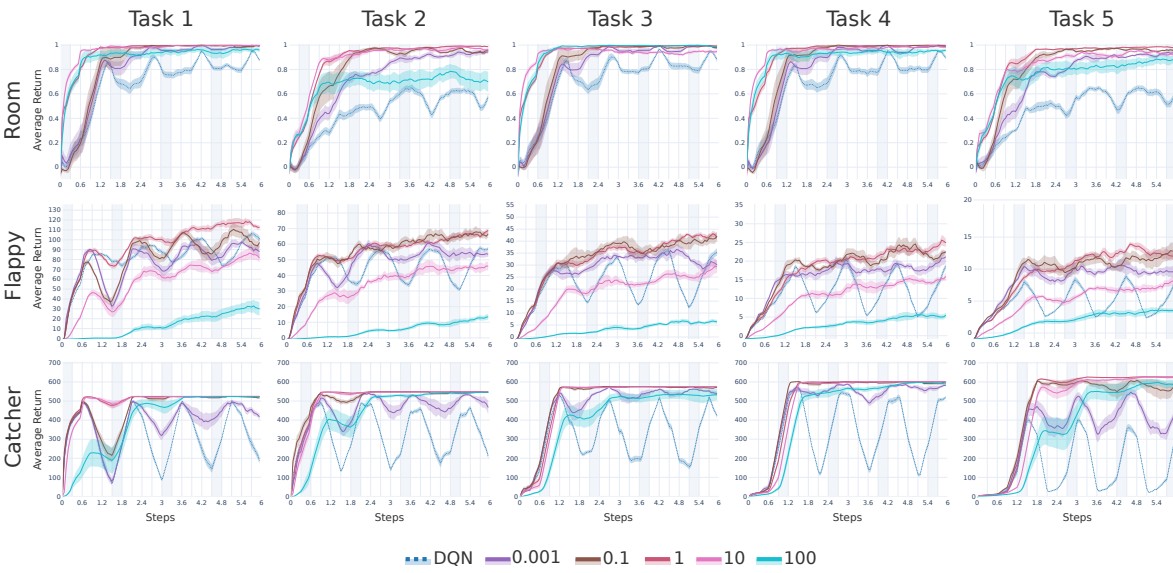

Figure 7: Learning curves of Qreg+NWLU total return $\mathcal{G}$ using various $\lambda$ values (scaling factor for Qreg loss), averaged over 5 seeds.

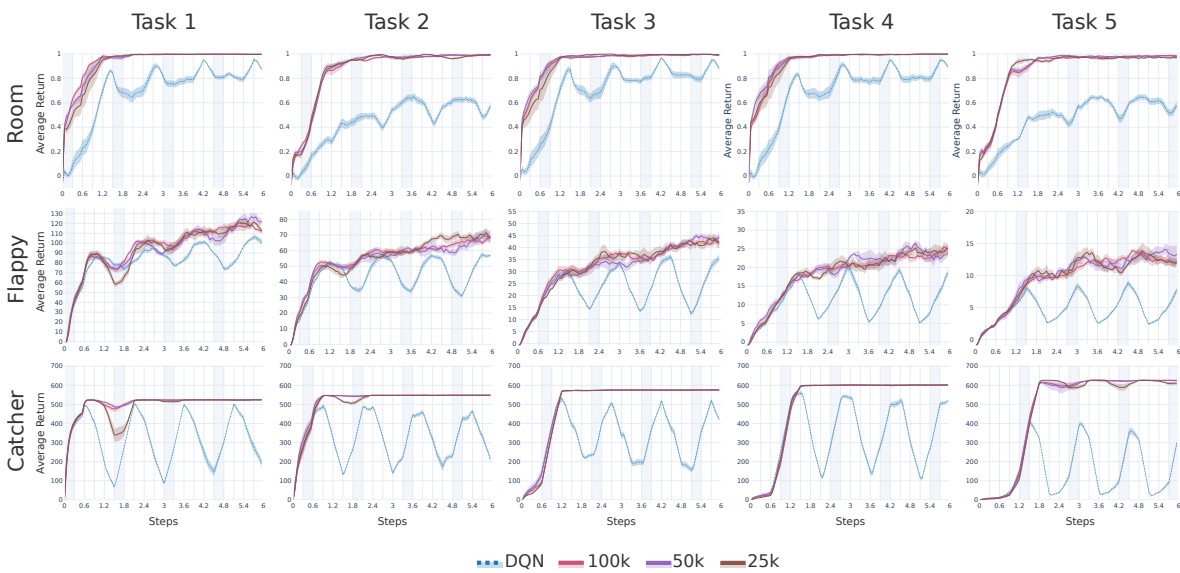

Figure 8: Learning curves of Qreg+NWLU total return $\mathcal{G}$ using various $N_{\mathrm{RRB}}$ values (size of RRB memory), averaged over 5 seeds.

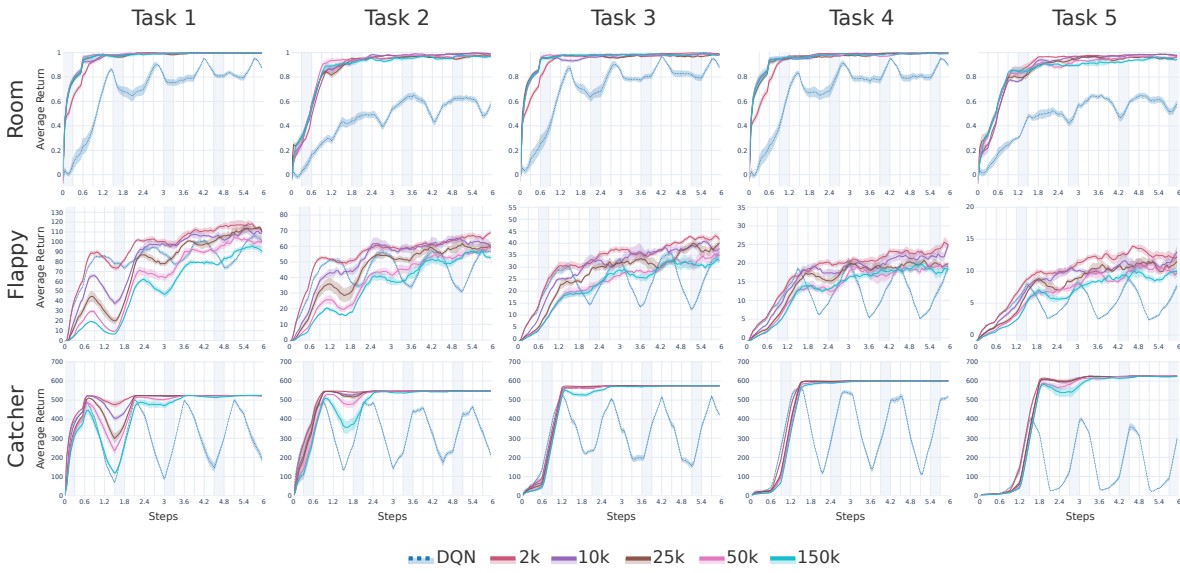

Figure 9: Learning curves of Qreg+NWLU total return $\mathcal{G}$ using various $F_{\mathrm{RUF}}$ values (rate at which samples in RRB are updated), averaged over 5 seeds.

# G    Additional Main Results

Section G.1 presents grand average tables for the Flappy and Catcher task sequences (main and ablation results). Section G.2 contains Q-value norm plots for all Qreg algorithms. Section G.3 presents per-task transfer metrics showing average scores of $\mathcal{F}$ and $\mathcal{W}$ for each training-evaluation task pair.

## G.1    Grand Average Metrics

### G.1.1    Flappy

Table 9: Final transfer grand average $\bar{\mathcal{F}}$ with standard error for Flappy. Near-zero values are indicated by *.

| Algorithm | Task 1 | Task 2 | Task 3 | Task 4 | Task 5 |
|---|---|---|---|---|---|
| DQN | -0.51 ± 0.05 | -0.98 ± 0.06 | 1.05 ± 0.07 | 1.45 ± 0.06 | 0.96 ± 0.06 |
| DDQN | -0.47 ± 0.05 | -1.02 ± 0.04 | 1.05 ± 0.06 | 1.39 ± 0.06 | 1.05 ± 0.05 |
| PM | 0.82 ± 0.05 | 0.41 ± 0.08 | 0.36 ± 0.12 | 0.03 ± 0.08 | 0.23 ± 0.09 |
| PackNet | -0.44 ± 0.09 | -0.52 ± 0.04 | 1.19 ± 0.06 | 1.13 ± 0.04 | 0.50 ± 0.06 |
| EWC | 0.83 ± 0.04 | 0.51 ± 0.05 | 0.49 ± 0.07 | 0.02 ± 0.08 | -0.00* ± 0.06 |
| L2 | 0.89 ± 0.07 | 0.04 ± 0.13 | 0.32 ± 0.06 | 0.30 ± 0.08 | 0.32 ± 0.10 |
| MER | -0.45 ± 0.06 | -0.50 ± 0.06 | 1.02 ± 0.08 | 1.24 ± 0.04 | 0.67 ± 0.07 |
| Qreg | 0.58 ± 0.05 | 0.41 ± 0.05 | 0.48 ± 0.05 | 0.25 ± 0.06 | 0.23 ± 0.06 |
| Qreg+NWLU | 0.57 ± 0.04 | 0.48 ± 0.05 | 0.56 ± 0.05 | 0.24 ± 0.05 | 0.17 ± 0.05 |

Table 10: Worst transfer grand average $\bar{\mathcal{W}}$ with standard error for Flappy.

| Algorithm | Task 1 | Task 2 | Task 3 | Task 4 | Task 5 |
|---|---|---|---|---|---|
| DQN | -0.98 ± 0.05 | -1.33 ± 0.03 | -0.14 ± 0.02 | 0.26 ± 0.03 | -0.01 ± 0.03 |
| DDQN | -1.00 ± 0.05 | -1.44 ± 0.04 | -0.19 ± 0.03 | 0.24 ± 0.02 | 0.01 ± 0.03 |
| PM | -0.21 ± 0.02 | -0.18 ± 0.05 | -0.24 ± 0.07 | -0.36 ± 0.04 | -0.24 ± 0.06 |
| PackNet | -1.01 ± 0.06 | -1.07 ± 0.03 | -0.07 ± 0.03 | 0.06 ± 0.02 | -0.27 ± 0.04 |
| EWC | -0.24 ± 0.02 | -0.15 ± 0.05 | -0.12 ± 0.03 | -0.37 ± 0.04 | -0.35 ± 0.05 |
| L2 | -0.24 ± 0.04 | -0.43 ± 0.10 | -0.20 ± 0.04 | -0.23 ± 0.05 | -0.19 ± 0.06 |
| MER | -0.65 ± 0.06 | -0.95 ± 0.06 | -0.12 ± 0.04 | 0.18 ± 0.03 | -0.08 ± 0.04 |
| Qreg | -0.15 ± 0.03 | -0.16 ± 0.04 | -0.12 ± 0.04 | -0.25 ± 0.04 | -0.23 ± 0.03 |
| Qreg+NWLU | -0.16 ± 0.03 | -0.12 ± 0.03 | -0.11 ± 0.03 | -0.21 ± 0.03 | -0.22 ± 0.03 |

Table 11: Return grand average $\bar{\mathcal{G}}$ with standard error for Flappy.

| Algorithm | Task 1 | Task 2 | Task 3 | Task 4 | Task 5 |
|---|---|---|---|---|---|
| DQN | 81.36 ± 0.69 | 42.73 ± 0.53 | 22.63 ± 0.26 | 11.12 ± 0.16 | 4.69 ± 0.06 |
| DDQN | 81.87 ± 0.98 | 43.07 ± 0.57 | 22.17 ± 0.33 | 10.41 ± 0.07 | 4.47 ± 0.07 |
| PM | 88.82 ± 0.88 | 48.47 ± 0.73 | 26.93 ± 0.36 | 14.29 ± 0.23 | 7.29 ± 0.17 |
| PackNet | 73.46 ± 1.22 | 45.00 ± 0.65 | 23.15 ± 0.41 | 11.75 ± 0.24 | 5.15 ± 0.06 |
| EWC | 69.39 ± 0.82 | 35.73 ± 1.57 | 14.98 ± 0.87 | 7.22 ± 0.49 | 3.30 ± 0.26 |
| L2 | 61.62 ± 2.21 | 26.21 ± 1.27 | 10.42 ± 0.53 | 5.45 ± 0.28 | 2.68 ± 0.15 |
| MER | 79.07 ± 0.86 | 42.83 ± 0.35 | 22.69 ± 0.39 | 10.99 ± 0.24 | 4.86 ± 0.07 |
| Qreg | 71.14 ± 2.49 | 49.86 ± 0.49 | 30.41 ± 0.70 | 17.69 ± 0.32 | 8.83 ± 0.18 |
| Qreg+NWLU | 94.53 ± 0.99 | 54.31 ± 0.55 | 32.29 ± 0.48 | 18.01 ± 0.28 | 9.55 ± 0.21 |

Table 12: Ablation final transfer grand average $\bar{\mathcal{F}}$ with standard error for Flappy.

| Algorithm | Task 1 | Task 2 | Task 3 | Task 4 | Task 5 |
|---|---|---|---|---|---|
| DQN | -0.51 ± 0.05 | -0.98 ± 0.06 | 1.05 ± 0.07 | 1.45 ± 0.06 | 0.96 ± 0.06 |
| DDQN | -0.47 ± 0.05 | -1.02 ± 0.04 | 1.05 ± 0.06 | 1.39 ± 0.06 | 1.05 ± 0.05 |
| Qreg | 0.58 ± 0.05 | 0.41 ± 0.05 | 0.48 ± 0.05 | 0.25 ± 0.06 | 0.23 ± 0.06 |
| U | 0.59 ± 0.04 | 0.45 ± 0.05 | 0.45 ± 0.05 | 0.31 ± 0.06 | 0.20 ± 0.04 |
| L | 0.73 ± 0.02 | 0.53 ± 0.05 | 0.35 ± 0.05 | 0.19 ± 0.04 | 0.10 ± 0.07 |
| LU | 0.46 ± 0.02 | 0.50 ± 0.03 | 0.55 ± 0.04 | 0.35 ± 0.04 | 0.22 ± 0.07 |
| NWL | 0.44 ± 0.04 | 0.58 ± 0.03 | 0.54 ± 0.06 | 0.31 ± 0.05 | 0.14 ± 0.03 |
| NWLU | 0.57 ± 0.04 | 0.48 ± 0.05 | 0.56 ± 0.05 | 0.24 ± 0.05 | 0.17 ± 0.05 |

Table 13: Ablation worst transfer grand average $\bar{\mathcal{W}}$ with standard error for Flappy.

| Algorithm | Task 1 | Task 2 | Task 3 | Task 4 | Task 5 |
|---|---|---|---|---|---|
| DQN | -0.98 ± 0.05 | -1.33 ± 0.03 | -0.14 ± 0.02 | 0.26 ± 0.03 | -0.01 ± 0.03 |
| DDQN | -1.00 ± 0.05 | -1.44 ± 0.04 | -0.19 ± 0.03 | 0.24 ± 0.02 | 0.01 ± 0.03 |
| Qreg | -0.15 ± 0.03 | -0.16 ± 0.04 | -0.12 ± 0.04 | -0.25 ± 0.04 | -0.23 ± 0.03 |
| U | -0.18 ± 0.03 | -0.13 ± 0.02 | -0.15 ± 0.02 | -0.24 ± 0.04 | -0.27 ± 0.04 |
| L | -0.18 ± 0.02 | -0.04 ± 0.03 | -0.10 ± 0.03 | -0.33 ± 0.03 | -0.33 ± 0.03 |
| LU | -0.25 ± 0.02 | -0.11 ± 0.02 | -0.06 ± 0.03 | -0.17 ± 0.03 | -0.26 ± 0.04 |
| NWL | -0.12 ± 0.02 | -0.07 ± 0.02 | -0.04 ± 0.03 | -0.19 ± 0.03 | -0.34 ± 0.03 |
| NWLU | -0.16 ± 0.03 | -0.12 ± 0.03 | -0.11 ± 0.03 | -0.21 ± 0.03 | -0.22 ± 0.03 |

Table 14: Ablation return grand average $\bar{\mathcal{G}}$ with standard error for Flappy.

| Algorithm | Task 1 | Task 2 | Task 3 | Task 4 | Task 5 |
|---|---|---|---|---|---|
| DQN | 81.36 ± 0.69 | 42.73 ± 0.53 | 22.63 ± 0.26 | 11.12 ± 0.16 | 4.69 ± 0.06 |
| DDQN | 81.87 ± 0.98 | 43.07 ± 0.57 | 22.17 ± 0.33 | 10.41 ± 0.07 | 4.47 ± 0.07 |
| Qreg | 71.14 ± 2.49 | 49.86 ± 0.49 | 30.41 ± 0.70 | 17.69 ± 0.32 | 8.83 ± 0.18 |
| U | 75.43 ± 2.81 | 48.94 ± 0.95 | 29.45 ± 0.67 | 17.76 ± 0.40 | 8.85 ± 0.15 |
| L | 52.34 ± 1.07 | 37.54 ± 0.31 | 22.30 ± 0.37 | 12.93 ± 0.28 | 6.04 ± 0.15 |
| LU | 87.79 ± 0.77 | 53.93 ± 0.59 | 31.47 ± 0.56 | 17.98 ± 0.36 | 9.69 ± 0.23 |
| NWL | 31.69 ± 0.79 | 23.60 ± 0.43 | 17.52 ± 0.35 | 10.71 ± 0.20 | 5.19 ± 0.09 |
| NWLU | 94.53 ± 0.99 | 54.31 ± 0.55 | 32.29 ± 0.48 | 18.01 ± 0.28 | 9.55 ± 0.21 |

### G.1.2 Catcher

Table 15: Final transfer grand average $\bar{\mathcal{F}}$ with standard error for Catcher. Near-zero values are indicated by $^*$.

| Algorithm | Task 1 | Task 2 | Task 3 | Task 4 | Task 5 |
|---|---|---|---|---|---|
| DQN | $0.40 \pm 0.09$ | $-0.00^* \pm 0.07$ | $0.86 \pm 0.07$ | $0.53 \pm 0.09$ | $-0.30 \pm 0.07$ |
| DDQN | $0.17 \pm 0.09$ | $-0.25 \pm 0.05$ | $0.82 \pm 0.06$ | $0.85 \pm 0.06$ | $0.06 \pm 0.08$ |
| PM | $0.85 \pm 0.06$ | $0.31 \pm 0.05$ | $0.40 \pm 0.03$ | $0.43 \pm 0.04$ | $0.35 \pm 0.06$ |
| PackNet | $-0.02 \pm 0.05$ | $-0.23 \pm 0.08$ | $1.13 \pm 0.06$ | $1.01 \pm 0.06$ | $-0.10 \pm 0.07$ |
| EWC | $0.74 \pm 0.06$ | $0.31 \pm 0.05$ | $0.33 \pm 0.05$ | $0.24 \pm 0.05$ | $0.17 \pm 0.07$ |
| L2 | $0.80 \pm 0.13$ | $-0.20 \pm 0.14$ | $0.15 \pm 0.07$ | $0.41 \pm 0.13$ | $0.44 \pm 0.17$ |
| MER | $0.08 \pm 0.06$ | $-0.05 \pm 0.05$ | $1.08 \pm 0.04$ | $0.99 \pm 0.05$ | $-0.38 \pm 0.04$ |
| Qreg | $0.84 \pm 0.04$ | $0.38 \pm 0.02$ | $0.57 \pm 0.02$ | $0.44 \pm 0.03$ | $0.28 \pm 0.02$ |
| Qreg+NWLU | $0.86 \pm 0.05$ | $0.29 \pm 0.01$ | $0.51 \pm 0.03$ | $0.49 \pm 0.02$ | $0.36 \pm 0.02$ |

Table 16: Worst transfer grand average $\bar{\mathcal{W}}$ with standard error for Catcher. Near-zero values are indicated by $^*$.

| Algorithm | Task 1 | Task 2 | Task 3 | Task 4 | Task 5 |
|---|---|---|---|---|---|
| DQN | $-0.92 \pm 0.05$ | $-1.25 \pm 0.05$ | $-0.32 \pm 0.04$ | $-0.71 \pm 0.06$ | $-1.24 \pm 0.05$ |
| DDQN | $-0.93 \pm 0.05$ | $-1.47 \pm 0.03$ | $-0.38 \pm 0.04$ | $-0.58 \pm 0.04$ | $-1.21 \pm 0.07$ |
| PM | $-0.05 \pm 0.04$ | $-0.00^* \pm 0.04$ | $0.01 \pm 0.03$ | $-0.02 \pm 0.03$ | $-0.06 \pm 0.03$ |
| PackNet | $-1.08 \pm 0.04$ | $-1.32 \pm 0.06$ | $-0.09 \pm 0.04$ | $-0.17 \pm 0.03$ | $-0.85 \pm 0.07$ |
| EWC | $-0.06 \pm 0.03$ | $-0.01 \pm 0.03$ | $0.01 \pm 0.02$ | $-0.06 \pm 0.04$ | $-0.13 \pm 0.04$ |
| L2 | $-0.17 \pm 0.06$ | $-0.77 \pm 0.09$ | $-0.29 \pm 0.04$ | $-0.25 \pm 0.08$ | $-0.38 \pm 0.12$ |
| MER | $-0.98 \pm 0.04$ | $-1.40 \pm 0.03$ | $-0.08 \pm 0.02$ | $-0.43 \pm 0.03$ | $-1.37 \pm 0.04$ |
| Qreg | $0.03 \pm 0.01$ | $0.07 \pm 0.01$ | $0.10 \pm 0.01$ | $0.03 \pm 0.01$ | $0.03 \pm 0.01$ |
| Qreg+NWLU | $0.06 \pm 0.01$ | $0.08 \pm 0.00^*$ | $0.11 \pm 0.01$ | $0.06 \pm 0.01$ | $0.07 \pm 0.01$ |

Table 17: Return grand average $\bar{\mathcal{G}}$ with standard error for Catcher.

| Algorithm | Task 1 | Task 2 | Task 3 | Task 4 | Task 5 |
|---|---|---|---|---|---|
| DQN | $314.46 \pm 5.49$ | $326.56 \pm 5.16$ | $313.30 \pm 7.04$ | $318.40 \pm 6.26$ | $149.09 \pm 4.75$ |
| DDQN | $319.80 \pm 5.37$ | $316.64 \pm 7.42$ | $285.15 \pm 4.67$ | $279.02 \pm 4.70$ | $130.57 \pm 2.52$ |
| PM | $492.31 \pm 2.55$ | $487.62 \pm 4.07$ | $463.62 \pm 3.06$ | $470.46 \pm 2.96$ | $418.77 \pm 3.53$ |
| PackNet | $344.59 \pm 8.03$ | $416.85 \pm 4.69$ | $410.98 \pm 8.03$ | $391.26 \pm 6.11$ | $202.22 \pm 4.84$ |
| EWC | $422.11 \pm 30.65$ | $464.74 \pm 24.25$ | $418.03 \pm 26.02$ | $312.65 \pm 40.54$ | $52.77 \pm 10.59$ |
| L2 | $422.51 \pm 15.89$ | $418.59 \pm 17.13$ | $220.87 \pm 19.64$ | $110.23 \pm 12.54$ | $31.32 \pm 3.49$ |
| MER | $329.08 \pm 2.25$ | $374.88 \pm 4.10$ | $355.91 \pm 3.70$ | $341.55 \pm 2.03$ | $159.39 \pm 1.68$ |
| Qreg | $479.63 \pm 4.47$ | $507.87 \pm 3.98$ | $495.61 \pm 3.18$ | $505.66 \pm 1.98$ | $479.24 \pm 2.01$ |
| Qreg+NWLU | $500.48 \pm 1.87$ | $512.28 \pm 2.18$ | $495.57 \pm 1.30$ | $497.51 \pm 1.51$ | $476.28 \pm 2.81$ |

Table 18: Ablation final transfer grand average $\bar{\mathcal{F}}$ with standard error for Catcher. Near-zero values are indicated by $^*$.

| Algorithm | Task 1 | Task 2 | Task 3 | Task 4 | Task 5 |
|---|---|---|---|---|---|
| DQN | 0.40 ± 0.09 | -0.00$^*$ ± 0.07 | 0.86 ± 0.07 | 0.53 ± 0.09 | -0.30 ± 0.07 |
| DDQN | 0.17 ± 0.09 | -0.25 ± 0.05 | 0.82 ± 0.06 | 0.85 ± 0.06 | 0.06 ± 0.08 |
| Qreg | 0.84 ± 0.04 | 0.38 ± 0.02 | 0.57 ± 0.02 | 0.44 ± 0.03 | 0.28 ± 0.02 |
| U | 0.87 ± 0.03 | 0.40 ± 0.03 | 0.58 ± 0.02 | 0.40 ± 0.02 | 0.26 ± 0.02 |
| L | 1.12 ± 0.04 | 0.59 ± 0.04 | 0.33 ± 0.03 | 0.21 ± 0.02 | 0.24 ± 0.02 |
| LU | 0.84 ± 0.03 | 0.39 ± 0.02 | 0.60 ± 0.02 | 0.41 ± 0.03 | 0.26 ± 0.02 |
| NWL | 0.76 ± 0.03 | 0.74 ± 0.03 | 0.55 ± 0.02 | 0.22 ± 0.03 | 0.17 ± 0.02 |
| NWLU | 0.86 ± 0.05 | 0.29 ± 0.01 | 0.51 ± 0.03 | 0.49 ± 0.02 | 0.36 ± 0.02 |

Table 19: Ablation worst transfer grand average $\bar{\mathcal{W}}$ with standard error for Catcher. Near-zero values are indicated by $^*$.

| Algorithm | Task 1 | Task 2 | Task 3 | Task 4 | Task 5 |
|---|---|---|---|---|---|
| DQN | -0.92 ± 0.05 | -1.25 ± 0.05 | -0.32 ± 0.04 | -0.71 ± 0.06 | -1.24 ± 0.05 |
| DDQN | -0.93 ± 0.05 | -1.47 ± 0.03 | -0.38 ± 0.04 | -0.58 ± 0.04 | -1.21 ± 0.07 |
| Qreg | 0.03 ± 0.01 | 0.07 ± 0.01 | 0.10 ± 0.01 | 0.03 ± 0.01 | 0.03 ± 0.01 |
| U | 0.04 ± 0.01 | 0.08 ± 0.01 | 0.11 ± 0.01 | 0.03 ± 0.01 | 0.02 ± 0.02 |
| L | 0.02 ± 0.01 | 0.10 ± 0.02 | -0.04 ± 0.02 | -0.21 ± 0.02 | -0.16 ± 0.01 |
| LU | 0.04 ± 0.00$^*$ | 0.08 ± 0.01 | 0.12 ± 0.01 | 0.02 ± 0.02 | 0.03 ± 0.01 |
| NWL | -0.07 ± 0.01 | 0.09 ± 0.02 | 0.02 ± 0.02 | -0.24 ± 0.02 | -0.23 ± 0.02 |
| NWLU | 0.06 ± 0.01 | 0.08 ± 0.00$^*$ | 0.11 ± 0.01 | 0.06 ± 0.01 | 0.07 ± 0.01 |

Table 20: Ablation return grand average $\bar{\mathcal{G}}$ with standard error for Catcher.

| Algorithm | Task 1 | Task 2 | Task 3 | Task 4 | Task 5 |
|---|---|---|---|---|---|
| DQN | 314.46 ± 5.49 | 326.56 ± 5.16 | 313.30 ± 7.04 | 318.40 ± 6.26 | 149.09 ± 4.75 |
| DDQN | 319.80 ± 5.37 | 316.64 ± 7.42 | 285.15 ± 4.67 | 279.02 ± 4.70 | 130.57 ± 2.52 |
| Qreg | 479.63 ± 4.47 | 507.87 ± 3.98 | 495.61 ± 3.18 | 505.66 ± 1.98 | 479.24 ± 2.01 |
| U | 481.42 ± 4.97 | 510.56 ± 2.90 | 499.32 ± 2.83 | 505.59 ± 2.05 | 472.52 ± 2.63 |
| L | 366.26 ± 5.84 | 472.13 ± 3.38 | 483.47 ± 2.01 | 483.81 ± 3.30 | 436.27 ± 9.63 |
| LU | 481.76 ± 4.87 | 511.14 ± 2.00 | 501.34 ± 1.67 | 506.34 ± 1.48 | 477.75 ± 3.27 |
| NWL | 336.63 ± 7.97 | 412.78 ± 6.34 | 445.24 ± 5.02 | 474.15 ± 2.96 | 425.12 ± 7.49 |
| NWLU | 500.48 ± 1.87 | 512.28 ± 2.18 | 495.57 ± 1.30 | 497.51 ± 1.51 | 476.28 ± 2.81 |

## G.2 Q-Value Norms

Below are the Q-value norm plots for different Qreg variations throughout training.

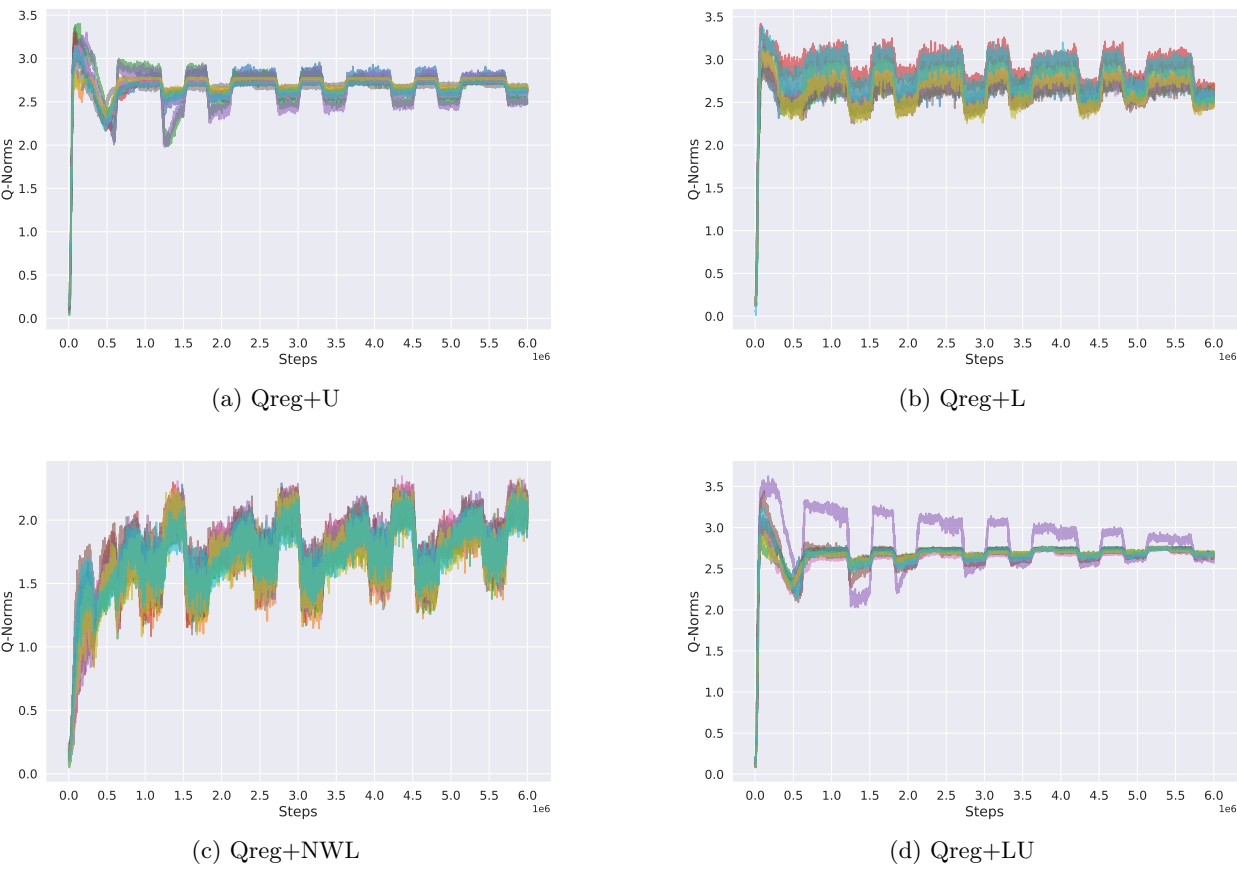

Figure 10: Room comparison of Q-value norms between Qreg variants. Color indicates each unique run.

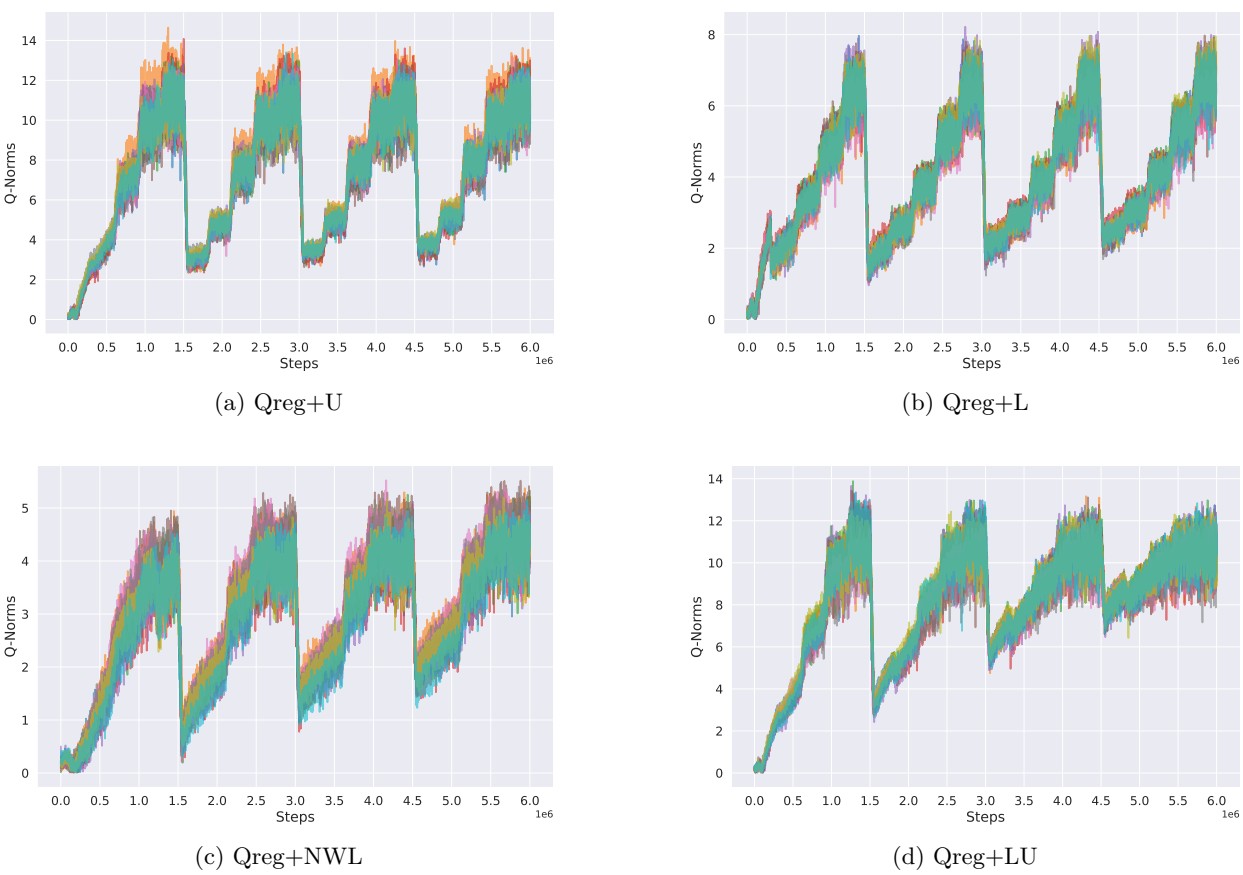

Figure 11: Flappy comparison of Q-value norms between Qreg variants. Color indicates each unique run.

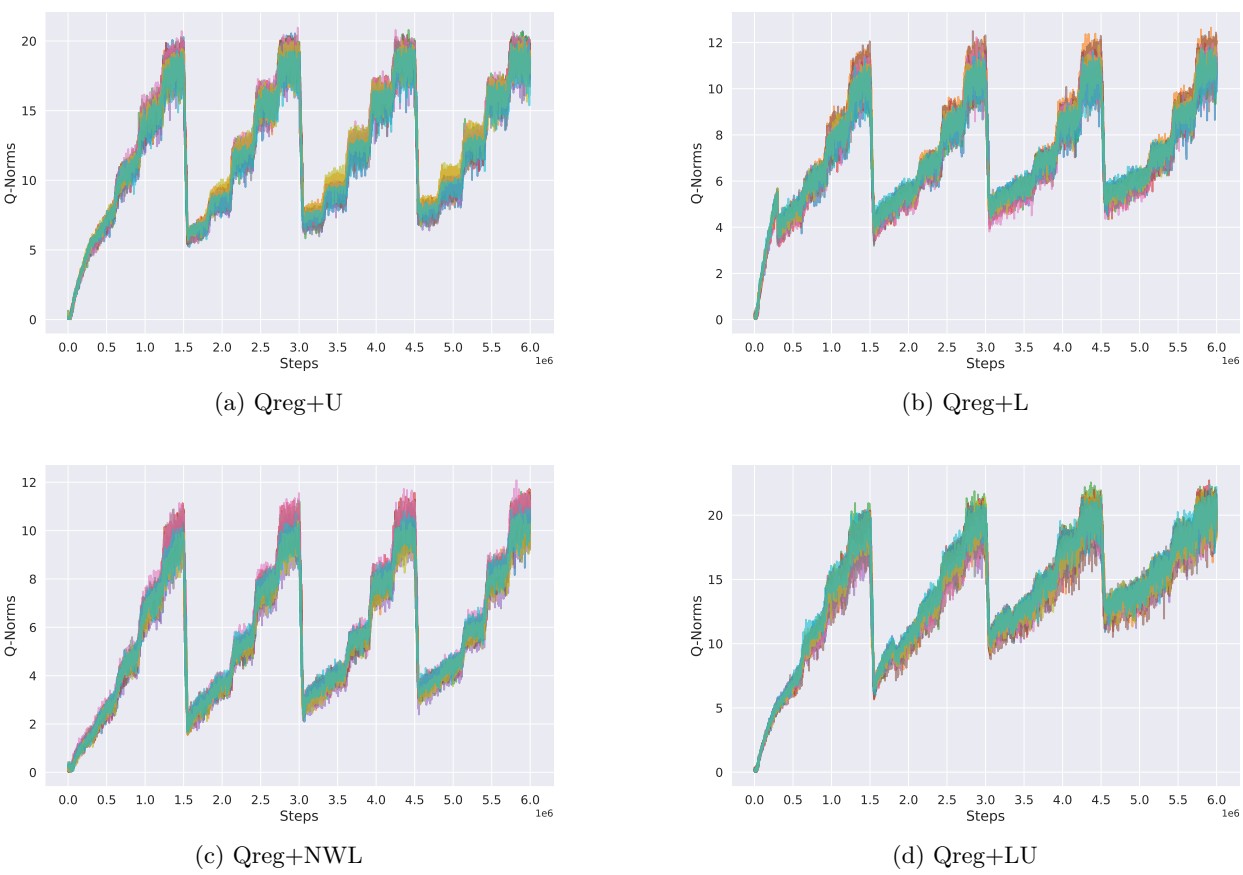

(a) Qreg+U

(b) Qreg+L

(c) Qreg+NWL

(d) Qreg+LU

Figure 12: Catcher comparison of Q-value norms between Qreg variants. Color indicates each unique run.

### G.3 Per-Task Transfer Metrics

In these tables, columns represent evaluation tasks, while rows represent training tasks with their cycle numbers. The right most column shows the grand average over evaluation tasks for each training task, while the bottom row shows the grand average over training tasks for each evaluation task.

Table 21: Room $\mathcal{F}$ for DQN.

| | T1 | T2 | T3 | T4 | T5 | Avg ± SEM |
|---|---|---|---|---|---|---|
| T1-C1 | 1.03 ± 0.30 | 0.03 ± 0.26 | 0.99 ± 0.52 | 0.82 ± 0.37 | 0.49 ± 0.59 | -0.08 ± 0.13 |
| T2-C1 | 1.47 ± 0.33 | 1.52 ± 0.28 | 1.45 ± 0.34 | 1.56 ± 0.33 | 1.38 ± 0.30 | 0.45 ± 0.11 |
| T3-C1 | 1.54 ± 0.35 | 1.53 ± 0.34 | 1.58 ± 0.44 | 1.52 ± 0.42 | 1.31 ± 0.33 | 0.38 ± 0.18 |
| T4-C1 | 3.08 ± 0.42 | 0.64 ± 0.35 | 2.92 ± 0.38 | 2.99 ± 0.39 | 0.73 ± 0.38 | 0.78 ± 0.16 |
| T5-C1 | 2.20 ± 0.32 | 1.34 ± 0.23 | 2.11 ± 0.31 | 1.90 ± 0.31 | 1.85 ± 0.29 | 0.61 ± 0.09 |
| T1-C2 | -1.76 ± 0.36 | 0.29 ± 0.31 | -1.63 ± 0.36 | -1.61 ± 0.35 | 0.46 ± 0.30 | – |
| T2-C2 | -0.42 ± 0.26 | 0.60 ± 0.29 | -0.61 ± 0.19 | -0.10 ± 0.16 | 0.26 ± 0.27 | – |
| T3-C2 | 0.43 ± 0.53 | 0.18 ± 0.25 | 0.31 ± 0.53 | 0.16 ± 0.52 | 0.07 ± 0.27 | – |
| T4-C2 | 1.49 ± 0.22 | -0.73 ± 0.21 | 1.54 ± 0.13 | 1.46 ± 0.17 | -0.66 ± 0.29 | – |
| T5-C2 | 0.16 ± 0.43 | 0.85 ± 0.29 | 0.42 ± 0.36 | 0.44 ± 0.31 | 1.01 ± 0.26 | – |
| T1-C3 | -0.99 ± 0.13 | 1.04 ± 0.26 | -0.95 ± 0.13 | -0.83 ± 0.16 | 0.92 ± 0.25 | – |
| T2-C3 | 0.20 ± 0.22 | 0.66 ± 0.22 | -0.01 ± 0.18 | -0.07 ± 0.19 | 0.44 ± 0.33 | – |
| T3-C3 | 0.14 ± 0.15 | -0.34 ± 0.19 | 0.27 ± 0.21 | 0.01 ± 0.21 | -0.19 ± 0.25 | – |
| T4-C3 | 1.20 ± 0.21 | -1.37 ± 0.27 | 1.18 ± 0.15 | 1.28 ± 0.21 | -1.37 ± 0.26 | – |
| T5-C3 | -0.47 ± 0.21 | 0.29 ± 0.35 | -0.39 ± 0.17 | -0.41 ± 0.16 | 0.40 ± 0.20 | – |
| T1-C4 | -0.55 ± 0.08 | 0.86 ± 0.29 | -0.58 ± 0.12 | -0.78 ± 0.15 | 1.13 ± 0.33 | – |
| T2-C4 | 0.20 ± 0.18 | 0.39 ± 0.24 | -0.03 ± 0.16 | 0.14 ± 0.16 | 0.02 ± 0.26 | – |
| T3-C4 | -0.35 ± 0.13 | 0.02 ± 0.30 | -0.32 ± 0.17 | -0.08 ± 0.19 | -0.25 ± 0.31 | – |
| T4-C4 | 1.13 ± 0.10 | -1.27 ± 0.31 | 1.13 ± 0.15 | 1.11 ± 0.12 | -0.97 ± 0.32 | – |
| T5-C4 | -0.30 ± 0.18 | 0.62 ± 0.20 | -0.26 ± 0.28 | -0.11 ± 0.26 | 0.48 ± 0.29 | – |
| Avg ± SEM | 0.47 ± 0.01 | 0.36 ± 0.02 | 0.46 ± 0.01 | 0.47 ± 0.01 | 0.37 ± 0.03 | 0.43 ± 0.01 |

Table 22: Room $\mathcal{W}$ for DQN.

| | T1 | T2 | T3 | T4 | T5 | Avg ± SEM |
|---|---|---|---|---|---|---|
| T1-C1 | -0.06 ± 0.06 | -0.43 ± 0.19 | -0.21 ± 0.16 | -0.15 ± 0.09 | -0.55 ± 0.29 | -0.50 ± 0.07 |
| T2-C1 | 0.47 ± 0.20 | 0.40 ± 0.12 | 0.52 ± 0.19 | 0.50 ± 0.15 | 0.28 ± 0.15 | -0.06 ± 0.07 |
| T3-C1 | 0.34 ± 0.11 | 0.51 ± 0.15 | 0.16 ± 0.14 | 0.20 ± 0.13 | 0.36 ± 0.15 | -0.22 ± 0.08 |
| T4-C1 | 0.50 ± 0.15 | -0.05 ± 0.17 | 0.43 ± 0.14 | 0.44 ± 0.15 | -0.19 ± 0.17 | -0.27 ± 0.08 |
| T5-C1 | 0.52 ± 0.09 | -0.22 ± 0.10 | 0.54 ± 0.10 | 0.56 ± 0.05 | 0.04 ± 0.18 | -0.24 ± 0.05 |
| T1-C2 | -1.76 ± 0.36 | -0.15 ± 0.23 | -1.64 ± 0.36 | -1.63 ± 0.34 | -0.05 ± 0.20 | – |
| T2-C2 | -0.60 ± 0.21 | -0.07 ± 0.14 | -0.63 ± 0.18 | -0.28 ± 0.13 | -0.23 ± 0.21 | – |
| T3-C2 | -0.34 ± 0.29 | -0.28 ± 0.17 | -0.62 ± 0.24 | -0.67 ± 0.30 | -0.27 ± 0.15 | – |
| T4-C2 | 0.08 ± 0.08 | -0.74 ± 0.21 | 0.03 ± 0.07 | 0.06 ± 0.08 | -0.73 ± 0.26 | – |
| T5-C2 | -0.19 ± 0.22 | -0.60 ± 0.10 | 0.05 ± 0.20 | 0.18 ± 0.23 | -0.52 ± 0.15 | – |
| T1-C3 | -0.99 ± 0.13 | 0.39 ± 0.12 | -0.95 ± 0.13 | -0.83 ± 0.16 | 0.26 ± 0.26 | – |
| T2-C3 | -0.23 ± 0.12 | 0.03 ± 0.08 | -0.28 ± 0.08 | -0.33 ± 0.09 | -0.09 ± 0.20 | – |
| T3-C3 | -0.15 ± 0.07 | -0.55 ± 0.13 | -0.15 ± 0.12 | -0.27 ± 0.12 | -0.48 ± 0.18 | – |
| T4-C3 | -0.02 ± 0.02 | -1.43 ± 0.25 | -0.01 ± 0.03 | -0.07 ± 0.07 | -1.41 ± 0.24 | – |
| T5-C3 | -0.47 ± 0.21 | -1.00 ± 0.23 | -0.39 ± 0.17 | -0.41 ± 0.16 | -0.78 ± 0.17 | – |
| T1-C4 | -0.55 ± 0.08 | 0.22 ± 0.18 | -0.58 ± 0.12 | -0.78 ± 0.15 | 0.36 ± 0.13 | – |
| T2-C4 | -0.18 ± 0.09 | 0.07 ± 0.17 | -0.26 ± 0.11 | -0.18 ± 0.10 | -0.20 ± 0.21 | – |
| T3-C4 | -0.41 ± 0.12 | -0.38 ± 0.22 | -0.41 ± 0.15 | -0.19 ± 0.15 | -0.72 ± 0.23 | – |
| T4-C4 | 0.00 ± 0.00 | -1.30 ± 0.30 | -0.00 ± 0.02 | 0.00 ± 0.00 | -1.07 ± 0.29 | – |
| T5-C4 | -0.33 ± 0.16 | -0.63 ± 0.14 | -0.41 ± 0.22 | -0.18 ± 0.23 | -0.59 ± 0.23 | – |
| Avg ± SEM | -0.22 ± 0.03 | -0.31 ± 0.02 | -0.24 ± 0.02 | -0.20 ± 0.03 | -0.33 ± 0.04 | -0.26 ± 0.02 |

Table 23: Flappy $\mathcal{F}$ for DQN.

| | T1 | T2 | T3 | T4 | T5 | Avg ± SEM |
|---|---|---|---|---|---|---|
| T1-C1 | 2.90 ± 0.13 | 2.09 ± 0.19 | 1.46 ± 0.13 | 1.24 ± 0.13 | 1.59 ± 0.12 | -0.51 ± 0.05 |
| T2-C1 | 1.80 ± 0.09 | 1.73 ± 0.18 | 1.10 ± 0.16 | 0.86 ± 0.11 | 0.89 ± 0.08 | -0.98 ± 0.06 |
| T3-C1 | 2.22 ± 0.08 | 2.36 ± 0.10 | 2.29 ± 0.18 | 1.34 ± 0.14 | 1.34 ± 0.14 | 1.05 ± 0.07 |
| T4-C1 | 0.43 ± 0.12 | 1.08 ± 0.23 | 1.53 ± 0.21 | 1.56 ± 0.18 | 1.23 ± 0.19 | 1.45 ± 0.06 |
| T5-C1 | -0.47 ± 0.11 | 0.09 ± 0.14 | 0.35 ± 0.26 | 1.71 ± 0.19 | 1.68 ± 0.31 | 0.96 ± 0.06 |
| T1-C2 | -0.44 ± 0.08 | -1.54 ± 0.22 | -1.59 ± 0.25 | -1.07 ± 0.24 | -0.56 ± 0.19 | – |
| T2-C2 | 0.42 ± 0.19 | -0.83 ± 0.27 | -1.54 ± 0.23 | -2.51 ± 0.14 | -2.39 ± 0.17 | – |
| T3-C2 | 0.85 ± 0.20 | 1.60 ± 0.26 | 0.99 ± 0.15 | 0.28 ± 0.20 | -0.37 ± 0.23 | – |
| T4-C2 | 0.40 ± 0.17 | 1.33 ± 0.16 | 1.73 ± 0.24 | 1.82 ± 0.18 | 1.29 ± 0.22 | – |
| T5-C2 | -0.49 ± 0.21 | -0.11 ± 0.31 | 0.97 ± 0.20 | 2.33 ± 0.36 | 2.54 ± 0.27 | – |
| T1-C3 | -0.81 ± 0.16 | -1.59 ± 0.23 | -1.77 ± 0.21 | -1.57 ± 0.21 | -0.81 ± 0.18 | – |
| T2-C3 | 0.15 ± 0.17 | -1.23 ± 0.27 | -2.07 ± 0.15 | -3.23 ± 0.38 | -2.99 ± 0.34 | – |
| T3-C3 | 1.17 ± 0.16 | 1.45 ± 0.23 | 1.08 ± 0.17 | 0.52 ± 0.25 | -0.01 ± 0.16 | – |
| T4-C3 | 0.41 ± 0.18 | 1.54 ± 0.16 | 2.44 ± 0.15 | 2.07 ± 0.24 | 1.46 ± 0.25 | – |
| T5-C3 | -0.65 ± 0.18 | -0.04 ± 0.25 | 0.94 ± 0.17 | 1.82 ± 0.25 | 2.63 ± 0.22 | – |
| T1-C4 | -1.16 ± 0.12 | -1.90 ± 0.21 | -2.22 ± 0.19 | -1.68 ± 0.22 | -0.85 ± 0.21 | – |
| T2-C4 | -0.04 ± 0.11 | -1.44 ± 0.17 | -2.26 ± 0.24 | -2.78 ± 0.19 | -3.23 ± 0.26 | – |
| T3-C4 | 1.28 ± 0.16 | 1.59 ± 0.21 | 0.88 ± 0.30 | 0.14 ± 0.23 | -0.21 ± 0.19 | – |
| T4-C4 | 0.94 ± 0.15 | 1.86 ± 0.11 | 2.38 ± 0.27 | 2.06 ± 0.24 | 1.43 ± 0.21 | – |
| T5-C4 | -0.30 ± 0.17 | 0.11 ± 0.20 | 1.21 ± 0.37 | 2.33 ± 0.34 | 2.46 ± 0.35 | – |
| Avg ± SEM | 0.43 ± 0.01 | 0.41 ± 0.01 | 0.40 ± 0.02 | 0.37 ± 0.02 | 0.36 ± 0.02 | 0.39 ± 0.01 |

Table 24: Flappy $\mathcal{W}$ for DQN.

| | T1 | T2 | T3 | T4 | T5 | Avg ± SEM |
|---|---|---|---|---|---|---|
| T1-C1 | 0.00 ± 0.00 | 0.00 ± 0.00 | 0.00 ± 0.00 | 0.00 ± 0.00 | 0.00 ± 0.00 | -0.98 ± 0.05 |
| T2-C1 | 0.38 ± 0.05 | 0.16 ± 0.05 | 0.23 ± 0.07 | 0.21 ± 0.07 | 0.24 ± 0.05 | -1.33 ± 0.03 |
| T3-C1 | 0.66 ± 0.04 | 0.65 ± 0.05 | 0.48 ± 0.05 | 0.31 ± 0.04 | 0.32 ± 0.03 | -0.14 ± 0.02 |
| T4-C1 | 0.17 ± 0.06 | 0.42 ± 0.08 | 0.30 ± 0.08 | 0.11 ± 0.06 | 0.10 ± 0.03 | 0.26 ± 0.03 |
| T5-C1 | -0.53 ± 0.09 | -0.13 ± 0.06 | -0.38 ± 0.18 | 0.33 ± 0.08 | 0.24 ± 0.11 | -0.01 ± 0.03 |
| T1-C2 | -0.49 ± 0.06 | -1.54 ± 0.22 | -1.59 ± 0.25 | -1.07 ± 0.24 | -0.58 ± 0.19 | – |
| T2-C2 | -0.20 ± 0.04 | -0.97 ± 0.20 | -1.55 ± 0.23 | -2.51 ± 0.14 | -2.39 ± 0.17 | – |
| T3-C2 | 0.09 ± 0.11 | 0.03 ± 0.10 | -0.29 ± 0.09 | -0.68 ± 0.14 | -0.88 ± 0.18 | – |
| T4-C2 | -0.01 ± 0.10 | 0.40 ± 0.11 | 0.28 ± 0.08 | 0.26 ± 0.08 | 0.14 ± 0.05 | – |
| T5-C2 | -0.60 ± 0.19 | -0.34 ± 0.22 | 0.15 ± 0.16 | 0.22 ± 0.12 | 0.18 ± 0.08 | – |
| T1-C3 | -0.82 ± 0.16 | -1.59 ± 0.23 | -1.77 ± 0.21 | -1.57 ± 0.21 | -0.81 ± 0.18 | – |
| T2-C3 | -0.34 ± 0.12 | -1.38 ± 0.23 | -2.07 ± 0.15 | -3.23 ± 0.38 | -2.99 ± 0.34 | – |
| T3-C3 | 0.12 ± 0.09 | -0.11 ± 0.11 | -0.36 ± 0.09 | -0.60 ± 0.17 | -0.64 ± 0.10 | – |
| T4-C3 | 0.14 ± 0.10 | 0.57 ± 0.06 | 0.46 ± 0.10 | 0.27 ± 0.08 | 0.15 ± 0.08 | – |
| T5-C3 | -0.68 ± 0.17 | -0.27 ± 0.23 | 0.28 ± 0.13 | 0.29 ± 0.09 | 0.50 ± 0.13 | – |
| T1-C4 | -1.17 ± 0.12 | -1.90 ± 0.21 | -2.22 ± 0.19 | -1.68 ± 0.22 | -0.85 ± 0.21 | – |
| T2-C4 | -0.49 ± 0.08 | -1.50 ± 0.17 | -2.27 ± 0.23 | -2.78 ± 0.19 | -3.23 ± 0.26 | – |
| T3-C4 | 0.15 ± 0.12 | -0.12 ± 0.07 | -0.55 ± 0.08 | -0.65 ± 0.12 | -0.77 ± 0.12 | – |
| T4-C4 | 0.29 ± 0.07 | 0.46 ± 0.08 | 0.49 ± 0.09 | 0.24 ± 0.06 | 0.05 ± 0.05 | – |
| T5-C4 | -0.37 ± 0.17 | -0.19 ± 0.13 | 0.30 ± 0.15 | 0.47 ± 0.10 | 0.44 ± 0.11 | – |
| Avg ± SEM | -0.18 ± 0.02 | -0.37 ± 0.03 | -0.50 ± 0.03 | -0.60 ± 0.03 | -0.54 ± 0.02 | -0.44 ± 0.01 |

Table 25: Catcher $\mathcal{F}$ for DQN.

| | T1 | T2 | T3 | T4 | T5 | Avg ± SEM |
|---|---|---|---|---|---|---|
| T1-C1 | 7.18 ± 0.23 | 4.69 ± 0.32 | 0.92 ± 0.14 | 0.36 ± 0.06 | 0.17 ± 0.01 | 0.40 ± 0.09 |
| T2-C1 | 1.25 ± 0.16 | 2.22 ± 0.19 | 1.11 ± 0.19 | 0.27 ± 0.07 | 0.08 ± 0.02 | -0.00 ± 0.07 |
| T3-C1 | -0.26 ± 0.22 | 1.99 ± 0.23 | 3.93 ± 0.13 | 2.80 ± 0.20 | 0.47 ± 0.09 | 0.86 ± 0.07 |
| T4-C1 | -3.52 ± 0.16 | -2.07 ± 0.11 | 3.10 ± 0.17 | 4.18 ± 0.08 | 1.49 ± 0.27 | 0.53 ± 0.09 |
| T5-C1 | -2.84 ± 0.18 | -3.57 ± 0.16 | -1.58 ± 0.20 | 1.66 ± 0.17 | 4.71 ± 0.08 | -0.30 ± 0.07 |
| T1-C2 | 2.68 ± 0.16 | 1.26 ± 0.19 | -2.38 ± 0.20 | -2.46 ± 0.12 | -0.07 ± 0.28 | – |
| T2-C2 | 4.38 ± 0.10 | 3.76 ± 0.14 | -0.97 ± 0.23 | -3.86 ± 0.16 | -5.02 ± 0.08 | – |
| T3-C2 | -0.47 ± 0.17 | 0.58 ± 0.13 | 2.07 ± 0.25 | 1.00 ± 0.25 | -1.07 ± 0.06 | – |
| T4-C2 | -3.23 ± 0.21 | -2.06 ± 0.22 | 2.53 ± 0.14 | 3.91 ± 0.15 | 1.36 ± 0.21 | – |
| T5-C2 | -2.97 ± 0.21 | -3.50 ± 0.14 | -2.10 ± 0.24 | 1.12 ± 0.32 | 4.64 ± 0.11 | – |
| T1-C3 | 2.24 ± 0.16 | 0.57 ± 0.27 | -2.49 ± 0.20 | -2.60 ± 0.17 | 0.13 ± 0.28 | – |
| T2-C3 | 4.35 ± 0.09 | 3.67 ± 0.15 | -0.61 ± 0.24 | -3.25 ± 0.20 | -4.88 ± 0.12 | – |
| T3-C3 | -0.57 ± 0.25 | 0.90 ± 0.30 | 2.29 ± 0.20 | 1.06 ± 0.22 | -1.22 ± 0.16 | – |
| T4-C3 | -2.87 ± 0.28 | -2.06 ± 0.38 | 2.73 ± 0.28 | 3.26 ± 0.13 | 0.88 ± 0.19 | – |
| T5-C3 | -2.16 ± 0.24 | -2.64 ± 0.28 | -1.53 ± 0.18 | 1.00 ± 0.14 | 4.29 ± 0.20 | – |
| T1-C4 | 2.01 ± 0.30 | 0.50 ± 0.35 | -2.82 ± 0.20 | -2.45 ± 0.11 | 0.54 ± 0.21 | – |
| T2-C4 | 3.67 ± 0.21 | 3.01 ± 0.24 | -1.25 ± 0.19 | -3.40 ± 0.17 | -4.59 ± 0.23 | – |
| T3-C4 | -0.40 ± 0.26 | 1.21 ± 0.30 | 2.57 ± 0.22 | 1.42 ± 0.28 | -1.17 ± 0.14 | – |
| T4-C4 | -2.75 ± 0.24 | -1.93 ± 0.23 | 3.06 ± 0.14 | 3.63 ± 0.12 | 0.88 ± 0.19 | – |
| T5-C4 | -2.07 ± 0.26 | -2.59 ± 0.17 | -1.10 ± 0.16 | 1.05 ± 0.27 | 4.27 ± 0.15 | – |
| Avg ± SEM | 0.18 ± 0.02 | 0.20 ± 0.01 | 0.37 ± 0.01 | 0.43 ± 0.01 | 0.30 ± 0.01 | 0.30 ± 0.01 |

Table 26: Catcher $\mathcal{W}$ for DQN.

| | T1 | T2 | T3 | T4 | T5 | Avg ± SEM |
|---|---|---|---|---|---|---|
| T1-C1 | 0.00 ± 0.00 | 0.00 ± 0.00 | 0.00 ± 0.00 | 0.00 ± 0.00 | 0.00 ± 0.00 | -0.92 ± 0.05 |
| T2-C1 | 0.35 ± 0.12 | 0.34 ± 0.16 | -0.00 ± 0.03 | -0.01 ± 0.01 | 0.01 ± 0.00 | -1.25 ± 0.05 |
| T3-C1 | -0.27 ± 0.22 | 0.97 ± 0.03 | 0.48 ± 0.09 | 0.12 ± 0.09 | 0.04 ± 0.09 | -0.32 ± 0.04 |
| T4-C1 | -3.52 ± 0.16 | -2.07 ± 0.11 | 0.93 ± 0.02 | 0.79 ± 0.06 | 0.15 ± 0.03 | -0.71 ± 0.06 |
| T5-C1 | -2.84 ± 0.18 | -3.57 ± 0.16 | -1.58 ± 0.20 | 0.88 ± 0.04 | 0.42 ± 0.06 | -1.24 ± 0.05 |
| T1-C2 | -0.53 ± 0.09 | -0.80 ± 0.06 | -2.38 ± 0.20 | -2.46 ± 0.12 | -0.07 ± 0.28 | – |
| T2-C2 | 0.88 ± 0.02 | 0.43 ± 0.08 | -1.38 ± 0.14 | -3.86 ± 0.16 | -5.02 ± 0.08 | – |
| T3-C2 | -0.47 ± 0.17 | 0.39 ± 0.09 | -0.04 ± 0.11 | -1.02 ± 0.11 | -1.33 ± 0.07 | – |
| T4-C2 | -3.23 ± 0.21 | -2.06 ± 0.22 | 0.88 ± 0.03 | 0.75 ± 0.09 | 0.07 ± 0.02 | – |
| T5-C2 | -2.97 ± 0.21 | -3.50 ± 0.14 | -2.10 ± 0.24 | 0.58 ± 0.12 | 0.42 ± 0.11 | – |
| T1-C3 | -0.57 ± 0.09 | -0.72 ± 0.06 | -2.49 ± 0.20 | -2.60 ± 0.17 | 0.13 ± 0.28 | – |
| T2-C3 | 0.82 ± 0.04 | 0.28 ± 0.12 | -1.06 ± 0.17 | -3.25 ± 0.20 | -4.88 ± 0.12 | – |
| T3-C3 | -0.60 ± 0.23 | 0.28 ± 0.16 | -0.07 ± 0.12 | -0.85 ± 0.09 | -1.47 ± 0.13 | – |
| T4-C3 | -2.87 ± 0.28 | -2.07 ± 0.37 | 0.78 ± 0.07 | 0.50 ± 0.13 | 0.05 ± 0.02 | – |
| T5-C3 | -2.22 ± 0.22 | -2.71 ± 0.26 | -1.54 ± 0.18 | 0.63 ± 0.10 | 0.26 ± 0.10 | – |
| T1-C4 | -0.40 ± 0.09 | -0.63 ± 0.12 | -2.82 ± 0.20 | -2.45 ± 0.11 | 0.46 ± 0.16 | – |
| T2-C4 | 0.61 ± 0.11 | 0.03 ± 0.14 | -1.32 ± 0.16 | -3.40 ± 0.17 | -4.59 ± 0.23 | – |
| T3-C4 | -0.41 ± 0.26 | 0.44 ± 0.20 | -0.19 ± 0.11 | -0.87 ± 0.18 | -1.49 ± 0.12 | – |
| T4-C4 | -2.75 ± 0.24 | -1.93 ± 0.23 | 0.86 ± 0.04 | 0.48 ± 0.10 | 0.02 ± 0.02 | – |
| T5-C4 | -2.07 ± 0.26 | -2.59 ± 0.17 | -1.10 ± 0.16 | 0.56 ± 0.11 | 0.14 ± 0.06 | – |
| Avg ± SEM | -1.15 ± 0.03 | -0.97 ± 0.03 | -0.71 ± 0.02 | -0.77 ± 0.01 | -0.84 ± 0.02 | -0.89 ± 0.01 |

Table 27: Room $\mathcal{F}$ for DDQN.

| | T1 | T2 | T3 | T4 | T5 | Avg ± SEM |
|---|---|---|---|---|---|---|
| T1-C1 | 6.94 ± 0.39 | 3.27 ± 0.50 | 6.98 ± 0.29 | 6.98 ± 0.36 | 2.90 ± 0.33 | 1.57 ± 0.06 |
| T2-C1 | 1.00 ± 0.11 | 1.80 ± 0.31 | 1.03 ± 0.13 | 0.96 ± 0.10 | 2.05 ± 0.19 | 0.29 ± 0.05 |
| T3-C1 | 1.05 ± 0.12 | 1.69 ± 0.19 | 1.17 ± 0.14 | 1.05 ± 0.08 | 1.88 ± 0.24 | 0.25 ± 0.04 |
| T4-C1 | 0.46 ± 0.07 | -0.92 ± 0.16 | 0.45 ± 0.06 | 0.41 ± 0.07 | -0.88 ± 0.26 | -0.06 ± 0.05 |
| T5-C1 | -0.20 ± 0.13 | 0.77 ± 0.29 | -0.27 ± 0.10 | -0.39 ± 0.09 | 0.71 ± 0.15 | 0.11 ± 0.04 |
| T1-C2 | -0.24 ± 0.06 | 1.23 ± 0.26 | -0.26 ± 0.06 | -0.26 ± 0.03 | 0.97 ± 0.19 | – |
| T2-C2 | -0.23 ± 0.11 | 0.14 ± 0.21 | -0.24 ± 0.11 | -0.24 ± 0.11 | 0.16 ± 0.21 | – |
| T3-C2 | -0.08 ± 0.12 | -0.12 ± 0.27 | -0.12 ± 0.07 | -0.16 ± 0.10 | 0.38 ± 0.16 | – |
| T4-C2 | 0.80 ± 0.13 | -1.30 ± 0.21 | 0.78 ± 0.09 | 0.91 ± 0.15 | -1.44 ± 0.19 | – |
| T5-C2 | -0.24 ± 0.13 | 0.58 ± 0.18 | -0.08 ± 0.09 | -0.17 ± 0.10 | 0.32 ± 0.15 | – |
| T1-C3 | -0.13 ± 0.03 | 0.84 ± 0.20 | -0.19 ± 0.07 | -0.20 ± 0.05 | 0.97 ± 0.15 | – |
| T2-C3 | -0.20 ± 0.11 | 0.11 ± 0.16 | -0.30 ± 0.10 | -0.23 ± 0.16 | -0.00 ± 0.14 | – |
| T3-C3 | -0.13 ± 0.06 | 0.22 ± 0.25 | -0.06 ± 0.10 | -0.16 ± 0.07 | -0.02 ± 0.22 | – |
| T4-C3 | 0.75 ± 0.08 | -1.02 ± 0.16 | 0.76 ± 0.08 | 0.82 ± 0.10 | -1.17 ± 0.18 | – |
| T5-C3 | -0.36 ± 0.10 | 0.31 ± 0.23 | -0.37 ± 0.14 | -0.26 ± 0.14 | 0.71 ± 0.19 | – |
| T1-C4 | -0.18 ± 0.04 | 0.94 ± 0.28 | -0.16 ± 0.02 | -0.18 ± 0.04 | 1.28 ± 0.21 | – |
| T2-C4 | -0.18 ± 0.12 | 0.40 ± 0.17 | -0.02 ± 0.11 | -0.11 ± 0.18 | -0.00 ± 0.14 | – |
| T3-C4 | -0.14 ± 0.06 | -0.24 ± 0.23 | -0.29 ± 0.05 | -0.20 ± 0.06 | -0.76 ± 0.26 | – |
| T4-C4 | 0.81 ± 0.12 | -1.71 ± 0.25 | 0.67 ± 0.09 | 0.79 ± 0.14 | -1.19 ± 0.19 | – |
| T5-C4 | -0.09 ± 0.07 | 0.66 ± 0.17 | -0.12 ± 0.09 | -0.07 ± 0.10 | 0.81 ± 0.16 | – |
| Avg ± SEM | 0.47 ± 0.01 | 0.38 ± 0.02 | 0.47 ± 0.02 | 0.46 ± 0.01 | 0.38 ± 0.02 | 0.43 ± 0.01 |

Table 28

Table 29: Room $\mathcal{W}$ for DDQN.

| | T1 | T2 | T3 | T4 | T5 | Avg ± SEM |
|---|---|---|---|---|---|---|
| T1-C1 | 0.00 ± 0.00 | 0.00 ± 0.00 | 0.00 ± 0.00 | 0.00 ± 0.00 | -0.06 ± 0.06 | -0.01 ± 0.02 |
| T2-C1 | 0.42 ± 0.07 | -0.01 ± 0.14 | 0.45 ± 0.05 | 0.44 ± 0.04 | 0.08 ± 0.12 | -0.11 ± 0.03 |
| T3-C1 | 0.69 ± 0.07 | 0.81 ± 0.08 | 0.80 ± 0.08 | 0.72 ± 0.07 | 0.85 ± 0.08 | -0.04 ± 0.02 |
| T4-C1 | -0.00 ± 0.00 | -0.92 ± 0.16 | 0.02 ± 0.01 | -0.01 ± 0.01 | -0.88 ± 0.26 | -0.49 ± 0.03 |
| T5-C1 | -0.23 ± 0.12 | -0.60 ± 0.10 | -0.27 ± 0.10 | -0.39 ± 0.09 | -0.38 ± 0.08 | -0.37 ± 0.03 |
| T1-C2 | -0.24 ± 0.06 | 0.36 ± 0.11 | -0.27 ± 0.05 | -0.26 ± 0.03 | 0.26 ± 0.09 | – |
| T2-C2 | -0.32 ± 0.08 | -0.26 ± 0.15 | -0.34 ± 0.07 | -0.29 ± 0.10 | -0.24 ± 0.11 | – |
| T3-C2 | -0.09 ± 0.11 | -0.45 ± 0.16 | -0.18 ± 0.06 | -0.22 ± 0.09 | -0.15 ± 0.10 | – |
| T4-C2 | -0.02 ± 0.01 | -1.30 ± 0.21 | -0.02 ± 0.03 | -0.04 ± 0.03 | -1.44 ± 0.19 | – |
| T5-C2 | -0.24 ± 0.13 | -0.83 ± 0.15 | -0.08 ± 0.09 | -0.17 ± 0.10 | -0.60 ± 0.05 | – |
| T1-C3 | -0.14 ± 0.03 | 0.45 ± 0.09 | -0.21 ± 0.07 | -0.21 ± 0.05 | 0.29 ± 0.09 | – |
| T2-C3 | -0.27 ± 0.09 | -0.12 ± 0.16 | -0.36 ± 0.09 | -0.38 ± 0.11 | -0.16 ± 0.12 | – |
| T3-C3 | -0.14 ± 0.06 | -0.39 ± 0.16 | -0.10 ± 0.08 | -0.17 ± 0.08 | -0.55 ± 0.19 | – |
| T4-C3 | -0.01 ± 0.01 | -1.03 ± 0.16 | 0.01 ± 0.01 | -0.01 ± 0.01 | -1.17 ± 0.18 | – |
| T5-C3 | -0.36 ± 0.10 | -0.83 ± 0.11 | -0.38 ± 0.14 | -0.26 ± 0.14 | -0.40 ± 0.08 | – |
| T1-C4 | -0.18 ± 0.04 | 0.12 ± 0.14 | -0.17 ± 0.03 | -0.18 ± 0.04 | 0.25 ± 0.08 | – |
| T2-C4 | -0.26 ± 0.10 | -0.02 ± 0.08 | -0.14 ± 0.07 | -0.32 ± 0.08 | -0.18 ± 0.12 | – |
| T3-C4 | -0.14 ± 0.06 | -0.64 ± 0.13 | -0.32 ± 0.05 | -0.20 ± 0.06 | -0.97 ± 0.18 | – |
| T4-C4 | -0.01 ± 0.01 | -1.71 ± 0.25 | -0.00 ± 0.00 | -0.00 ± 0.00 | -1.19 ± 0.19 | – |
| T5-C4 | -0.09 ± 0.07 | -0.57 ± 0.10 | -0.12 ± 0.09 | -0.07 ± 0.10 | -0.54 ± 0.03 | – |
| Avg ± SEM | -0.08 ± 0.01 | -0.40 ± 0.02 | -0.08 ± 0.01 | -0.10 ± 0.01 | -0.36 ± 0.01 | -0.20 ± 0.01 |

Table 30

Table 31: Flappy $\mathcal{F}$ for DDQN.

| | T1 | T2 | T3 | T4 | T5 | Avg ± SEM |
|---|---|---|---|---|---|---|
| T1-C1 | 2.92 ± 0.17 | 2.01 ± 0.10 | 1.66 ± 0.16 | 1.75 ± 0.14 | 2.00 ± 0.16 | -0.47 ± 0.05 |
| T2-C1 | 2.05 ± 0.14 | 1.74 ± 0.14 | 1.35 ± 0.13 | 0.92 ± 0.12 | 0.94 ± 0.07 | -1.02 ± 0.04 |
| T3-C1 | 2.25 ± 0.13 | 2.38 ± 0.15 | 2.36 ± 0.16 | 2.04 ± 0.13 | 1.39 ± 0.15 | 1.05 ± 0.06 |
| T4-C1 | 0.18 ± 0.20 | 0.75 ± 0.19 | 1.89 ± 0.18 | 2.10 ± 0.22 | 1.48 ± 0.23 | 1.39 ± 0.06 |
| T5-C1 | -0.47 ± 0.10 | 0.11 ± 0.13 | 0.87 ± 0.19 | 1.26 ± 0.20 | 2.48 ± 0.43 | 1.05 ± 0.05 |
| T1-C2 | -0.66 ± 0.12 | -1.09 ± 0.17 | -2.03 ± 0.27 | -1.97 ± 0.20 | -0.88 ± 0.26 | – |
| T2-C2 | 0.33 ± 0.18 | -0.77 ± 0.16 | -1.99 ± 0.16 | -2.34 ± 0.25 | -2.99 ± 0.39 | – |
| T3-C2 | 1.17 ± 0.14 | 1.22 ± 0.16 | 0.91 ± 0.29 | 0.55 ± 0.22 | -0.00 ± 0.21 | – |
| T4-C2 | 0.37 ± 0.13 | 1.15 ± 0.14 | 1.77 ± 0.29 | 1.78 ± 0.25 | 1.50 ± 0.23 | – |
| T5-C2 | -0.88 ± 0.12 | -0.27 ± 0.13 | 1.24 ± 0.14 | 2.11 ± 0.14 | 2.73 ± 0.36 | – |
| T1-C3 | -0.72 ± 0.15 | -1.67 ± 0.16 | -1.61 ± 0.31 | -1.33 ± 0.17 | -1.15 ± 0.26 | – |
| T2-C3 | 0.50 ± 0.11 | -0.59 ± 0.14 | -2.27 ± 0.18 | -3.05 ± 0.20 | -3.54 ± 0.37 | – |
| T3-C3 | 1.04 ± 0.22 | 1.59 ± 0.10 | 0.67 ± 0.17 | -0.25 ± 0.23 | -0.26 ± 0.13 | – |
| T4-C3 | 0.10 ± 0.12 | 1.08 ± 0.16 | 1.89 ± 0.24 | 2.01 ± 0.21 | 1.46 ± 0.20 | – |
| T5-C3 | -0.61 ± 0.19 | -0.30 ± 0.13 | 1.59 ± 0.27 | 3.19 ± 0.30 | 2.78 ± 0.10 | – |
| T1-C4 | -1.03 ± 0.17 | -1.75 ± 0.21 | -1.52 ± 0.38 | -1.45 ± 0.18 | -0.91 ± 0.25 | – |
| T2-C4 | 0.15 ± 0.23 | -1.02 ± 0.21 | -2.86 ± 0.24 | -3.91 ± 0.26 | -3.15 ± 0.18 | – |
| T3-C4 | 1.46 ± 0.20 | 1.56 ± 0.16 | 0.71 ± 0.27 | 0.24 ± 0.22 | 0.03 ± 0.29 | – |
| T4-C4 | 0.39 ± 0.22 | 1.64 ± 0.21 | 2.28 ± 0.28 | 2.32 ± 0.22 | 1.77 ± 0.19 | – |
| T5-C4 | -0.77 ± 0.19 | -0.08 ± 0.14 | 0.93 ± 0.23 | 2.56 ± 0.42 | 2.53 ± 0.28 | – |
| Avg ± SEM | 0.39 ± 0.01 | 0.38 ± 0.01 | 0.39 ± 0.02 | 0.43 ± 0.02 | 0.41 ± 0.01 | 0.40 ± 0.01 |

Table 32: Flappy $\mathcal{W}$ for DDQN.

| | T1 | T2 | T3 | T4 | T5 | Avg ± SEM |
|---|---|---|---|---|---|---|
| T1-C1 | 0.00 ± 0.00 | 0.00 ± 0.00 | 0.00 ± 0.00 | 0.00 ± 0.00 | 0.00 ± 0.00 | -1.00 ± 0.05 |
| T2-C1 | 0.38 ± 0.06 | 0.23 ± 0.09 | 0.10 ± 0.08 | 0.17 ± 0.06 | 0.20 ± 0.07 | -1.44 ± 0.04 |
| T3-C1 | 0.72 ± 0.03 | 0.62 ± 0.06 | 0.51 ± 0.08 | 0.46 ± 0.05 | 0.49 ± 0.04 | -0.19 ± 0.03 |
| T4-C1 | -0.04 ± 0.12 | 0.24 ± 0.08 | 0.51 ± 0.08 | 0.34 ± 0.09 | 0.20 ± 0.05 | 0.24 ± 0.02 |
| T5-C1 | -0.52 ± 0.09 | -0.09 ± 0.11 | 0.16 ± 0.13 | 0.26 ± 0.12 | 0.23 ± 0.13 | 0.01 ± 0.03 |
| T1-C2 | -0.69 ± 0.12 | -1.11 ± 0.16 | -2.03 ± 0.27 | -1.97 ± 0.20 | -0.89 ± 0.25 | – |
| T2-C2 | -0.26 ± 0.08 | -0.87 ± 0.14 | -2.01 ± 0.15 | -2.34 ± 0.25 | -2.99 ± 0.39 | – |
| T3-C2 | 0.22 ± 0.10 | -0.15 ± 0.08 | -0.42 ± 0.16 | -0.41 ± 0.14 | -0.80 ± 0.14 | – |
| T4-C2 | 0.15 ± 0.07 | 0.27 ± 0.06 | 0.34 ± 0.10 | 0.17 ± 0.15 | 0.21 ± 0.09 | – |
| T5-C2 | -0.89 ± 0.12 | -0.37 ± 0.12 | 0.38 ± 0.11 | 0.45 ± 0.10 | 0.41 ± 0.18 | – |
| T1-C3 | -0.79 ± 0.14 | -1.68 ± 0.16 | -1.62 ± 0.31 | -1.34 ± 0.17 | -1.19 ± 0.25 | – |
| T2-C3 | -0.19 ± 0.07 | -0.71 ± 0.10 | -2.27 ± 0.18 | -3.05 ± 0.20 | -3.54 ± 0.37 | – |
| T3-C3 | 0.03 ± 0.08 | -0.13 ± 0.05 | -0.69 ± 0.08 | -0.94 ± 0.12 | -0.77 ± 0.14 | – |
| T4-C3 | -0.11 ± 0.07 | 0.19 ± 0.10 | 0.41 ± 0.09 | 0.26 ± 0.07 | 0.09 ± 0.10 | – |
| T5-C3 | -0.67 ± 0.16 | -0.36 ± 0.10 | 0.35 ± 0.08 | 0.61 ± 0.08 | 0.35 ± 0.10 | – |
| T1-C4 | -1.03 ± 0.17 | -1.75 ± 0.21 | -1.53 ± 0.38 | -1.45 ± 0.18 | -0.92 ± 0.25 | – |
| T2-C4 | -0.45 ± 0.10 | -1.21 ± 0.15 | -2.86 ± 0.24 | -3.91 ± 0.26 | -3.15 ± 0.18 | – |
| T3-C4 | 0.07 ± 0.11 | -0.30 ± 0.03 | -0.78 ± 0.14 | -0.84 ± 0.09 | -0.73 ± 0.22 | – |
| T4-C4 | 0.13 ± 0.16 | 0.50 ± 0.07 | 0.48 ± 0.10 | 0.33 ± 0.09 | 0.20 ± 0.06 | – |
| T5-C4 | -0.85 ± 0.17 | -0.16 ± 0.11 | 0.32 ± 0.13 | 0.56 ± 0.17 | 0.09 ± 0.05 | – |
| Avg ± SEM | -0.24 ± 0.01 | -0.34 ± 0.02 | -0.53 ± 0.04 | -0.63 ± 0.02 | -0.63 ± 0.04 | -0.47 ± 0.02 |

Table 33: Catcher $\mathcal{F}$ for DDQN.

| | T1 | T2 | T3 | T4 | T5 | Avg ± SEM |
|---|---|---|---|---|---|---|
| T1-C1 | 6.70 ± 0.28 | 3.63 ± 0.46 | 0.55 ± 0.08 | 0.24 ± 0.03 | 0.16 ± 0.02 | 0.17 ± 0.09 |
| T2-C1 | 1.43 ± 0.18 | 2.59 ± 0.22 | 0.96 ± 0.19 | 0.20 ± 0.04 | 0.07 ± 0.01 | -0.25 ± 0.05 |
| T3-C1 | -0.14 ± 0.25 | 2.50 ± 0.33 | 3.92 ± 0.09 | 2.47 ± 0.16 | 0.41 ± 0.04 | 0.82 ± 0.06 |
| T4-C1 | -3.19 ± 0.25 | -1.80 ± 0.24 | 3.45 ± 0.20 | 4.07 ± 0.10 | 1.14 ± 0.15 | 0.85 ± 0.06 |
| T5-C1 | -2.70 ± 0.23 | -3.58 ± 0.16 | -1.81 ± 0.17 | 1.72 ± 0.24 | 5.35 ± 0.09 | 0.06 ± 0.08 |
| T1-C2 | 2.34 ± 0.31 | 0.67 ± 0.25 | -2.67 ± 0.11 | -2.27 ± 0.17 | 0.56 ± 0.20 | – |
| T2-C2 | 4.26 ± 0.11 | 3.82 ± 0.12 | -1.31 ± 0.16 | -3.93 ± 0.10 | -5.68 ± 0.09 | – |
| T3-C2 | -0.96 ± 0.15 | 0.53 ± 0.16 | 1.74 ± 0.18 | -0.21 ± 0.22 | -1.50 ± 0.09 | – |
| T4-C2 | -3.13 ± 0.21 | -2.24 ± 0.23 | 3.38 ± 0.23 | 4.16 ± 0.14 | 1.03 ± 0.28 | – |
| T5-C2 | -2.32 ± 0.16 | -2.89 ± 0.10 | -1.57 ± 0.14 | 2.17 ± 0.16 | 5.20 ± 0.13 | – |
| T1-C3 | 2.32 ± 0.19 | 0.30 ± 0.23 | -3.00 ± 0.16 | -2.85 ± 0.17 | 0.67 ± 0.36 | – |
| T2-C3 | 4.19 ± 0.11 | 3.18 ± 0.12 | -1.49 ± 0.20 | -3.81 ± 0.20 | -5.41 ± 0.16 | – |
| T3-C3 | -0.17 ± 0.20 | 1.71 ± 0.23 | 2.54 ± 0.08 | 0.51 ± 0.20 | -1.48 ± 0.16 | – |
| T4-C3 | -2.69 ± 0.28 | -1.42 ± 0.26 | 3.91 ± 0.14 | 4.15 ± 0.12 | 0.99 ± 0.16 | – |
| T5-C3 | -2.68 ± 0.23 | -3.13 ± 0.22 | -1.40 ± 0.24 | 2.00 ± 0.28 | 5.33 ± 0.12 | – |
| T1-C4 | 1.98 ± 0.28 | -0.48 ± 0.31 | -3.40 ± 0.19 | -2.81 ± 0.20 | 0.74 ± 0.20 | – |
| T2-C4 | 3.79 ± 0.25 | 3.05 ± 0.16 | -1.68 ± 0.27 | -3.77 ± 0.13 | -5.56 ± 0.06 | – |
| T3-C4 | 0.14 ± 0.21 | 2.38 ± 0.14 | 2.81 ± 0.13 | 0.83 ± 0.30 | -1.59 ± 0.09 | – |
| T4-C4 | -2.10 ± 0.30 | -1.27 ± 0.25 | 3.65 ± 0.24 | 4.02 ± 0.09 | 0.85 ± 0.17 | – |
| T5-C4 | -2.43 ± 0.24 | -2.63 ± 0.19 | -0.69 ± 0.23 | 1.93 ± 0.35 | 5.40 ± 0.16 | – |
| Avg ± SEM | 0.23 ± 0.02 | 0.25 ± 0.02 | 0.39 ± 0.01 | 0.44 ± 0.01 | 0.33 ± 0.01 | 0.33 ± 0.01 |

Table 34: Catcher $\mathcal{W}$ for DDQN.

| | T1 | T2 | T3 | T4 | T5 | Avg ± SEM |
|---|---|---|---|---|---|---|
| T1-C1 | 0.00 ± 0.00 | 0.00 ± 0.00 | 0.00 ± 0.00 | 0.00 ± 0.00 | 0.00 ± 0.00 | -0.93 ± 0.05 |
| T2-C1 | 0.48 ± 0.06 | 0.39 ± 0.16 | 0.00 ± 0.02 | 0.00 ± 0.01 | -0.01 ± 0.01 | -1.47 ± 0.03 |
| T3-C1 | -0.14 ± 0.25 | 0.87 ± 0.09 | 0.30 ± 0.01 | 0.08 ± 0.02 | 0.03 ± 0.00 | -0.38 ± 0.04 |
| T4-C1 | -3.19 ± 0.25 | -1.80 ± 0.24 | 0.93 ± 0.01 | 0.58 ± 0.07 | 0.10 ± 0.02 | -0.58 ± 0.04 |
| T5-C1 | -2.70 ± 0.23 | -3.58 ± 0.16 | -1.81 ± 0.17 | 0.71 ± 0.10 | 0.25 ± 0.03 | -1.21 ± 0.07 |
| T1-C2 | -0.52 ± 0.09 | -0.73 ± 0.07 | -2.67 ± 0.11 | -2.27 ± 0.17 | 0.54 ± 0.19 | – |
| T2-C2 | 0.82 ± 0.06 | 0.41 ± 0.10 | -1.42 ± 0.15 | -3.93 ± 0.10 | -5.68 ± 0.09 | – |
| T3-C2 | -0.97 ± 0.15 | 0.25 ± 0.12 | -0.36 ± 0.15 | -1.46 ± 0.14 | -1.63 ± 0.09 | – |
| T4-C2 | -3.13 ± 0.21 | -2.25 ± 0.23 | 0.78 ± 0.08 | 0.58 ± 0.13 | 0.19 ± 0.09 | – |
| T5-C2 | -2.34 ± 0.15 | -2.89 ± 0.10 | -1.57 ± 0.14 | 0.81 ± 0.07 | 0.14 ± 0.03 | – |
| T1-C3 | -0.29 ± 0.13 | -0.55 ± 0.10 | -3.00 ± 0.16 | -2.85 ± 0.17 | 0.50 ± 0.29 | – |
| T2-C3 | 0.84 ± 0.03 | -0.08 ± 0.04 | -1.52 ± 0.21 | -3.81 ± 0.20 | -5.41 ± 0.16 | – |
| T3-C3 | -0.22 ± 0.19 | 0.57 ± 0.10 | -0.25 ± 0.09 | -0.95 ± 0.13 | -1.69 ± 0.15 | – |
| T4-C3 | -2.70 ± 0.28 | -1.42 ± 0.26 | 0.96 ± 0.01 | 0.55 ± 0.08 | 0.06 ± 0.01 | – |
| T5-C3 | -2.68 ± 0.23 | -3.13 ± 0.22 | -1.40 ± 0.24 | 0.80 ± 0.06 | 0.12 ± 0.03 | – |
| T1-C4 | -0.41 ± 0.11 | -0.87 ± 0.22 | -3.40 ± 0.19 | -2.81 ± 0.20 | 0.69 ± 0.17 | – |
| T2-C4 | 0.59 ± 0.11 | -0.09 ± 0.09 | -1.72 ± 0.26 | -3.77 ± 0.13 | -5.56 ± 0.06 | – |
| T3-C4 | 0.12 ± 0.21 | 0.74 ± 0.07 | -0.27 ± 0.08 | -0.87 ± 0.10 | -1.69 ± 0.09 | – |
| T4-C4 | -2.10 ± 0.30 | -1.27 ± 0.25 | 0.91 ± 0.04 | 0.50 ± 0.08 | 0.12 ± 0.07 | – |
| T5-C4 | -2.43 ± 0.24 | -2.65 ± 0.18 | -0.69 ± 0.23 | 0.77 ± 0.08 | 0.17 ± 0.06 | – |
| Avg ± SEM | -1.05 ± 0.02 | -0.90 ± 0.02 | -0.81 ± 0.02 | -0.87 ± 0.01 | -0.94 ± 0.02 | -0.91 ± 0.01 |

Table 35: Room $\mathcal{F}$ for MER.

|  | T1 | T2 | T3 | T4 | T5 | Avg ± SEM |
|---|---|---|---|---|---|---|
| T1-C1 | 3.75 ± 0.48 | 2.07 ± 0.34 | 4.35 ± 0.52 | 3.80 ± 0.53 | 1.84 ± 0.35 | 0.92 ± 0.14 |
| T2-C1 | 1.90 ± 0.26 | 1.82 ± 0.21 | 2.03 ± 0.26 | 1.97 ± 0.26 | 1.88 ± 0.23 | 0.45 ± 0.06 |
| T3-C1 | 1.66 ± 0.43 | 1.83 ± 0.23 | 1.61 ± 0.44 | 1.62 ± 0.13 | 1.87 ± 0.31 | 0.25 ± 0.12 |
| T4-C1 | 1.28 ± 0.29 | -0.49 ± 0.25 | 1.13 ± 0.30 | 1.23 ± 0.26 | -0.21 ± 0.33 | 0.19 ± 0.10 |
| T5-C1 | 1.01 ± 0.28 | 0.36 ± 0.20 | 1.14 ± 0.29 | 1.04 ± 0.29 | 0.35 ± 0.29 | 0.44 ± 0.11 |
| T1-C2 | -0.81 ± 0.24 | 1.14 ± 0.33 | -0.71 ± 0.25 | -0.65 ± 0.23 | 0.93 ± 0.24 | – |
| T2-C2 | -0.02 ± 0.15 | 0.19 ± 0.14 | -0.17 ± 0.15 | -0.10 ± 0.13 | 0.22 ± 0.26 | – |
| T3-C2 | -0.18 ± 0.25 | -0.55 ± 0.19 | -0.17 ± 0.27 | -0.45 ± 0.26 | -0.58 ± 0.13 | – |
| T4-C2 | 0.51 ± 0.16 | -0.62 ± 0.31 | 0.62 ± 0.11 | 0.61 ± 0.14 | -0.90 ± 0.26 | – |
| T5-C2 | 0.58 ± 0.23 | 0.98 ± 0.17 | 0.44 ± 0.28 | 0.70 ± 0.33 | 0.81 ± 0.24 | – |
| T1-C3 | -0.22 ± 0.08 | 0.84 ± 0.16 | -0.26 ± 0.07 | -0.30 ± 0.10 | 1.18 ± 0.24 | – |
| T2-C3 | -0.36 ± 0.07 | 0.22 ± 0.14 | -0.31 ± 0.12 | -0.44 ± 0.15 | 0.13 ± 0.19 | – |
| T3-C3 | -0.59 ± 0.17 | -0.07 ± 0.16 | -0.42 ± 0.09 | -0.26 ± 0.13 | -0.20 ± 0.15 | – |
| T4-C3 | 0.75 ± 0.07 | -1.34 ± 0.33 | 0.77 ± 0.13 | 0.83 ± 0.13 | -1.16 ± 0.25 | – |
| T5-C3 | 0.04 ± 0.19 | 0.61 ± 0.18 | 0.03 ± 0.13 | -0.12 ± 0.19 | 0.71 ± 0.11 | – |
| T1-C4 | -0.48 ± 0.07 | 1.11 ± 0.39 | -0.26 ± 0.09 | -0.40 ± 0.10 | 1.38 ± 0.31 | – |
| T2-C4 | 0.03 ± 0.17 | 0.13 ± 0.18 | -0.14 ± 0.15 | -0.11 ± 0.22 | 0.19 ± 0.10 | – |
| T3-C4 | -0.16 ± 0.14 | 0.36 ± 0.23 | -0.31 ± 0.15 | -0.11 ± 0.17 | -0.00 ± 0.20 | – |
| T4-C4 | 0.75 ± 0.14 | -0.95 ± 0.26 | 0.80 ± 0.13 | 0.82 ± 0.16 | -0.71 ± 0.20 | – |
| T5-C4 | -0.19 ± 0.16 | 0.17 ± 0.26 | -0.10 ± 0.13 | -0.12 ± 0.21 | 0.45 ± 0.23 | – |
| Avg ± SEM | 0.46 ± 0.02 | 0.39 ± 0.02 | 0.50 ± 0.01 | 0.48 ± 0.01 | 0.41 ± 0.02 | 0.45 ± 0.01 |

Table 36: Room $\mathcal{W}$ for MER.

|  | T1 | T2 | T3 | T4 | T5 | Avg ± SEM |
|---|---|---|---|---|---|---|
| T1-C1 | 0.00 ± 0.00 | -0.06 ± 0.06 | 0.00 ± 0.00 | 0.00 ± 0.00 | -0.09 ± 0.09 | -0.12 ± 0.05 |
| T2-C1 | 0.68 ± 0.17 | 0.35 ± 0.12 | 0.77 ± 0.16 | 0.75 ± 0.15 | 0.28 ± 0.14 | -0.05 ± 0.03 |
| T3-C1 | 0.31 ± 0.21 | 0.76 ± 0.12 | 0.28 ± 0.19 | 0.33 ± 0.17 | 0.75 ± 0.14 | -0.20 ± 0.07 |
| T4-C1 | 0.21 ± 0.12 | -0.57 ± 0.23 | 0.16 ± 0.09 | 0.17 ± 0.10 | -0.41 ± 0.25 | -0.32 ± 0.05 |
| T5-C1 | 0.02 ± 0.07 | -0.66 ± 0.07 | 0.12 ± 0.06 | 0.14 ± 0.06 | -0.86 ± 0.13 | -0.29 ± 0.03 |
| T1-C2 | -0.81 ± 0.24 | 0.32 ± 0.15 | -0.73 ± 0.24 | -0.65 ± 0.23 | 0.31 ± 0.12 | – |
| T2-C2 | -0.20 ± 0.08 | -0.22 ± 0.10 | -0.20 ± 0.05 | -0.28 ± 0.09 | -0.25 ± 0.16 | – |
| T3-C2 | -0.39 ± 0.20 | -0.67 ± 0.15 | -0.45 ± 0.17 | -0.63 ± 0.19 | -0.74 ± 0.09 | – |
| T4-C2 | -0.01 ± 0.01 | -0.72 ± 0.26 | 0.01 ± 0.01 | -0.02 ± 0.01 | -0.93 ± 0.24 | – |
| T5-C2 | 0.16 ± 0.09 | -0.64 ± 0.09 | 0.02 ± 0.10 | 0.09 ± 0.10 | -0.65 ± 0.12 | – |
| T1-C3 | -0.22 ± 0.08 | 0.37 ± 0.09 | -0.27 ± 0.07 | -0.31 ± 0.10 | 0.45 ± 0.11 | – |
| T2-C3 | -0.40 ± 0.05 | -0.06 ± 0.07 | -0.39 ± 0.10 | -0.49 ± 0.13 | -0.19 ± 0.14 | – |
| T3-C3 | -0.60 ± 0.17 | -0.31 ± 0.11 | -0.43 ± 0.09 | -0.31 ± 0.09 | -0.40 ± 0.14 | – |
| T4-C3 | -0.02 ± 0.01 | -1.34 ± 0.33 | -0.03 ± 0.02 | -0.02 ± 0.02 | -1.16 ± 0.25 | – |
| T5-C3 | -0.09 ± 0.13 | -0.68 ± 0.11 | -0.03 ± 0.12 | -0.21 ± 0.14 | -0.69 ± 0.07 | – |
| T1-C4 | -0.48 ± 0.07 | 0.17 ± 0.12 | -0.26 ± 0.09 | -0.40 ± 0.10 | 0.33 ± 0.14 | – |
| T2-C4 | -0.26 ± 0.09 | -0.16 ± 0.15 | -0.37 ± 0.14 | -0.36 ± 0.14 | 0.02 ± 0.13 | – |
| T3-C4 | -0.24 ± 0.13 | -0.23 ± 0.11 | -0.36 ± 0.14 | -0.22 ± 0.14 | -0.39 ± 0.13 | – |
| T4-C4 | 0.00 ± 0.00 | -0.95 ± 0.26 | -0.00 ± 0.00 | -0.00 ± 0.00 | -0.73 ± 0.19 | – |
| T5-C4 | -0.20 ± 0.16 | -0.79 ± 0.12 | -0.11 ± 0.13 | -0.21 ± 0.14 | -0.55 ± 0.08 | – |
| Avg ± SEM | -0.13 ± 0.03 | -0.30 ± 0.02 | -0.11 ± 0.03 | -0.13 ± 0.03 | -0.30 ± 0.02 | -0.19 ± 0.02 |

Table 37: Flappy $\mathcal{F}$ for MER.

|  | T1 | T2 | T3 | T4 | T5 | Avg ± SEM |
|---|---|---|---|---|---|---|
| T1-C1 | 0.55 ± 0.09 | 0.37 ± 0.05 | 0.32 ± 0.04 | 0.31 ± 0.02 | 0.53 ± 0.05 | -0.45 ± 0.06 |
| T2-C1 | 2.24 ± 0.11 | 1.48 ± 0.11 | 1.04 ± 0.09 | 0.98 ± 0.10 | 0.90 ± 0.09 | -0.50 ± 0.06 |
| T3-C1 | 3.05 ± 0.12 | 2.73 ± 0.13 | 2.13 ± 0.14 | 1.75 ± 0.05 | 1.55 ± 0.07 | 1.02 ± 0.08 |
| T4-C1 | 1.03 ± 0.18 | 1.85 ± 0.23 | 1.70 ± 0.19 | 1.51 ± 0.14 | 1.57 ± 0.18 | 1.24 ± 0.04 |
| T5-C1 | 0.12 ± 0.15 | 0.76 ± 0.19 | 0.92 ± 0.14 | 1.50 ± 0.29 | 2.11 ± 0.22 | 0.67 ± 0.07 |
| T1-C2 | -0.24 ± 0.21 | -0.29 ± 0.20 | -0.69 ± 0.21 | -0.71 ± 0.23 | -0.54 ± 0.25 | – |
| T2-C2 | 0.36 ± 0.23 | -0.31 ± 0.20 | -1.11 ± 0.23 | -1.59 ± 0.34 | -2.21 ± 0.22 | – |
| T3-C2 | 0.79 ± 0.23 | 0.73 ± 0.19 | 1.03 ± 0.18 | 0.63 ± 0.10 | 0.22 ± 0.15 | – |
| T4-C2 | -0.22 ± 0.16 | 0.76 ± 0.25 | 1.71 ± 0.22 | 1.79 ± 0.31 | 1.49 ± 0.20 | – |
| T5-C2 | -0.68 ± 0.23 | -0.20 ± 0.17 | 0.40 ± 0.26 | 1.11 ± 0.26 | 1.83 ± 0.30 | – |
| T1-C3 | 0.44 ± 0.20 | -0.78 ± 0.19 | -1.24 ± 0.33 | -1.55 ± 0.32 | -0.83 ± 0.34 | – |
| T2-C3 | 0.79 ± 0.19 | -0.02 ± 0.15 | -1.30 ± 0.25 | -2.00 ± 0.17 | -2.75 ± 0.30 | – |
| T3-C3 | 0.19 ± 0.18 | 0.75 ± 0.19 | 0.82 ± 0.43 | 0.50 ± 0.20 | -0.21 ± 0.25 | – |
| T4-C3 | -0.44 ± 0.14 | 0.54 ± 0.21 | 1.48 ± 0.22 | 1.88 ± 0.13 | 1.40 ± 0.15 | – |
| T5-C3 | -0.95 ± 0.18 | -0.14 ± 0.19 | 0.65 ± 0.26 | 1.32 ± 0.28 | 2.18 ± 0.27 | – |
| T1-C4 | 0.22 ± 0.20 | -1.14 ± 0.23 | -1.62 ± 0.21 | -1.46 ± 0.20 | -0.67 ± 0.23 | – |
| T2-C4 | 1.08 ± 0.15 | -0.44 ± 0.18 | -1.93 ± 0.24 | -2.34 ± 0.22 | -2.88 ± 0.22 | – |
| T3-C4 | 0.53 ± 0.30 | 1.37 ± 0.21 | 1.32 ± 0.23 | 0.50 ± 0.18 | 0.08 ± 0.29 | – |
| T4-C4 | -0.44 ± 0.15 | 0.84 ± 0.17 | 2.18 ± 0.22 | 2.44 ± 0.24 | 1.82 ± 0.11 | – |
| T5-C4 | -0.99 ± 0.15 | -0.71 ± 0.28 | 0.50 ± 0.25 | 1.52 ± 0.22 | 2.18 ± 0.36 | – |
| Avg ± SEM | 0.37 ± 0.01 | 0.41 ± 0.01 | 0.42 ± 0.01 | 0.40 ± 0.02 | 0.39 ± 0.02 | 0.40 ± 0.01 |

Table 38: Flappy $\mathcal{W}$ for MER.

| | T1 | T2 | T3 | T4 | T5 | Avg ± SEM |
|---|---|---|---|---|---|---|
| T1-C1 | 0.00 ± 0.00 | -0.00 ± 0.00 | -0.00 ± 0.00 | -0.00 ± 0.00 | -0.01 ± 0.01 | -0.65 ± 0.06 |
| T2-C1 | 0.42 ± 0.10 | 0.27 ± 0.05 | 0.13 ± 0.03 | 0.15 ± 0.05 | 0.18 ± 0.03 | -0.95 ± 0.06 |
| T3-C1 | 0.58 ± 0.04 | 0.53 ± 0.06 | 0.32 ± 0.03 | 0.27 ± 0.02 | 0.26 ± 0.03 | -0.12 ± 0.04 |
| T4-C1 | 0.31 ± 0.06 | 0.45 ± 0.07 | 0.40 ± 0.09 | 0.36 ± 0.07 | 0.32 ± 0.05 | 0.18 ± 0.03 |
| T5-C1 | -0.07 ± 0.12 | 0.11 ± 0.09 | 0.28 ± 0.08 | 0.32 ± 0.08 | 0.37 ± 0.08 | -0.08 ± 0.04 |
| T1-C2 | -0.47 ± 0.14 | -0.41 ± 0.16 | -0.78 ± 0.18 | -0.73 ± 0.23 | -0.56 ± 0.25 | – |
| T2-C2 | -0.15 ± 0.15 | -0.47 ± 0.15 | -1.18 ± 0.24 | -1.60 ± 0.34 | -2.23 ± 0.21 | – |
| T3-C2 | 0.19 ± 0.10 | -0.06 ± 0.09 | -0.10 ± 0.05 | -0.24 ± 0.11 | -0.46 ± 0.14 | – |
| T4-C2 | -0.27 ± 0.15 | -0.07 ± 0.13 | 0.30 ± 0.11 | 0.18 ± 0.06 | 0.17 ± 0.05 | – |
| T5-C2 | -0.73 ± 0.21 | -0.31 ± 0.15 | -0.13 ± 0.16 | 0.09 ± 0.11 | 0.13 ± 0.13 | – |
| T1-C3 | -0.31 ± 0.10 | -0.80 ± 0.19 | -1.27 ± 0.31 | -1.56 ± 0.31 | -0.83 ± 0.34 | – |
| T2-C3 | -0.03 ± 0.07 | -0.42 ± 0.14 | -1.37 ± 0.23 | -2.02 ± 0.17 | -2.75 ± 0.30 | – |
| T3-C3 | 0.02 ± 0.16 | 0.04 ± 0.10 | -0.47 ± 0.19 | -0.49 ± 0.10 | -0.83 ± 0.21 | – |
| T4-C3 | -0.48 ± 0.13 | 0.13 ± 0.13 | 0.23 ± 0.11 | 0.27 ± 0.08 | 0.14 ± 0.06 | – |
| T5-C3 | -1.06 ± 0.17 | -0.25 ± 0.15 | 0.08 ± 0.18 | 0.28 ± 0.13 | 0.28 ± 0.15 | – |
| T1-C4 | -0.26 ± 0.14 | -1.18 ± 0.23 | -1.62 ± 0.20 | -1.46 ± 0.20 | -0.67 ± 0.23 | – |
| T2-C4 | 0.07 ± 0.08 | -0.77 ± 0.14 | -1.93 ± 0.24 | -2.34 ± 0.22 | -2.88 ± 0.22 | – |
| T3-C4 | 0.02 ± 0.17 | -0.10 ± 0.11 | -0.30 ± 0.07 | -0.80 ± 0.18 | -0.83 ± 0.20 | – |
| T4-C4 | -0.49 ± 0.14 | 0.33 ± 0.10 | 0.46 ± 0.12 | 0.62 ± 0.06 | 0.27 ± 0.14 | – |
| T5-C4 | -1.02 ± 0.14 | -0.85 ± 0.22 | 0.14 ± 0.17 | 0.47 ± 0.07 | 0.28 ± 0.14 | – |
| Avg ± SEM | -0.19 ± 0.02 | -0.19 ± 0.01 | -0.34 ± 0.03 | -0.41 ± 0.03 | -0.48 ± 0.03 | -0.32 ± 0.02 |

Table 39: Catcher $\mathcal{F}$ for MER.

| | T1 | T2 | T3 | T4 | T5 | Avg ± SEM |
|---|---|---|---|---|---|---|
| T1-C1 | 5.34 ± 0.23 | 3.21 ± 0.30 | 0.47 ± 0.06 | 0.19 ± 0.02 | 0.12 ± 0.01 | 0.08 ± 0.06 |
| T2-C1 | 1.98 ± 0.11 | 2.76 ± 0.17 | 1.26 ± 0.07 | 0.27 ± 0.05 | 0.07 ± 0.01 | -0.05 ± 0.05 |
| T3-C1 | 0.66 ± 0.19 | 3.11 ± 0.17 | 4.06 ± 0.08 | 2.60 ± 0.14 | 0.28 ± 0.04 | 1.08 ± 0.04 |
| T4-C1 | -3.68 ± 0.11 | -1.87 ± 0.19 | 3.23 ± 0.11 | 4.40 ± 0.08 | 1.25 ± 0.14 | 0.99 ± 0.05 |
| T5-C1 | -3.04 ± 0.21 | -4.11 ± 0.12 | -2.05 ± 0.19 | 1.94 ± 0.16 | 4.81 ± 0.15 | -0.38 ± 0.04 |
| T1-C2 | 2.97 ± 0.13 | 1.36 ± 0.19 | -2.64 ± 0.15 | -2.74 ± 0.10 | 0.16 ± 0.14 | – |
| T2-C2 | 4.72 ± 0.03 | 4.27 ± 0.09 | -0.25 ± 0.26 | -4.03 ± 0.11 | -4.94 ± 0.16 | – |
| T3-C2 | 0.40 ± 0.18 | 1.00 ± 0.11 | 2.52 ± 0.18 | 1.54 ± 0.14 | -1.01 ± 0.05 | – |
| T4-C2 | -2.37 ± 0.17 | -1.06 ± 0.21 | 2.93 ± 0.14 | 4.34 ± 0.04 | 1.36 ± 0.15 | – |
| T5-C2 | -3.67 ± 0.20 | -3.46 ± 0.19 | -0.94 ± 0.17 | 1.30 ± 0.10 | 4.69 ± 0.11 | – |
| T1-C3 | 1.42 ± 0.18 | 0.16 ± 0.24 | -2.15 ± 0.25 | -2.65 ± 0.13 | 0.17 ± 0.19 | – |
| T2-C3 | 4.30 ± 0.05 | 3.34 ± 0.14 | -1.61 ± 0.30 | -4.21 ± 0.10 | -5.00 ± 0.08 | – |
| T3-C3 | 0.28 ± 0.12 | 0.70 ± 0.18 | 1.66 ± 0.24 | 1.29 ± 0.16 | -1.03 ± 0.11 | – |
| T4-C3 | -2.40 ± 0.23 | -0.91 ± 0.25 | 2.87 ± 0.19 | 4.27 ± 0.09 | 1.89 ± 0.11 | – |
| T5-C3 | -3.34 ± 0.20 | -3.25 ± 0.21 | -0.69 ± 0.20 | 1.26 ± 0.12 | 4.76 ± 0.13 | – |
| T1-C4 | 1.51 ± 0.26 | -0.05 ± 0.30 | -2.40 ± 0.17 | -2.44 ± 0.13 | -0.33 ± 0.15 | – |
| T2-C4 | 3.90 ± 0.15 | 3.35 ± 0.15 | -1.82 ± 0.28 | -4.11 ± 0.11 | -5.23 ± 0.10 | – |
| T3-C4 | 0.40 ± 0.15 | 0.96 ± 0.13 | 1.97 ± 0.17 | 1.19 ± 0.22 | -1.05 ± 0.14 | – |
| T4-C4 | -2.23 ± 0.23 | -1.32 ± 0.25 | 2.96 ± 0.17 | 4.13 ± 0.10 | 2.02 ± 0.32 | – |
| T5-C4 | -3.38 ± 0.18 | -3.43 ± 0.13 | -0.95 ± 0.23 | 1.24 ± 0.20 | 4.77 ± 0.15 | – |
| Avg ± SEM | 0.19 ± 0.02 | 0.24 ± 0.01 | 0.42 ± 0.01 | 0.49 ± 0.00 | 0.39 ± 0.01 | 0.34 ± 0.01 |

Table 40: Catcher $\mathcal{W}$ for MER.

| | T1 | T2 | T3 | T4 | T5 | Avg ± SEM |
|---|---|---|---|---|---|---|
| T1-C1 | 0.00 ± 0.00 | 0.00 ± 0.00 | 0.00 ± 0.00 | 0.00 ± 0.00 | 0.00 ± 0.00 | -0.98 ± 0.04 |
| T2-C1 | 0.62 ± 0.07 | 0.54 ± 0.12 | 0.04 ± 0.01 | 0.01 ± 0.00 | 0.01 ± 0.00 | -1.40 ± 0.03 |
| T3-C1 | 0.51 ± 0.17 | 1.00 ± 0.00 | 0.48 ± 0.08 | 0.09 ± 0.02 | 0.03 ± 0.00 | -0.08 ± 0.02 |
| T4-C1 | -3.68 ± 0.11 | -1.89 ± 0.18 | 0.87 ± 0.05 | 0.80 ± 0.08 | 0.09 ± 0.02 | -0.43 ± 0.03 |
| T5-C1 | -3.04 ± 0.21 | -4.11 ± 0.12 | -2.05 ± 0.19 | 0.38 ± 0.10 | 0.38 ± 0.10 | -1.37 ± 0.04 |
| T1-C2 | -0.36 ± 0.06 | -0.77 ± 0.05 | -2.64 ± 0.15 | -2.74 ± 0.10 | 0.16 ± 0.14 | – |
| T2-C2 | 0.90 ± 0.02 | 0.63 ± 0.07 | -1.08 ± 0.12 | -4.03 ± 0.11 | -4.94 ± 0.16 | – |
| T3-C2 | 0.39 ± 0.18 | 0.82 ± 0.06 | 0.28 ± 0.11 | -1.03 ± 0.12 | -1.32 ± 0.03 | – |
| T4-C2 | -2.37 ± 0.17 | -1.06 ± 0.21 | 0.85 ± 0.02 | 0.90 ± 0.03 | 0.09 ± 0.02 | – |
| T5-C2 | -3.67 ± 0.20 | -3.46 ± 0.19 | -0.94 ± 0.17 | 0.81 ± 0.04 | 0.29 ± 0.04 | – |
| T1-C3 | -0.81 ± 0.06 | -0.90 ± 0.07 | -2.15 ± 0.25 | -2.65 ± 0.13 | 0.16 ± 0.19 | – |
| T2-C3 | 0.80 ± 0.03 | 0.17 ± 0.10 | -1.91 ± 0.22 | -4.21 ± 0.10 | -5.00 ± 0.08 | – |
| T3-C3 | 0.26 ± 0.11 | 0.54 ± 0.12 | -0.06 ± 0.09 | -1.06 ± 0.10 | -1.42 ± 0.09 | – |
| T4-C3 | -2.40 ± 0.23 | -0.92 ± 0.25 | 0.87 ± 0.02 | 0.89 ± 0.04 | 0.13 ± 0.05 | – |
| T5-C3 | -3.34 ± 0.20 | -3.25 ± 0.21 | -0.69 ± 0.20 | 0.81 ± 0.05 | 0.53 ± 0.11 | – |
| T1-C4 | -0.82 ± 0.05 | -0.99 ± 0.10 | -2.40 ± 0.17 | -2.44 ± 0.13 | -0.33 ± 0.15 | – |
| T2-C4 | 0.60 ± 0.10 | 0.26 ± 0.08 | -2.05 ± 0.20 | -4.11 ± 0.11 | -5.23 ± 0.10 | – |
| T3-C4 | 0.40 ± 0.15 | 0.74 ± 0.09 | 0.09 ± 0.17 | -0.92 ± 0.12 | -1.45 ± 0.11 | – |
| T4-C4 | -2.23 ± 0.23 | -1.32 ± 0.25 | 0.83 ± 0.03 | 0.86 ± 0.04 | 0.13 ± 0.04 | – |
| T5-C4 | -3.38 ± 0.18 | -3.43 ± 0.13 | -0.96 ± 0.22 | 0.69 ± 0.10 | 0.53 ± 0.10 | – |
| Avg ± SEM | -1.08 ± 0.03 | -0.87 ± 0.02 | -0.63 ± 0.02 | -0.82 ± 0.01 | -0.86 ± 0.01 | -0.85 ± 0.01 |

Table 41: Room $\mathcal{F}$ for L2.

| | T1 | T2 | T3 | T4 | T5 | Avg ± SEM |
|---|---|---|---|---|---|---|
| T1-C1 | 0.31 ± 0.31 | 0.59 ± 0.82 | 0.56 ± 0.36 | 0.52 ± 0.29 | -0.36 ± 0.67 | 0.04 ± 0.07 |
| T2-C1 | 1.72 ± 0.55 | 1.39 ± 0.33 | 1.77 ± 0.50 | 1.72 ± 0.52 | 1.39 ± 0.40 | 0.48 ± 0.08 |
| T3-C1 | 3.08 ± 0.40 | 2.45 ± 0.31 | 3.08 ± 0.42 | 2.95 ± 0.43 | 2.60 ± 0.37 | 0.78 ± 0.08 |
| T4-C1 | 2.19 ± 0.49 | 1.82 ± 0.52 | 2.00 ± 0.34 | 1.79 ± 0.38 | 1.74 ± 0.34 | 0.65 ± 0.08 |
| T5-C1 | 0.85 ± 0.26 | 0.65 ± 0.35 | 0.58 ± 0.29 | 0.59 ± 0.26 | 0.48 ± 0.38 | 0.05 ± 0.09 |
| T1-C2 | 0.10 ± 0.16 | 0.04 ± 0.50 | 0.32 ± 0.19 | 0.48 ± 0.19 | 0.23 ± 0.32 | – |
| T2-C2 | 0.07 ± 0.09 | 0.01 ± 0.26 | 0.17 ± 0.13 | 0.35 ± 0.09 | 0.10 ± 0.38 | – |
| T3-C2 | -0.06 ± 0.09 | -0.07 ± 0.26 | -0.09 ± 0.12 | 0.01 ± 0.11 | 0.07 ± 0.30 | – |
| T4-C2 | 0.31 ± 0.09 | -0.22 ± 0.26 | 0.50 ± 0.17 | 0.29 ± 0.14 | -0.01 ± 0.23 | – |
| T5-C2 | 0.02 ± 0.15 | -0.27 ± 0.30 | -0.07 ± 0.15 | -0.14 ± 0.12 | -0.11 ± 0.38 | – |
| T1-C3 | -0.15 ± 0.11 | -0.00 ± 0.40 | -0.30 ± 0.11 | 0.01 ± 0.13 | 0.16 ± 0.27 | – |
| T2-C3 | 0.02 ± 0.11 | 0.19 ± 0.39 | 0.20 ± 0.14 | 0.06 ± 0.11 | -0.05 ± 0.31 | – |
| T3-C3 | -0.14 ± 0.13 | 0.67 ± 0.52 | 0.24 ± 0.18 | 0.03 ± 0.10 | 0.27 ± 0.42 | – |
| T4-C3 | 0.27 ± 0.12 | 0.65 ± 0.31 | 0.30 ± 0.15 | 0.41 ± 0.16 | 0.27 ± 0.32 | – |
| T5-C3 | 0.08 ± 0.13 | -0.54 ± 0.36 | -0.16 ± 0.08 | -0.22 ± 0.09 | -0.21 ± 0.30 | – |
| T1-C4 | -0.06 ± 0.09 | -0.71 ± 0.31 | -0.17 ± 0.13 | -0.07 ± 0.10 | -0.65 ± 0.32 | – |
| T2-C4 | -0.15 ± 0.10 | 0.32 ± 0.37 | 0.17 ± 0.10 | 0.06 ± 0.13 | 0.02 ± 0.39 | – |
| T3-C4 | 0.15 ± 0.13 | 0.06 ± 0.34 | -0.08 ± 0.09 | -0.11 ± 0.13 | 0.57 ± 0.38 | – |
| T4-C4 | 0.42 ± 0.10 | -0.30 ± 0.39 | 0.13 ± 0.09 | 0.33 ± 0.19 | 0.03 ± 0.30 | – |
| T5-C4 | -0.25 ± 0.12 | 0.06 ± 0.43 | -0.01 ± 0.09 | -0.14 ± 0.16 | -0.19 ± 0.23 | – |
| Avg ± SEM | 0.44 ± 0.02 | 0.34 ± 0.03 | 0.46 ± 0.02 | 0.45 ± 0.03 | 0.32 ± 0.02 | 0.40 ± 0.01 |

Table 42: Room $\mathcal{W}$ for L2.

| | T1 | T2 | T3 | T4 | T5 | Avg ± SEM |
|---|---|---|---|---|---|---|
| T1-C1 | -0.31 ± 0.16 | -0.96 ± 0.29 | -0.26 ± 0.15 | -0.16 ± 0.08 | -1.22 ± 0.42 | -0.42 ± 0.04 |
| T2-C1 | 0.48 ± 0.21 | 0.33 ± 0.16 | 0.39 ± 0.18 | 0.45 ± 0.22 | 0.14 ± 0.12 | -0.11 ± 0.04 |
| T3-C1 | 0.42 ± 0.12 | 0.32 ± 0.16 | 0.48 ± 0.13 | 0.49 ± 0.15 | 0.28 ± 0.15 | -0.11 ± 0.04 |
| T4-C1 | 0.42 ± 0.11 | 0.23 ± 0.26 | 0.51 ± 0.09 | 0.45 ± 0.11 | 0.54 ± 0.11 | -0.03 ± 0.04 |
| T5-C1 | 0.32 ± 0.11 | 0.12 ± 0.21 | 0.17 ± 0.13 | 0.19 ± 0.13 | -0.18 ± 0.24 | -0.23 ± 0.05 |
| T1-C2 | -0.15 ± 0.11 | -0.59 ± 0.31 | -0.05 ± 0.08 | -0.05 ± 0.09 | -0.40 ± 0.17 | – |
| T2-C2 | -0.16 ± 0.06 | -0.40 ± 0.19 | -0.08 ± 0.07 | 0.02 ± 0.07 | -0.47 ± 0.27 | – |
| T3-C2 | -0.21 ± 0.05 | -0.58 ± 0.22 | -0.25 ± 0.09 | -0.13 ± 0.09 | -0.18 ± 0.25 | – |
| T4-C2 | 0.03 ± 0.06 | -0.46 ± 0.25 | -0.03 ± 0.05 | -0.06 ± 0.06 | -0.50 ± 0.16 | – |
| T5-C2 | -0.09 ± 0.13 | -0.48 ± 0.22 | -0.21 ± 0.10 | -0.24 ± 0.09 | -0.69 ± 0.27 | – |
| T1-C3 | -0.24 ± 0.09 | -0.51 ± 0.32 | -0.37 ± 0.10 | -0.22 ± 0.07 | -0.21 ± 0.16 | – |
| T2-C3 | -0.19 ± 0.08 | -0.38 ± 0.23 | -0.14 ± 0.06 | -0.16 ± 0.09 | -0.57 ± 0.28 | – |
| T3-C3 | -0.29 ± 0.10 | -0.33 ± 0.22 | -0.10 ± 0.10 | -0.20 ± 0.05 | -0.48 ± 0.23 | – |
| T4-C3 | -0.06 ± 0.07 | 0.01 ± 0.14 | 0.00 ± 0.11 | -0.05 ± 0.08 | -0.30 ± 0.24 | – |
| T5-C3 | -0.07 ± 0.09 | -0.70 ± 0.35 | -0.18 ± 0.08 | -0.31 ± 0.08 | -0.58 ± 0.26 | – |
| T1-C4 | -0.21 ± 0.06 | -0.92 ± 0.22 | -0.35 ± 0.10 | -0.28 ± 0.08 | -0.90 ± 0.24 | – |
| T2-C4 | -0.27 ± 0.06 | -0.42 ± 0.21 | -0.04 ± 0.05 | -0.10 ± 0.09 | -0.53 ± 0.28 | – |
| T3-C4 | -0.12 ± 0.07 | -0.48 ± 0.28 | -0.15 ± 0.07 | -0.22 ± 0.12 | -0.54 ± 0.18 | – |
| T4-C4 | 0.04 ± 0.04 | -0.68 ± 0.32 | -0.08 ± 0.08 | -0.09 ± 0.10 | -0.45 ± 0.23 | – |
| T5-C4 | -0.27 ± 0.11 | -0.62 ± 0.26 | -0.10 ± 0.08 | -0.18 ± 0.16 | -0.49 ± 0.19 | – |
| Avg ± SEM | -0.05 ± 0.01 | -0.38 ± 0.03 | -0.04 ± 0.02 | -0.04 ± 0.02 | -0.39 ± 0.02 | -0.18 ± 0.01 |

Table 43: Flappy $\mathcal{F}$ for L2.

| | T1 | T2 | T3 | T4 | T5 | Avg ± SEM |
|---|---|---|---|---|---|---|
| T1-C1 | 4.17 ± 0.22 | 2.56 ± 0.27 | 3.70 ± 0.30 | 3.45 ± 0.34 | 4.59 ± 0.33 | 0.89 ± 0.07 |
| T2-C1 | 1.78 ± 0.24 | 1.92 ± 0.30 | 1.41 ± 0.35 | 2.01 ± 0.17 | 1.67 ± 0.25 | 0.04 ± 0.13 |
| T3-C1 | 1.67 ± 0.27 | 1.70 ± 0.31 | 1.98 ± 0.35 | 2.16 ± 0.17 | 2.05 ± 0.22 | 0.32 ± 0.06 |
| T4-C1 | 0.08 ± 0.14 | -0.15 ± 0.24 | 0.30 ± 0.33 | 0.16 ± 0.18 | 0.10 ± 0.16 | 0.30 ± 0.08 |
| T5-C1 | 0.25 ± 0.26 | 0.28 ± 0.24 | -0.03 ± 0.34 | -0.28 ± 0.24 | 0.29 ± 0.06 | 0.32 ± 0.10 |
| T1-C2 | -0.31 ± 0.20 | -0.15 ± 0.24 | 0.21 ± 0.23 | -0.33 ± 0.14 | -0.15 ± 0.33 | – |
| T2-C2 | -0.41 ± 0.27 | -0.56 ± 0.26 | -0.70 ± 0.29 | -0.31 ± 0.21 | -0.91 ± 0.36 | – |
| T3-C2 | -0.05 ± 0.18 | -0.25 ± 0.19 | -0.79 ± 0.15 | -0.01 ± 0.19 | -0.02 ± 0.20 | – |
| T4-C2 | 0.21 ± 0.24 | 0.39 ± 0.30 | 0.48 ± 0.28 | 0.47 ± 0.23 | 0.31 ± 0.26 | – |
| T5-C2 | 0.31 ± 0.18 | 0.65 ± 0.16 | 0.78 ± 0.35 | 0.13 ± 0.32 | 0.35 ± 0.32 | – |
| T1-C3 | -0.20 ± 0.21 | 0.01 ± 0.30 | 0.08 ± 0.28 | -0.11 ± 0.26 | 0.28 ± 0.27 | – |
| T2-C3 | -0.41 ± 0.16 | -0.93 ± 0.15 | -0.82 ± 0.41 | -0.44 ± 0.23 | -0.23 ± 0.22 | – |
| T3-C3 | 0.18 ± 0.21 | -0.32 ± 0.23 | -0.73 ± 0.23 | -0.17 ± 0.17 | -0.59 ± 0.25 | – |
| T4-C3 | 0.20 ± 0.17 | 0.68 ± 0.13 | 0.50 ± 0.20 | 0.39 ± 0.22 | 0.07 ± 0.27 | – |
| T5-C3 | 0.02 ± 0.16 | 0.44 ± 0.07 | 0.71 ± 0.34 | 0.33 ± 0.16 | 0.45 ± 0.27 | – |
| T1-C4 | 0.19 ± 0.16 | -0.17 ± 0.14 | -0.03 ± 0.24 | -0.06 ± 0.22 | 0.13 ± 0.32 | – |
| T2-C4 | -0.08 ± 0.24 | -0.53 ± 0.17 | -0.74 ± 0.24 | -0.55 ± 0.18 | -0.32 ± 0.30 | – |
| T3-C4 | -0.08 ± 0.18 | 0.05 ± 0.22 | -0.11 ± 0.21 | -0.17 ± 0.20 | -0.12 ± 0.24 | – |
| T4-C4 | -0.19 ± 0.17 | 0.45 ± 0.15 | 0.74 ± 0.14 | 0.67 ± 0.21 | 0.09 ± 0.36 | – |
| T5-C4 | 0.05 ± 0.26 | 0.25 ± 0.20 | 0.26 ± 0.30 | 1.04 ± 0.18 | 0.16 ± 0.26 | – |
| Avg ± SEM | 0.37 ± 0.02 | 0.32 ± 0.02 | 0.36 ± 0.02 | 0.42 ± 0.03 | 0.41 ± 0.02 | 0.37 ± 0.02 |

Table 44: Flappy $\mathcal{W}$ for L2.

| | T1 | T2 | T3 | T4 | T5 | Avg ± SEM |
|---|---|---|---|---|---|---|
| T1-C1 | 0.00 ± 0.00 | 0.00 ± 0.00 | 0.00 ± 0.00 | -0.01 ± 0.01 | 0.00 ± 0.00 | -0.24 ± 0.04 |
| T2-C1 | 0.43 ± 0.10 | 0.48 ± 0.15 | 0.29 ± 0.15 | 0.62 ± 0.14 | 0.50 ± 0.14 | -0.43 ± 0.10 |
| T3-C1 | 0.73 ± 0.08 | 0.52 ± 0.08 | 0.73 ± 0.08 | 0.84 ± 0.10 | 0.74 ± 0.07 | -0.20 ± 0.04 |
| T4-C1 | -0.17 ± 0.11 | -0.41 ± 0.20 | -0.22 ± 0.17 | -0.26 ± 0.10 | -0.34 ± 0.17 | -0.23 ± 0.05 |
| T5-C1 | -0.14 ± 0.12 | -0.08 ± 0.12 | -0.52 ± 0.22 | -0.53 ± 0.16 | -0.02 ± 0.07 | -0.19 ± 0.06 |
| T1-C2 | -0.45 ± 0.15 | -0.33 ± 0.18 | -0.42 ± 0.21 | -0.52 ± 0.13 | -0.38 ± 0.20 | – |
| T2-C2 | -0.57 ± 0.24 | -0.82 ± 0.22 | -0.84 ± 0.22 | -0.59 ± 0.24 | -1.06 ± 0.31 | – |
| T3-C2 | -0.31 ± 0.13 | -0.49 ± 0.16 | -0.99 ± 0.14 | -0.32 ± 0.11 | -0.23 ± 0.19 | – |
| T4-C2 | -0.16 ± 0.15 | -0.18 ± 0.23 | -0.19 ± 0.13 | -0.21 ± 0.10 | -0.32 ± 0.30 | – |
| T5-C2 | -0.16 ± 0.09 | -0.05 ± 0.07 | 0.04 ± 0.16 | -0.50 ± 0.23 | -0.31 ± 0.18 | – |
| T1-C3 | -0.41 ± 0.14 | -0.27 ± 0.24 | -0.21 ± 0.18 | -0.33 ± 0.20 | -0.14 ± 0.16 | – |
| T2-C3 | -0.49 ± 0.17 | -0.96 ± 0.15 | -1.02 ± 0.31 | -0.73 ± 0.16 | -0.43 ± 0.15 | – |
| T3-C3 | -0.07 ± 0.16 | -0.62 ± 0.17 | -0.97 ± 0.19 | -0.49 ± 0.11 | -0.86 ± 0.21 | – |
| T4-C3 | -0.17 ± 0.11 | -0.10 ± 0.05 | -0.13 ± 0.14 | -0.19 ± 0.21 | -0.40 ± 0.22 | – |
| T5-C3 | -0.28 ± 0.10 | 0.12 ± 0.07 | -0.14 ± 0.20 | -0.16 ± 0.13 | -0.04 ± 0.14 | – |
| T1-C4 | -0.15 ± 0.09 | -0.24 ± 0.14 | -0.25 ± 0.22 | -0.36 ± 0.21 | -0.36 ± 0.23 | – |
| T2-C4 | -0.38 ± 0.18 | -0.70 ± 0.14 | -0.86 ± 0.18 | -0.74 ± 0.16 | -0.71 ± 0.20 | – |
| T3-C4 | -0.32 ± 0.12 | -0.35 ± 0.17 | -0.54 ± 0.11 | -0.54 ± 0.12 | -0.49 ± 0.21 | – |
| T4-C4 | -0.39 ± 0.11 | -0.02 ± 0.06 | -0.23 ± 0.10 | -0.03 ± 0.10 | -0.42 ± 0.18 | – |
| T5-C4 | -0.40 ± 0.13 | -0.17 ± 0.16 | -0.28 ± 0.17 | 0.14 ± 0.12 | -0.34 ± 0.15 | – |
| Avg ± SEM | -0.19 ± 0.02 | -0.23 ± 0.01 | -0.33 ± 0.02 | -0.25 ± 0.02 | -0.28 ± 0.03 | -0.26 ± 0.01 |

Table 45: Catcher $\mathcal{F}$ for L2.

| | T1 | T2 | T3 | T4 | T5 | Avg ± SEM |
|---|---|---|---|---|---|---|
| T1-C1 | 7.11 ± 0.22 | 4.32 ± 0.29 | 0.77 ± 0.10 | 0.47 ± 0.05 | 0.66 ± 0.04 | 0.80 ± 0.13 |
| T2-C1 | 1.16 ± 0.16 | 2.10 ± 0.27 | 0.80 ± 0.22 | 0.41 ± 0.11 | 0.31 ± 0.10 | -0.20 ± 0.14 |
| T3-C1 | 0.82 ± 0.20 | 2.21 ± 0.24 | 1.31 ± 0.22 | 0.57 ± 0.11 | 0.49 ± 0.11 | 0.15 ± 0.07 |
| T4-C1 | -1.07 ± 0.31 | -0.74 ± 0.31 | 1.08 ± 0.23 | 0.69 ± 0.16 | 0.57 ± 0.10 | 0.41 ± 0.13 |
| T5-C1 | -1.29 ± 0.34 | -1.25 ± 0.37 | 0.77 ± 0.11 | 1.20 ± 0.23 | 1.55 ± 0.24 | 0.44 ± 0.17 |
| T1-C2 | 0.53 ± 0.20 | 0.28 ± 0.23 | -0.18 ± 0.34 | 0.41 ± 0.27 | 0.85 ± 0.27 | – |
| T2-C2 | 1.81 ± 0.36 | 1.26 ± 0.32 | -1.07 ± 0.25 | -1.36 ± 0.22 | -1.69 ± 0.20 | – |
| T3-C2 | 0.56 ± 0.25 | 0.69 ± 0.44 | -0.11 ± 0.31 | -0.75 ± 0.20 | -1.22 ± 0.23 | – |
| T4-C2 | -1.21 ± 0.35 | -0.67 ± 0.31 | 1.24 ± 0.34 | 1.04 ± 0.19 | 0.65 ± 0.12 | – |
| T5-C2 | -1.67 ± 0.45 | -1.15 ± 0.32 | 0.97 ± 0.33 | 1.78 ± 0.20 | 2.22 ± 0.29 | – |
| T1-C3 | 0.49 ± 0.28 | 0.23 ± 0.31 | -0.29 ± 0.44 | 0.11 ± 0.26 | 0.48 ± 0.20 | – |
| T2-C3 | 1.75 ± 0.46 | 1.10 ± 0.36 | -1.64 ± 0.33 | -2.16 ± 0.25 | -2.51 ± 0.34 | – |
| T3-C3 | 0.41 ± 0.22 | 0.37 ± 0.21 | -0.23 ± 0.26 | -0.81 ± 0.09 | -1.05 ± 0.19 | – |
| T4-C3 | -1.18 ± 0.40 | -0.50 ± 0.18 | 1.67 ± 0.23 | 1.69 ± 0.32 | 1.06 ± 0.21 | – |
| T5-C3 | -1.39 ± 0.37 | -0.64 ± 0.33 | 1.12 ± 0.26 | 2.07 ± 0.33 | 2.15 ± 0.26 | – |
| T1-C4 | 0.59 ± 0.38 | 0.10 ± 0.21 | -0.68 ± 0.31 | -0.56 ± 0.34 | 0.32 ± 0.24 | – |
| T2-C4 | 1.61 ± 0.42 | 0.87 ± 0.35 | -1.93 ± 0.33 | -2.48 ± 0.30 | -2.44 ± 0.26 | – |
| T3-C4 | 0.58 ± 0.13 | 0.53 ± 0.24 | 0.41 ± 0.40 | -0.56 ± 0.32 | -1.18 ± 0.21 | – |
| T4-C4 | -0.66 ± 0.20 | -0.08 ± 0.18 | 2.08 ± 0.33 | 1.57 ± 0.26 | 0.92 ± 0.15 | – |
| T5-C4 | -1.59 ± 0.37 | -0.90 ± 0.31 | 0.18 ± 0.33 | 1.89 ± 0.19 | 2.83 ± 0.28 | – |
| Avg ± SEM | 0.37 ± 0.03 | 0.41 ± 0.02 | 0.31 ± 0.03 | 0.26 ± 0.03 | 0.25 ± 0.02 | 0.32 ± 0.02 |

Table 46: Catcher $\mathcal{W}$ for L2.

| | T1 | T2 | T3 | T4 | T5 | Avg ± SEM |
|---|---|---|---|---|---|---|
| T1-C1 | 0.00 ± 0.00 | 0.00 ± 0.00 | 0.00 ± 0.00 | 0.00 ± 0.00 | 0.00 ± 0.00 | -0.17 ± 0.06 |
| T2-C1 | 0.40 ± 0.06 | 0.60 ± 0.16 | 0.17 ± 0.06 | 0.04 ± 0.02 | 0.07 ± 0.03 | -0.77 ± 0.09 |
| T3-C1 | 0.62 ± 0.14 | 0.93 ± 0.03 | 0.21 ± 0.05 | 0.12 ± 0.03 | 0.11 ± 0.03 | -0.29 ± 0.04 |
| T4-C1 | -1.12 ± 0.30 | -0.81 ± 0.28 | 0.16 ± 0.09 | 0.05 ± 0.01 | 0.01 ± 0.03 | -0.25 ± 0.08 |
| T5-C1 | -1.36 ± 0.31 | -1.30 ± 0.35 | 0.18 ± 0.08 | 0.20 ± 0.04 | 0.20 ± 0.06 | -0.38 ± 0.12 |
| T1-C2 | -0.35 ± 0.10 | -0.23 ± 0.10 | -0.45 ± 0.22 | 0.12 ± 0.13 | 0.59 ± 0.19 | – |
| T2-C2 | 0.13 ± 0.07 | 0.08 ± 0.09 | -1.10 ± 0.24 | -1.36 ± 0.22 | -1.69 ± 0.20 | – |
| T3-C2 | 0.33 ± 0.19 | 0.10 ± 0.25 | -0.74 ± 0.15 | -0.97 ± 0.19 | -1.32 ± 0.23 | – |
| T4-C2 | -1.23 ± 0.35 | -0.71 ± 0.31 | 0.07 ± 0.09 | 0.09 ± 0.03 | 0.02 ± 0.03 | – |
| T5-C2 | -1.73 ± 0.43 | -1.17 ± 0.31 | 0.19 ± 0.11 | 0.26 ± 0.07 | 0.30 ± 0.07 | – |
| T1-C3 | -0.44 ± 0.12 | -0.36 ± 0.14 | -0.51 ± 0.33 | 0.05 ± 0.24 | 0.33 ± 0.14 | – |
| T2-C3 | 0.29 ± 0.12 | -0.03 ± 0.12 | -1.72 ± 0.30 | -2.16 ± 0.25 | -2.51 ± 0.34 | – |
| T3-C3 | 0.08 ± 0.14 | -0.01 ± 0.14 | -0.80 ± 0.12 | -0.98 ± 0.11 | -1.17 ± 0.17 | – |
| T4-C3 | -1.21 ± 0.38 | -0.53 ± 0.17 | 0.38 ± 0.08 | 0.16 ± 0.06 | 0.06 ± 0.02 | – |
| T5-C3 | -1.40 ± 0.36 | -0.76 ± 0.30 | 0.34 ± 0.13 | 0.28 ± 0.10 | 0.24 ± 0.03 | – |
| T1-C4 | -0.45 ± 0.15 | -0.49 ± 0.13 | -0.79 ± 0.27 | -0.60 ± 0.32 | 0.28 ± 0.23 | – |
| T2-C4 | 0.30 ± 0.11 | 0.01 ± 0.12 | -1.99 ± 0.28 | -2.48 ± 0.30 | -2.44 ± 0.26 | – |
| T3-C4 | 0.45 ± 0.11 | 0.12 ± 0.15 | -0.54 ± 0.18 | -0.95 ± 0.12 | -1.32 ± 0.19 | – |
| T4-C4 | -0.73 ± 0.19 | -0.24 ± 0.13 | 0.36 ± 0.11 | 0.14 ± 0.06 | 0.05 ± 0.02 | – |
| T5-C4 | -1.60 ± 0.37 | -0.92 ± 0.31 | -0.19 ± 0.23 | 0.40 ± 0.08 | 0.20 ± 0.04 | – |
| Avg ± SEM | -0.45 ± 0.10 | -0.29 ± 0.06 | -0.34 ± 0.03 | -0.38 ± 0.04 | -0.40 ± 0.03 | -0.37 ± 0.03 |

Table 47: Room $\mathcal{F}$ for EWC.

| | T1 | T2 | T3 | T4 | T5 | Avg ± SEM |
|---|---|---|---|---|---|---|
| T1-C1 | 0.26 ± 0.33 | 0.44 ± 0.28 | 0.43 ± 0.25 | -0.24 ± 0.33 | 0.27 ± 0.36 | 0.30 ± 0.07 |
| T2-C1 | 0.93 ± 0.38 | 0.75 ± 0.29 | 0.83 ± 0.35 | 0.83 ± 0.42 | 0.72 ± 0.28 | 0.36 ± 0.09 |
| T3-C1 | 3.09 ± 0.51 | 1.69 ± 0.30 | 3.10 ± 0.49 | 3.01 ± 0.53 | 1.76 ± 0.32 | 0.68 ± 0.10 |
| T4-C1 | 3.57 ± 0.44 | 0.97 ± 0.27 | 3.67 ± 0.41 | 3.60 ± 0.45 | 1.29 ± 0.22 | 0.64 ± 0.10 |
| T5-C1 | 1.32 ± 0.59 | 0.93 ± 0.33 | 1.32 ± 0.58 | 1.22 ± 0.58 | 1.20 ± 0.35 | 0.37 ± 0.12 |
| T1-C2 | 0.22 ± 0.33 | 1.36 ± 0.32 | 0.20 ± 0.30 | 0.25 ± 0.32 | 1.23 ± 0.43 | – |
| T2-C2 | 0.18 ± 0.13 | 0.94 ± 0.16 | 0.02 ± 0.23 | 0.14 ± 0.17 | 0.57 ± 0.16 | – |
| T3-C2 | 0.03 ± 0.10 | 0.16 ± 0.17 | 0.06 ± 0.10 | -0.07 ± 0.09 | 0.18 ± 0.16 | – |
| T4-C2 | 0.36 ± 0.13 | -0.49 ± 0.35 | 0.44 ± 0.15 | 0.48 ± 0.12 | -0.40 ± 0.24 | – |
| T5-C2 | -0.02 ± 0.07 | 0.47 ± 0.17 | -0.06 ± 0.08 | 0.02 ± 0.09 | 0.28 ± 0.21 | – |
| T1-C3 | -0.02 ± 0.05 | 0.54 ± 0.26 | -0.14 ± 0.07 | -0.03 ± 0.04 | 0.66 ± 0.20 | – |
| T2-C3 | 0.07 ± 0.06 | 0.30 ± 0.13 | 0.00 ± 0.07 | 0.14 ± 0.05 | 0.25 ± 0.16 | – |
| T3-C3 | -0.04 ± 0.06 | 0.32 ± 0.10 | 0.14 ± 0.07 | -0.00 ± 0.08 | -0.00 ± 0.15 | – |
| T4-C3 | 0.20 ± 0.09 | -0.52 ± 0.21 | 0.24 ± 0.07 | 0.07 ± 0.06 | -0.40 ± 0.24 | – |
| T5-C3 | 0.07 ± 0.04 | -0.03 ± 0.16 | -0.09 ± 0.06 | -0.03 ± 0.06 | 0.21 ± 0.15 | – |
| T1-C4 | -0.02 ± 0.03 | 0.32 ± 0.22 | -0.03 ± 0.04 | 0.01 ± 0.05 | 0.32 ± 0.22 | – |
| T2-C4 | -0.05 ± 0.05 | 0.37 ± 0.18 | 0.05 ± 0.04 | -0.02 ± 0.07 | 0.10 ± 0.20 | – |
| T3-C4 | 0.03 ± 0.03 | 0.07 ± 0.15 | -0.03 ± 0.03 | -0.01 ± 0.05 | 0.14 ± 0.13 | – |
| T4-C4 | 0.11 ± 0.05 | -0.41 ± 0.25 | 0.10 ± 0.04 | 0.17 ± 0.05 | -0.18 ± 0.13 | – |
| T5-C4 | -0.03 ± 0.03 | 0.26 ± 0.17 | 0.06 ± 0.05 | 0.04 ± 0.04 | 0.19 ± 0.11 | – |
| Avg ± SEM | 0.51 ± 0.02 | 0.42 ± 0.03 | 0.52 ± 0.02 | 0.48 ± 0.02 | 0.42 ± 0.04 | 0.47 ± 0.01 |

Table 48: Room $\mathcal{W}$ for EWC.

| | T1 | T2 | T3 | T4 | T5 | Avg ± SEM |
|---|---|---|---|---|---|---|
| T1-C1 | -0.33 ± 0.21 | -0.20 ± 0.15 | -0.13 ± 0.09 | -0.57 ± 0.24 | -0.33 ± 0.20 | -0.08 ± 0.03 |
| T2-C1 | 0.27 ± 0.16 | 0.15 ± 0.14 | 0.18 ± 0.14 | 0.15 ± 0.18 | 0.14 ± 0.14 | -0.01 ± 0.04 |
| T3-C1 | 0.54 ± 0.10 | 0.46 ± 0.12 | 0.51 ± 0.09 | 0.35 ± 0.12 | 0.37 ± 0.10 | 0.05 ± 0.03 |
| T4-C1 | 0.62 ± 0.12 | 0.16 ± 0.13 | 0.68 ± 0.12 | 0.69 ± 0.11 | 0.17 ± 0.07 | -0.03 ± 0.04 |
| T5-C1 | 0.22 ± 0.20 | -0.15 ± 0.19 | 0.25 ± 0.22 | 0.27 ± 0.21 | -0.12 ± 0.14 | -0.08 ± 0.06 |
| T1-C2 | -0.11 ± 0.15 | 0.32 ± 0.13 | -0.05 ± 0.16 | -0.05 ± 0.12 | 0.30 ± 0.16 | – |
| T2-C2 | -0.13 ± 0.05 | 0.01 ± 0.12 | -0.28 ± 0.14 | -0.15 ± 0.09 | 0.09 ± 0.12 | – |
| T3-C2 | -0.04 ± 0.09 | -0.15 ± 0.11 | -0.03 ± 0.10 | -0.10 ± 0.08 | -0.18 ± 0.10 | – |
| T4-C2 | -0.01 ± 0.02 | -0.53 ± 0.34 | 0.01 ± 0.03 | -0.01 ± 0.03 | -0.47 ± 0.23 | – |
| T5-C2 | -0.05 ± 0.06 | -0.35 ± 0.07 | -0.06 ± 0.08 | -0.02 ± 0.08 | -0.42 ± 0.09 | – |
| T1-C3 | -0.06 ± 0.04 | 0.09 ± 0.13 | -0.18 ± 0.05 | -0.05 ± 0.04 | 0.09 ± 0.11 | – |
| T2-C3 | -0.10 ± 0.04 | 0.04 ± 0.12 | -0.15 ± 0.04 | -0.05 ± 0.04 | -0.00 ± 0.12 | – |
| T3-C3 | -0.09 ± 0.05 | -0.03 ± 0.06 | 0.02 ± 0.04 | -0.08 ± 0.06 | -0.22 ± 0.13 | – |
| T4-C3 | -0.00 ± 0.02 | -0.56 ± 0.20 | 0.01 ± 0.01 | -0.06 ± 0.03 | -0.50 ± 0.12 | – |
| T5-C3 | 0.06 ± 0.04 | -0.32 ± 0.11 | -0.10 ± 0.06 | -0.05 ± 0.05 | -0.21 ± 0.10 | – |
| T1-C4 | -0.06 ± 0.03 | -0.05 ± 0.10 | -0.05 ± 0.03 | -0.05 ± 0.04 | -0.05 ± 0.12 | – |
| T2-C4 | -0.09 ± 0.04 | -0.04 ± 0.10 | -0.04 ± 0.03 | -0.09 ± 0.05 | -0.09 ± 0.14 | – |
| T3-C4 | 0.02 ± 0.02 | -0.17 ± 0.08 | -0.06 ± 0.04 | -0.05 ± 0.04 | -0.12 ± 0.11 | – |
| T4-C4 | -0.01 ± 0.01 | -0.52 ± 0.21 | -0.01 ± 0.02 | -0.00 ± 0.02 | -0.28 ± 0.11 | – |
| T5-C4 | -0.04 ± 0.03 | -0.19 ± 0.08 | -0.01 ± 0.03 | 0.01 ± 0.03 | -0.22 ± 0.06 | – |
| Avg ± SEM | 0.03 ± 0.02 | -0.10 ± 0.03 | 0.02 ± 0.02 | 0.00 ± 0.02 | -0.10 ± 0.03 | -0.03 ± 0.01 |

Table 49: Flappy $\mathcal{F}$ for EWC.

| | T1 | T2 | T3 | T4 | T5 | Avg ± SEM |
|---|---|---|---|---|---|---|
| T1-C1 | 3.57 ± 0.22 | 3.14 ± 0.28 | 2.99 ± 0.15 | 2.95 ± 0.21 | 3.57 ± 0.28 | 0.83 ± 0.04 |
| T2-C1 | 2.05 ± 0.20 | 1.97 ± 0.31 | 1.94 ± 0.34 | 1.79 ± 0.35 | 1.40 ± 0.17 | 0.51 ± 0.05 |
| T3-C1 | 2.06 ± 0.22 | 2.04 ± 0.37 | 1.89 ± 0.28 | 2.12 ± 0.34 | 1.82 ± 0.42 | 0.49 ± 0.07 |
| T4-C1 | 0.10 ± 0.23 | 0.37 ± 0.29 | 0.21 ± 0.44 | 0.21 ± 0.37 | 0.59 ± 0.34 | 0.02 ± 0.08 |
| T5-C1 | 0.07 ± 0.18 | 0.06 ± 0.21 | 0.04 ± 0.20 | 0.19 ± 0.22 | -0.04 ± 0.24 | -0.00 ± 0.06 |
| T1-C2 | -0.21 ± 0.23 | -0.02 ± 0.19 | 0.06 ± 0.20 | 0.39 ± 0.13 | -0.10 ± 0.21 | – |
| T2-C2 | 0.20 ± 0.10 | 0.17 ± 0.34 | 0.36 ± 0.25 | -0.14 ± 0.30 | 0.27 ± 0.17 | – |
| T3-C2 | 0.31 ± 0.14 | 0.28 ± 0.24 | -0.34 ± 0.16 | -0.05 ± 0.28 | 0.32 ± 0.21 | – |
| T4-C2 | 0.06 ± 0.14 | 0.05 ± 0.22 | -0.20 ± 0.30 | -0.03 ± 0.23 | 0.13 ± 0.30 | – |
| T5-C2 | 0.09 ± 0.17 | -0.42 ± 0.21 | 0.13 ± 0.35 | -0.38 ± 0.18 | -0.18 ± 0.38 | – |
| T1-C3 | -0.00 ± 0.15 | 0.16 ± 0.18 | -0.19 ± 0.26 | 0.13 ± 0.21 | -0.02 ± 0.19 | – |
| T2-C3 | -0.02 ± 0.21 | 0.09 ± 0.27 | -0.24 ± 0.23 | 0.09 ± 0.27 | -0.09 ± 0.24 | – |
| T3-C3 | -0.11 ± 0.21 | -0.31 ± 0.28 | 0.20 ± 0.20 | -0.40 ± 0.16 | -0.36 ± 0.29 | – |
| T4-C3 | -0.27 ± 0.20 | 0.01 ± 0.15 | 0.47 ± 0.35 | 0.03 ± 0.35 | -0.14 ± 0.24 | – |
| T5-C3 | -0.22 ± 0.15 | 0.12 ± 0.25 | 0.14 ± 0.37 | 0.61 ± 0.15 | 0.01 ± 0.25 | – |
| T1-C4 | 0.17 ± 0.15 | -0.19 ± 0.28 | -0.30 ± 0.28 | 0.15 ± 0.19 | 0.36 ± 0.27 | – |
| T2-C4 | 0.47 ± 0.23 | -0.04 ± 0.23 | -0.41 ± 0.34 | -0.04 ± 0.26 | 0.30 ± 0.19 | – |
| T3-C4 | 0.05 ± 0.21 | 0.15 ± 0.24 | 0.34 ± 0.30 | -0.01 ± 0.38 | -0.24 ± 0.30 | – |
| T4-C4 | -0.66 ± 0.25 | -0.11 ± 0.22 | 0.15 ± 0.16 | -0.35 ± 0.21 | -0.28 ± 0.23 | – |
| T5-C4 | -0.07 ± 0.14 | -0.07 ± 0.25 | -0.18 ± 0.21 | -0.15 ± 0.40 | 0.24 ± 0.26 | – |
| Avg ± SEM | 0.38 ± 0.01 | 0.37 ± 0.02 | 0.35 ± 0.02 | 0.35 ± 0.02 | 0.38 ± 0.03 | 0.37 ± 0.02 |

Table 50: Flappy $\mathcal{W}$ for EWC.

| | T1 | T2 | T3 | T4 | T5 | Avg ± SEM |
|---|---|---|---|---|---|---|
| T1-C1 | 0.00 ± 0.00 | 0.00 ± 0.00 | 0.00 ± 0.00 | 0.00 ± 0.00 | 0.00 ± 0.00 | -0.24 ± 0.02 |
| T2-C1 | 0.56 ± 0.09 | 0.34 ± 0.13 | 0.45 ± 0.13 | 0.52 ± 0.15 | 0.40 ± 0.11 | -0.15 ± 0.05 |
| T3-C1 | 0.77 ± 0.06 | 0.66 ± 0.17 | 0.77 ± 0.09 | 0.81 ± 0.11 | 0.54 ± 0.15 | -0.12 ± 0.03 |
| T4-C1 | -0.21 ± 0.15 | -0.02 ± 0.21 | -0.36 ± 0.33 | -0.30 ± 0.26 | -0.10 ± 0.18 | -0.37 ± 0.04 |
| T5-C1 | -0.17 ± 0.14 | -0.30 ± 0.15 | -0.41 ± 0.18 | -0.18 ± 0.16 | -0.39 ± 0.20 | -0.35 ± 0.05 |
| T1-C2 | -0.52 ± 0.14 | -0.32 ± 0.14 | -0.36 ± 0.16 | -0.09 ± 0.13 | -0.36 ± 0.18 | – |
| T2-C2 | -0.19 ± 0.08 | -0.35 ± 0.24 | -0.05 ± 0.19 | -0.54 ± 0.22 | -0.25 ± 0.11 | – |
| T3-C2 | -0.14 ± 0.10 | -0.24 ± 0.12 | -0.54 ± 0.11 | -0.52 ± 0.20 | -0.34 ± 0.15 | – |
| T4-C2 | -0.16 ± 0.12 | -0.42 ± 0.12 | -0.63 ± 0.15 | -0.36 ± 0.17 | -0.35 ± 0.22 | – |
| T5-C2 | -0.25 ± 0.09 | -0.52 ± 0.16 | -0.27 ± 0.22 | -0.51 ± 0.14 | -0.56 ± 0.21 | – |
| T1-C3 | -0.17 ± 0.13 | -0.18 ± 0.12 | -0.47 ± 0.21 | -0.37 ± 0.11 | -0.36 ± 0.13 | – |
| T2-C3 | -0.31 ± 0.15 | -0.43 ± 0.16 | -0.59 ± 0.20 | -0.24 ± 0.19 | -0.44 ± 0.17 | – |
| T3-C3 | -0.43 ± 0.15 | -0.56 ± 0.19 | -0.25 ± 0.15 | -0.56 ± 0.14 | -0.58 ± 0.25 | – |
| T4-C3 | -0.45 ± 0.16 | -0.26 ± 0.11 | -0.18 ± 0.12 | -0.35 ± 0.28 | -0.44 ± 0.17 | – |
| T5-C3 | -0.44 ± 0.12 | -0.35 ± 0.15 | -0.42 ± 0.22 | 0.06 ± 0.14 | -0.36 ± 0.12 | – |
| T1-C4 | -0.12 ± 0.11 | -0.51 ± 0.19 | -0.59 ± 0.20 | -0.48 ± 0.13 | 0.02 ± 0.14 | – |
| T2-C4 | -0.15 ± 0.10 | -0.38 ± 0.15 | -0.77 ± 0.28 | -0.47 ± 0.15 | -0.11 ± 0.16 | – |
| T3-C4 | -0.31 ± 0.13 | -0.31 ± 0.14 | -0.27 ± 0.20 | -0.28 ± 0.18 | -0.54 ± 0.28 | – |
| T4-C4 | -0.85 ± 0.19 | -0.50 ± 0.12 | -0.29 ± 0.15 | -0.60 ± 0.16 | -0.64 ± 0.16 | – |
| T5-C4 | -0.29 ± 0.11 | -0.38 ± 0.17 | -0.46 ± 0.19 | -0.60 ± 0.32 | -0.26 ± 0.15 | – |
| Avg ± SEM | -0.19 ± 0.02 | -0.25 ± 0.03 | -0.29 ± 0.02 | -0.25 ± 0.03 | -0.26 ± 0.02 | -0.25 ± 0.01 |

Table 51: Catcher $\mathcal{F}$ for EWC.

| | T1 | T2 | T3 | T4 | T5 | Avg ± SEM |
|---|---|---|---|---|---|---|
| T1-C1 | 6.97 ± 0.17 | 4.25 ± 0.35 | 0.62 ± 0.05 | 0.25 ± 0.02 | 0.35 ± 0.03 | 0.74 ± 0.06 |
| T2-C1 | 1.39 ± 0.11 | 2.26 ± 0.14 | 0.96 ± 0.21 | 0.24 ± 0.06 | 0.13 ± 0.03 | 0.31 ± 0.05 |
| T3-C1 | 1.12 ± 0.18 | 2.59 ± 0.24 | 2.36 ± 0.33 | 0.48 ± 0.07 | 0.20 ± 0.04 | 0.33 ± 0.05 |
| T4-C1 | -0.53 ± 0.23 | 0.19 ± 0.25 | 2.58 ± 0.37 | 1.65 ± 0.37 | 0.38 ± 0.12 | 0.24 ± 0.05 |
| T5-C1 | -0.94 ± 0.35 | -0.47 ± 0.22 | 1.34 ± 0.44 | 2.40 ± 0.46 | 1.18 ± 0.27 | 0.17 ± 0.07 |
| T1-C2 | -0.45 ± 0.41 | -0.07 ± 0.17 | 0.34 ± 0.28 | 1.31 ± 0.23 | 1.04 ± 0.26 | – |
| T2-C2 | 0.43 ± 0.36 | 0.26 ± 0.27 | 0.11 ± 0.18 | 0.02 ± 0.20 | -0.28 ± 0.16 | – |
| T3-C2 | 0.17 ± 0.19 | -0.18 ± 0.28 | 0.10 ± 0.17 | -0.27 ± 0.16 | -0.39 ± 0.21 | – |
| T4-C2 | 0.02 ± 0.10 | 0.10 ± 0.08 | -0.05 ± 0.19 | 0.19 ± 0.20 | 0.15 ± 0.09 | – |
| T5-C2 | -0.03 ± 0.09 | 0.29 ± 0.23 | -0.12 ± 0.09 | 0.25 ± 0.14 | 0.30 ± 0.13 | – |
| T1-C3 | -0.12 ± 0.11 | -0.24 ± 0.15 | 0.06 ± 0.13 | 0.12 ± 0.14 | 0.12 ± 0.09 | – |
| T2-C3 | 0.08 ± 0.06 | -0.06 ± 0.11 | 0.13 ± 0.10 | 0.02 ± 0.08 | -0.01 ± 0.17 | – |
| T3-C3 | 0.22 ± 0.12 | 0.22 ± 0.15 | 0.33 ± 0.10 | 0.04 ± 0.09 | -0.00 ± 0.11 | – |
| T4-C3 | -0.04 ± 0.09 | -0.02 ± 0.05 | 0.23 ± 0.14 | 0.03 ± 0.06 | 0.05 ± 0.14 | – |
| T5-C3 | -0.18 ± 0.12 | -0.22 ± 0.20 | -0.05 ± 0.07 | -0.12 ± 0.12 | 0.23 ± 0.14 | – |
| T1-C4 | 0.05 ± 0.11 | 0.10 ± 0.05 | -0.13 ± 0.08 | 0.08 ± 0.06 | 0.23 ± 0.16 | – |
| T2-C4 | 0.21 ± 0.12 | 0.20 ± 0.20 | -0.01 ± 0.09 | 0.07 ± 0.09 | 0.09 ± 0.21 | – |
| T3-C4 | -0.11 ± 0.19 | -0.07 ± 0.05 | 0.05 ± 0.10 | -0.12 ± 0.07 | -0.09 ± 0.16 | – |
| T4-C4 | -0.08 ± 0.09 | 0.01 ± 0.06 | -0.05 ± 0.14 | -0.06 ± 0.09 | -0.02 ± 0.10 | – |
| T5-C4 | -0.11 ± 0.16 | -0.13 ± 0.11 | -0.12 ± 0.10 | 0.00 ± 0.08 | -0.04 ± 0.13 | – |
| Avg ± SEM | 0.40 ± 0.04 | 0.45 ± 0.03 | 0.43 ± 0.03 | 0.33 ± 0.04 | 0.18 ± 0.03 | 0.36 ± 0.03 |

Table 52: Catcher $\mathcal{W}$ for EWC.

| | T1 | T2 | T3 | T4 | T5 | Avg ± SEM |
|---|---|---|---|---|---|---|
| T1-C1 | 0.00 ± 0.00 | 0.00 ± 0.00 | 0.00 ± 0.00 | 0.00 ± 0.00 | 0.00 ± 0.00 | -0.06 ± 0.03 |
| T2-C1 | 0.46 ± 0.04 | 0.61 ± 0.13 | 0.13 ± 0.04 | 0.06 ± 0.01 | 0.05 ± 0.01 | -0.01 ± 0.03 |
| T3-C1 | 0.79 ± 0.08 | 0.96 ± 0.04 | 0.44 ± 0.06 | 0.10 ± 0.01 | 0.07 ± 0.01 | 0.01 ± 0.02 |
| T4-C1 | -0.55 ± 0.23 | -0.02 ± 0.18 | 0.54 ± 0.11 | 0.13 ± 0.03 | 0.02 ± 0.02 | -0.06 ± 0.04 |
| T5-C1 | -1.02 ± 0.32 | -0.56 ± 0.19 | 0.23 ± 0.14 | 0.45 ± 0.10 | 0.14 ± 0.04 | -0.13 ± 0.04 |
| T1-C2 | -0.79 ± 0.31 | -0.30 ± 0.08 | -0.02 ± 0.14 | 0.48 ± 0.08 | 0.36 ± 0.10 | – |
| T2-C2 | -0.09 ± 0.12 | -0.10 ± 0.13 | -0.09 ± 0.11 | -0.17 ± 0.12 | -0.32 ± 0.15 | – |
| T3-C2 | -0.12 ± 0.15 | -0.33 ± 0.24 | -0.08 ± 0.10 | -0.41 ± 0.12 | -0.46 ± 0.20 | – |
| T4-C2 | -0.09 ± 0.07 | -0.02 ± 0.03 | -0.21 ± 0.12 | -0.10 ± 0.12 | -0.01 ± 0.03 | – |
| T5-C2 | -0.08 ± 0.08 | -0.03 ± 0.06 | -0.20 ± 0.07 | -0.04 ± 0.07 | -0.00 ± 0.05 | – |
| T1-C3 | -0.22 ± 0.10 | -0.26 ± 0.15 | -0.09 ± 0.08 | -0.04 ± 0.09 | -0.05 ± 0.07 | – |
| T2-C3 | -0.03 ± 0.03 | -0.09 ± 0.10 | -0.06 ± 0.06 | -0.09 ± 0.06 | -0.16 ± 0.10 | – |
| T3-C3 | 0.03 ± 0.07 | 0.04 ± 0.04 | 0.03 ± 0.03 | -0.11 ± 0.06 | -0.10 ± 0.09 | – |
| T4-C3 | -0.11 ± 0.08 | -0.08 ± 0.05 | -0.00 ± 0.06 | -0.06 ± 0.05 | -0.11 ± 0.11 | – |
| T5-C3 | -0.23 ± 0.12 | -0.24 ± 0.20 | -0.08 ± 0.06 | -0.19 ± 0.09 | -0.04 ± 0.09 | – |
| T1-C4 | -0.07 ± 0.06 | 0.02 ± 0.01 | -0.14 ± 0.08 | -0.04 ± 0.03 | 0.03 ± 0.10 | – |
| T2-C4 | 0.02 ± 0.05 | 0.01 ± 0.05 | -0.09 ± 0.05 | -0.06 ± 0.03 | -0.10 ± 0.09 | – |
| T3-C4 | -0.18 ± 0.17 | -0.09 ± 0.05 | -0.05 ± 0.04 | -0.15 ± 0.05 | -0.19 ± 0.16 | – |
| T4-C4 | -0.13 ± 0.08 | -0.07 ± 0.06 | -0.16 ± 0.11 | -0.13 ± 0.06 | -0.13 ± 0.08 | – |
| T5-C4 | -0.20 ± 0.14 | -0.15 ± 0.10 | -0.20 ± 0.08 | -0.10 ± 0.06 | -0.12 ± 0.12 | – |
| Avg ± SEM | -0.13 ± 0.04 | -0.04 ± 0.04 | -0.01 ± 0.03 | -0.02 ± 0.02 | -0.06 ± 0.02 | -0.05 ± 0.02 |

Table 53: Room $\mathcal{F}$ for PackNet.

| | T1 | T2 | T3 | T4 | T5 | Avg ± SEM |
|---|---|---|---|---|---|---|
| T1-C1 | -0.23 ± 0.35 | -0.40 ± 0.31 | -0.19 ± 0.32 | -0.33 ± 0.23 | -0.21 ± 0.25 | 0.45 ± 0.04 |
| T2-C1 | 0.17 ± 0.11 | 1.00 ± 0.28 | 1.24 ± 0.25 | 1.01 ± 0.25 | 1.09 ± 0.26 | 0.81 ± 0.04 |
| T3-C1 | 0.05 ± 0.08 | 2.45 ± 0.47 | 4.63 ± 0.12 | 4.58 ± 0.10 | 2.67 ± 0.25 | 0.66 ± 0.06 |
| T4-C1 | -0.14 ± 0.11 | 0.97 ± 0.25 | 3.84 ± 0.28 | 4.16 ± 0.26 | 1.26 ± 0.33 | 0.30 ± 0.05 |
| T5-C1 | -0.08 ± 0.06 | -0.12 ± 0.26 | 0.23 ± 0.13 | 0.21 ± 0.22 | 0.95 ± 0.30 | 0.04 ± 0.04 |
| T1-C2 | 5.10 ± 0.07 | 1.89 ± 0.48 | 0.10 ± 0.06 | 0.01 ± 0.01 | 1.80 ± 0.16 | – |
| T2-C2 | 4.94 ± 0.09 | 2.21 ± 0.47 | 0.09 ± 0.04 | 0.21 ± 0.13 | 0.80 ± 0.15 | – |
| T3-C2 | -0.07 ± 0.05 | -0.28 ± 0.33 | -0.06 ± 0.04 | -0.08 ± 0.04 | -0.18 ± 0.23 | – |
| T4-C2 | 0.23 ± 0.05 | -1.05 ± 0.23 | 0.24 ± 0.04 | 0.20 ± 0.05 | -0.76 ± 0.12 | – |
| T5-C2 | -0.13 ± 0.08 | 0.51 ± 0.35 | -0.23 ± 0.06 | -0.24 ± 0.05 | -0.05 ± 0.30 | – |
| T1-C3 | -0.02 ± 0.02 | 0.67 ± 0.23 | -0.03 ± 0.04 | -0.04 ± 0.02 | 0.40 ± 0.28 | – |
| T2-C3 | 0.05 ± 0.04 | 0.69 ± 0.26 | 0.19 ± 0.10 | 0.07 ± 0.08 | 0.73 ± 0.20 | – |
| T3-C3 | 0.00 ± 0.03 | -0.26 ± 0.26 | -0.00 ± 0.04 | -0.01 ± 0.04 | -0.33 ± 0.33 | – |
| T4-C3 | 0.18 ± 0.04 | -1.25 ± 0.29 | 0.14 ± 0.08 | 0.26 ± 0.05 | -0.84 ± 0.23 | – |
| T5-C3 | -0.09 ± 0.05 | -0.06 ± 0.17 | -0.11 ± 0.06 | -0.19 ± 0.07 | 0.37 ± 0.29 | – |
| T1-C4 | -0.06 ± 0.02 | 0.29 ± 0.34 | -0.04 ± 0.03 | -0.05 ± 0.02 | 0.29 ± 0.26 | – |
| T2-C4 | 0.05 ± 0.05 | 1.00 ± 0.19 | -0.02 ± 0.06 | 0.16 ± 0.07 | 0.55 ± 0.21 | – |
| T3-C4 | -0.01 ± 0.05 | 0.23 ± 0.30 | 0.02 ± 0.04 | 0.01 ± 0.03 | -0.13 ± 0.26 | – |
| T4-C4 | 0.07 ± 0.02 | -1.05 ± 0.22 | 0.18 ± 0.04 | 0.08 ± 0.03 | -0.71 ± 0.24 | – |
| T5-C4 | -0.11 ± 0.07 | 0.26 ± 0.33 | -0.22 ± 0.06 | -0.25 ± 0.08 | 0.24 ± 0.16 | – |
| Avg ± SEM | 0.50 ± 0.02 | 0.38 ± 0.02 | 0.50 ± 0.01 | 0.49 ± 0.01 | 0.40 ± 0.02 | 0.45 ± 0.01 |

Table 54: Room $\mathcal{W}$ for PackNet.

| | T1 | T2 | T3 | T4 | T5 | Avg ± SEM |
|---|---|---|---|---|---|---|
| T1-C1 | -0.53 ± 0.28 | -0.72 ± 0.20 | -0.50 ± 0.23 | -0.52 ± 0.18 | -0.54 ± 0.16 | -0.07 ± 0.03 |
| T2-C1 | 0.01 ± 0.06 | -0.02 ± 0.04 | 0.05 ± 0.03 | 0.11 ± 0.07 | 0.06 ± 0.08 | 0.06 ± 0.02 |
| T3-C1 | -0.06 ± 0.05 | 0.57 ± 0.12 | 0.56 ± 0.12 | 0.52 ± 0.11 | 0.47 ± 0.11 | -0.05 ± 0.04 |
| T4-C1 | -0.18 ± 0.10 | 0.37 ± 0.14 | 0.98 ± 0.03 | 0.95 ± 0.04 | 0.31 ± 0.12 | -0.17 ± 0.03 |
| T5-C1 | -0.16 ± 0.07 | -0.53 ± 0.18 | 0.20 ± 0.13 | 0.14 ± 0.20 | -0.17 ± 0.17 | -0.31 ± 0.03 |
| T1-C2 | 0.96 ± 0.04 | 0.46 ± 0.17 | -0.02 ± 0.01 | 0.01 ± 0.01 | 0.53 ± 0.09 | – |
| T2-C2 | 1.03 ± 0.02 | 0.18 ± 0.13 | -0.05 ± 0.09 | -0.09 ± 0.05 | 0.20 ± 0.09 | – |
| T3-C2 | -0.07 ± 0.05 | -0.54 ± 0.30 | -0.08 ± 0.04 | -0.08 ± 0.04 | -0.47 ± 0.15 | – |
| T4-C2 | 0.00 ± 0.00 | -1.09 ± 0.21 | -0.00 ± 0.02 | -0.00 ± 0.00 | -0.76 ± 0.12 | – |
| T5-C2 | -0.14 ± 0.08 | -0.62 ± 0.15 | -0.25 ± 0.06 | -0.26 ± 0.05 | -0.69 ± 0.20 | – |
| T1-C3 | -0.03 ± 0.02 | 0.19 ± 0.13 | -0.05 ± 0.03 | -0.04 ± 0.02 | -0.12 ± 0.16 | – |
| T2-C3 | -0.03 ± 0.02 | -0.07 ± 0.13 | -0.05 ± 0.04 | -0.12 ± 0.04 | -0.07 ± 0.12 | – |
| T3-C3 | -0.00 ± 0.03 | -0.60 ± 0.13 | -0.02 ± 0.03 | -0.02 ± 0.03 | -0.67 ± 0.21 | – |
| T4-C3 | -0.00 ± 0.00 | -1.28 ± 0.28 | -0.01 ± 0.01 | 0.01 ± 0.01 | -0.84 ± 0.23 | – |
| T5-C3 | -0.10 ± 0.05 | -0.78 ± 0.12 | -0.12 ± 0.05 | -0.23 ± 0.06 | -0.59 ± 0.16 | – |
| T1-C4 | -0.06 ± 0.02 | -0.19 ± 0.19 | -0.05 ± 0.03 | -0.05 ± 0.02 | -0.14 ± 0.17 | – |
| T2-C4 | -0.03 ± 0.02 | 0.13 ± 0.09 | -0.09 ± 0.03 | -0.03 ± 0.02 | 0.06 ± 0.13 | – |
| T3-C4 | -0.02 ± 0.04 | -0.21 ± 0.20 | 0.02 ± 0.04 | -0.00 ± 0.03 | -0.35 ± 0.20 | – |
| T4-C4 | -0.00 ± 0.00 | -1.06 ± 0.22 | -0.01 ± 0.01 | -0.01 ± 0.01 | -0.82 ± 0.17 | – |
| T5-C4 | -0.13 ± 0.07 | -0.74 ± 0.15 | -0.22 ± 0.06 | -0.29 ± 0.06 | -0.51 ± 0.09 | – |
| Avg ± SEM | 0.02 ± 0.02 | -0.33 ± 0.03 | 0.01 ± 0.02 | -0.00 ± 0.01 | -0.25 ± 0.02 | -0.11 ± 0.01 |

Table 55: Flappy $\mathcal{F}$ for PackNet.

| | T1 | T2 | T3 | T4 | T5 | Avg ± SEM |
|---|---|---|---|---|---|---|
| T1-C1 | 2.39 ± 0.24 | 1.53 ± 0.18 | 1.05 ± 0.07 | 1.25 ± 0.09 | 1.68 ± 0.14 | -0.44 ± 0.09 |
| T2-C1 | 1.08 ± 0.13 | 1.52 ± 0.18 | 1.05 ± 0.15 | 0.74 ± 0.09 | 0.73 ± 0.10 | -0.52 ± 0.04 |
| T3-C1 | 0.43 ± 0.25 | 2.77 ± 0.16 | 2.43 ± 0.14 | 1.71 ± 0.09 | 1.43 ± 0.21 | 1.19 ± 0.06 |
| T4-C1 | -1.00 ± 0.26 | 1.20 ± 0.17 | 1.70 ± 0.21 | 2.02 ± 0.20 | 2.02 ± 0.22 | 1.13 ± 0.04 |
| T5-C1 | -0.70 ± 0.16 | -0.09 ± 0.21 | 0.57 ± 0.34 | 1.37 ± 0.27 | 2.92 ± 0.30 | 0.50 ± 0.06 |
| T1-C2 | 2.41 ± 0.26 | -0.30 ± 0.16 | -1.35 ± 0.16 | -1.46 ± 0.29 | -1.34 ± 0.28 | – |
| T2-C2 | 3.51 ± 0.31 | 0.25 ± 0.21 | -1.04 ± 0.27 | -2.17 ± 0.21 | -3.32 ± 0.26 | – |
| T3-C2 | 0.43 ± 0.19 | 0.92 ± 0.21 | 1.49 ± 0.12 | 0.45 ± 0.14 | 0.18 ± 0.27 | – |
| T4-C2 | -0.72 ± 0.16 | 0.45 ± 0.22 | 1.39 ± 0.23 | 2.18 ± 0.25 | 1.80 ± 0.22 | – |
| T5-C2 | -0.87 ± 0.13 | -0.62 ± 0.20 | 0.15 ± 0.28 | 2.08 ± 0.20 | 2.20 ± 0.23 | – |
| T1-C3 | -0.05 ± 0.18 | -1.51 ± 0.16 | -2.04 ± 0.39 | -1.93 ± 0.26 | -1.70 ± 0.17 | – |
| T2-C3 | 1.28 ± 0.22 | 0.09 ± 0.17 | -1.28 ± 0.24 | -3.02 ± 0.20 | -3.12 ± 0.23 | – |
| T3-C3 | 0.80 ± 0.16 | 1.87 ± 0.18 | 1.72 ± 0.24 | 0.70 ± 0.20 | -0.01 ± 0.20 | – |
| T4-C3 | -0.98 ± 0.18 | 0.38 ± 0.16 | 1.79 ± 0.15 | 2.18 ± 0.15 | 1.43 ± 0.18 | – |
| T5-C3 | -1.37 ± 0.19 | -1.28 ± 0.14 | 0.04 ± 0.25 | 1.67 ± 0.32 | 2.58 ± 0.22 | – |
| T1-C4 | 0.14 ± 0.19 | -1.46 ± 0.11 | -2.50 ± 0.16 | -2.29 ± 0.13 | -1.31 ± 0.26 | – |
| T2-C4 | 1.23 ± 0.19 | 0.07 ± 0.20 | -1.76 ± 0.19 | -2.91 ± 0.23 | -3.26 ± 0.19 | – |
| T3-C4 | 0.78 ± 0.29 | 2.08 ± 0.15 | 1.87 ± 0.18 | 1.25 ± 0.19 | 0.47 ± 0.15 | – |
| T4-C4 | -0.53 ± 0.17 | 0.94 ± 0.28 | 2.28 ± 0.23 | 2.37 ± 0.16 | 1.43 ± 0.18 | – |
| T5-C4 | -1.33 ± 0.14 | -1.23 ± 0.16 | 0.20 ± 0.32 | 1.23 ± 0.23 | 2.51 ± 0.32 | – |
| Avg ± SEM | 0.35 ± 0.01 | 0.38 ± 0.01 | 0.39 ± 0.02 | 0.37 ± 0.01 | 0.38 ± 0.01 | 0.37 ± 0.01 |

Table 56: Flappy $\mathcal{W}$ for PackNet.

| | T1 | T2 | T3 | T4 | T5 | Avg ± SEM |
|---|---|---|---|---|---|---|
| T1-C1 | 0.00 ± 0.00 | 0.00 ± 0.00 | 0.00 ± 0.00 | 0.00 ± 0.00 | 0.00 ± 0.00 | -1.01 ± 0.06 |
| T2-C1 | 0.34 ± 0.06 | 0.29 ± 0.06 | 0.19 ± 0.05 | 0.11 ± 0.03 | 0.19 ± 0.06 | -1.07 ± 0.03 |
| T3-C1 | 0.12 ± 0.15 | 0.63 ± 0.06 | 0.52 ± 0.06 | 0.40 ± 0.03 | 0.38 ± 0.04 | -0.07 ± 0.03 |
| T4-C1 | -1.05 ± 0.23 | 0.36 ± 0.06 | 0.50 ± 0.11 | 0.39 ± 0.11 | 0.33 ± 0.08 | 0.06 ± 0.02 |
| T5-C1 | -0.73 ± 0.15 | -0.23 ± 0.18 | -0.07 ± 0.10 | 0.31 ± 0.13 | 0.48 ± 0.15 | -0.27 ± 0.04 |
| T1-C2 | 0.24 ± 0.05 | -0.55 ± 0.10 | -1.37 ± 0.16 | -1.46 ± 0.29 | -1.34 ± 0.28 | – |
| T2-C2 | 0.55 ± 0.09 | -0.22 ± 0.10 | -1.17 ± 0.23 | -2.17 ± 0.21 | -3.33 ± 0.26 | – |
| T3-C2 | -0.07 ± 0.08 | 0.05 ± 0.08 | -0.22 ± 0.10 | -0.68 ± 0.09 | -0.76 ± 0.13 | – |
| T4-C2 | -0.74 ± 0.16 | 0.04 ± 0.11 | 0.30 ± 0.13 | 0.14 ± 0.04 | 0.24 ± 0.08 | – |
| T5-C2 | -0.87 ± 0.13 | -0.71 ± 0.17 | -0.26 ± 0.18 | 0.53 ± 0.07 | 0.38 ± 0.10 | – |
| T1-C3 | -0.46 ± 0.09 | -1.55 ± 0.17 | -2.04 ± 0.39 | -1.93 ± 0.26 | -1.70 ± 0.17 | – |
| T2-C3 | 0.10 ± 0.09 | -0.45 ± 0.08 | -1.37 ± 0.21 | -3.02 ± 0.20 | -3.12 ± 0.23 | – |
| T3-C3 | 0.26 ± 0.07 | 0.22 ± 0.06 | -0.10 ± 0.10 | -0.65 ± 0.12 | -0.60 ± 0.14 | – |
| T4-C3 | -0.99 ± 0.18 | 0.18 ± 0.12 | 0.37 ± 0.08 | 0.36 ± 0.07 | 0.28 ± 0.05 | – |
| T5-C3 | -1.37 ± 0.19 | -1.29 ± 0.14 | -0.25 ± 0.20 | 0.39 ± 0.08 | 0.40 ± 0.12 | – |
| T1-C4 | -0.44 ± 0.10 | -1.46 ± 0.11 | -2.50 ± 0.16 | -2.29 ± 0.13 | -1.31 ± 0.26 | – |
| T2-C4 | 0.06 ± 0.07 | -0.50 ± 0.13 | -1.76 ± 0.19 | -2.91 ± 0.23 | -3.26 ± 0.19 | – |
| T3-C4 | 0.18 ± 0.14 | 0.10 ± 0.08 | -0.22 ± 0.11 | -0.30 ± 0.06 | -0.70 ± 0.13 | – |
| T4-C4 | -0.56 ± 0.16 | 0.30 ± 0.17 | 0.53 ± 0.13 | 0.25 ± 0.05 | 0.08 ± 0.04 | – |
| T5-C4 | -1.35 ± 0.13 | -1.27 ± 0.13 | -0.20 ± 0.21 | 0.20 ± 0.11 | 0.47 ± 0.13 | – |
| Avg ± SEM | -0.34 ± 0.02 | -0.30 ± 0.02 | -0.46 ± 0.02 | -0.62 ± 0.03 | -0.65 ± 0.04 | -0.47 ± 0.01 |

Table 57: Catcher $\mathcal{F}$ for PackNet.

| | T1 | T2 | T3 | T4 | T5 | Avg ± SEM |
|---|---|---|---|---|---|---|
| T1-C1 | 6.80 ± 0.28 | 3.91 ± 0.37 | 1.15 ± 0.16 | 0.74 ± 0.08 | 0.27 ± 0.04 | -0.02 ± 0.05 |
| T2-C1 | -1.22 ± 0.43 | 2.17 ± 0.24 | 1.93 ± 0.29 | 1.17 ± 0.28 | 0.37 ± 0.10 | -0.23 ± 0.08 |
| T3-C1 | -1.58 ± 0.51 | 2.87 ± 0.20 | 3.46 ± 0.11 | 3.37 ± 0.13 | 1.02 ± 0.10 | 1.13 ± 0.06 |
| T4-C1 | -0.47 ± 0.65 | 0.60 ± 0.22 | 2.37 ± 0.38 | 3.35 ± 0.28 | 2.42 ± 0.24 | 1.01 ± 0.06 |
| T5-C1 | -0.52 ± 0.24 | -0.43 ± 0.30 | 0.69 ± 0.20 | 1.17 ± 0.08 | 3.84 ± 0.12 | -0.10 ± 0.07 |
| T1-C2 | 2.38 ± 0.53 | -0.69 ± 0.13 | -1.78 ± 0.37 | -2.32 ± 0.26 | -1.48 ± 0.22 | – |
| T2-C2 | 3.71 ± 0.41 | 0.57 ± 0.29 | -1.40 ± 0.31 | -3.02 ± 0.30 | -4.58 ± 0.13 | – |
| T3-C2 | 0.37 ± 0.19 | 0.71 ± 0.16 | 1.61 ± 0.34 | 1.59 ± 0.25 | -0.55 ± 0.16 | – |
| T4-C2 | -1.82 ± 0.27 | -1.19 ± 0.28 | 1.61 ± 0.29 | 3.10 ± 0.28 | 2.13 ± 0.15 | – |
| T5-C2 | -2.86 ± 0.18 | -2.84 ± 0.19 | -0.85 ± 0.17 | 0.73 ± 0.09 | 4.11 ± 0.17 | – |
| T1-C3 | 1.02 ± 0.28 | 0.19 ± 0.28 | -2.06 ± 0.30 | -2.78 ± 0.17 | -1.11 ± 0.15 | – |
| T2-C3 | 3.42 ± 0.12 | 3.04 ± 0.15 | -0.86 ± 0.21 | -3.45 ± 0.19 | -4.73 ± 0.08 | – |
| T3-C3 | 0.33 ± 0.13 | 0.75 ± 0.10 | 1.81 ± 0.33 | 2.07 ± 0.18 | -0.37 ± 0.11 | – |
| T4-C3 | -1.73 ± 0.24 | -1.49 ± 0.24 | 1.83 ± 0.21 | 3.54 ± 0.22 | 2.12 ± 0.23 | – |
| T5-C3 | -3.00 ± 0.18 | -2.99 ± 0.17 | -0.93 ± 0.24 | 0.57 ± 0.12 | 3.95 ± 0.11 | – |
| T1-C4 | 0.90 ± 0.25 | 0.54 ± 0.23 | -2.21 ± 0.20 | -2.83 ± 0.19 | -1.03 ± 0.24 | – |
| T2-C4 | 3.42 ± 0.16 | 3.23 ± 0.16 | -0.42 ± 0.30 | -3.21 ± 0.22 | -4.66 ± 0.08 | – |
| T3-C4 | 0.21 ± 0.14 | 0.59 ± 0.18 | 2.19 ± 0.20 | 2.28 ± 0.20 | -0.14 ± 0.13 | – |
| T4-C4 | -1.82 ± 0.25 | -1.51 ± 0.17 | 1.33 ± 0.22 | 3.33 ± 0.21 | 2.41 ± 0.15 | – |
| T5-C4 | -2.85 ± 0.22 | -2.73 ± 0.23 | -1.15 ± 0.16 | 0.47 ± 0.17 | 3.71 ± 0.16 | – |
| Avg ± SEM | 0.23 ± 0.01 | 0.26 ± 0.01 | 0.42 ± 0.01 | 0.49 ± 0.00 | 0.39 ± 0.01 | 0.36 ± 0.01 |

Table 58: Catcher $\mathcal{W}$ for PackNet.

| | T1 | T2 | T3 | T4 | T5 | Avg ± SEM |
|---|---|---|---|---|---|---|
| T1-C1 | 0.00 ± 0.00 | 0.00 ± 0.00 | 0.00 ± 0.00 | 0.00 ± 0.00 | 0.00 ± 0.00 | -1.08 ± 0.04 |
| T2-C1 | -1.33 ± 0.37 | 0.18 ± 0.17 | 0.14 ± 0.10 | 0.14 ± 0.09 | 0.05 ± 0.03 | -1.32 ± 0.06 |
| T3-C1 | -1.63 ± 0.48 | 0.96 ± 0.03 | 0.35 ± 0.06 | 0.36 ± 0.07 | 0.10 ± 0.02 | -0.09 ± 0.04 |
| T4-C1 | -1.00 ± 0.43 | 0.28 ± 0.13 | 0.72 ± 0.12 | 0.77 ± 0.06 | 0.15 ± 0.04 | -0.17 ± 0.03 |
| T5-C1 | -0.65 ± 0.20 | -0.53 ± 0.28 | 0.41 ± 0.16 | 0.62 ± 0.07 | 0.44 ± 0.10 | -0.85 ± 0.07 |
| T1-C2 | -0.25 ± 0.12 | -0.88 ± 0.06 | -1.78 ± 0.37 | -2.32 ± 0.26 | -1.48 ± 0.22 | – |
| T2-C2 | 0.69 ± 0.13 | 0.03 ± 0.06 | -1.46 ± 0.29 | -3.02 ± 0.30 | -4.58 ± 0.13 | – |
| T3-C2 | 0.36 ± 0.18 | 0.60 ± 0.13 | 0.16 ± 0.11 | -0.70 ± 0.08 | -1.12 ± 0.02 | – |
| T4-C2 | -1.82 ± 0.27 | -1.19 ± 0.28 | 0.65 ± 0.09 | 0.86 ± 0.03 | 0.26 ± 0.06 | – |
| T5-C2 | -2.86 ± 0.18 | -2.84 ± 0.19 | -0.85 ± 0.17 | 0.58 ± 0.04 | 0.49 ± 0.08 | – |
| T1-C3 | -0.70 ± 0.05 | -0.82 ± 0.04 | -2.06 ± 0.30 | -2.78 ± 0.17 | -1.11 ± 0.15 | – |
| T2-C3 | 0.39 ± 0.07 | 0.08 ± 0.04 | -1.26 ± 0.13 | -3.45 ± 0.19 | -4.73 ± 0.08 | – |
| T3-C3 | 0.31 ± 0.12 | 0.60 ± 0.07 | 0.05 ± 0.13 | -0.60 ± 0.07 | -1.09 ± 0.03 | – |
| T4-C3 | -1.73 ± 0.24 | -1.49 ± 0.24 | 0.77 ± 0.05 | 0.85 ± 0.03 | 0.19 ± 0.02 | – |
| T5-C3 | -3.01 ± 0.17 | -2.99 ± 0.17 | -0.93 ± 0.24 | 0.52 ± 0.11 | 0.55 ± 0.06 | – |
| T1-C4 | -0.61 ± 0.07 | -0.68 ± 0.05 | -2.21 ± 0.20 | -2.83 ± 0.19 | -1.03 ± 0.24 | – |
| T2-C4 | 0.41 ± 0.13 | 0.34 ± 0.10 | -1.05 ± 0.17 | -3.21 ± 0.22 | -4.66 ± 0.08 | – |
| T3-C4 | 0.20 ± 0.13 | 0.44 ± 0.15 | 0.34 ± 0.06 | -0.38 ± 0.11 | -1.04 ± 0.02 | – |
| T4-C4 | -1.82 ± 0.25 | -1.51 ± 0.17 | 0.63 ± 0.06 | 0.80 ± 0.07 | 0.17 ± 0.05 | – |
| T5-C4 | -2.85 ± 0.22 | -2.73 ± 0.23 | -1.15 ± 0.16 | 0.36 ± 0.13 | 0.39 ± 0.07 | – |
| Avg ± SEM | -0.90 ± 0.04 | -0.61 ± 0.04 | -0.43 ± 0.04 | -0.67 ± 0.04 | -0.90 ± 0.01 | -0.70 ± 0.03 |

Table 59: Room $\mathcal{F}$ for PM.

| | T1 | T2 | T3 | T4 | T5 | Avg ± SEM |
|---|---|---|---|---|---|---|
| T1-C1 | 0.58 ± 0.37 | -0.09 ± 0.24 | 0.89 ± 0.57 | 0.74 ± 0.41 | 0.04 ± 0.29 | 0.03 ± 0.18 |
| T2-C1 | 2.43 ± 0.19 | 1.55 ± 0.10 | 2.31 ± 0.23 | 2.42 ± 0.22 | 1.27 ± 0.10 | 0.56 ± 0.11 |
| T3-C1 | 3.04 ± 0.23 | 2.27 ± 0.18 | 3.20 ± 0.28 | 3.15 ± 0.26 | 2.23 ± 0.16 | 0.77 ± 0.07 |
| T4-C1 | 1.68 ± 0.19 | 1.27 ± 0.19 | 1.51 ± 0.18 | 1.75 ± 0.27 | 1.34 ± 0.17 | 0.48 ± 0.08 |
| T5-C1 | 1.06 ± 0.24 | 0.09 ± 0.35 | 0.86 ± 0.25 | 0.87 ± 0.30 | 0.06 ± 0.19 | 0.10 ± 0.09 |
| T1-C2 | -0.11 ± 0.19 | 0.28 ± 0.23 | 0.18 ± 0.20 | -0.08 ± 0.18 | 0.32 ± 0.30 | – |
| T2-C2 | 0.14 ± 0.18 | 0.84 ± 0.33 | 0.21 ± 0.18 | 0.23 ± 0.19 | 0.90 ± 0.23 | – |
| T3-C2 | 0.34 ± 0.22 | 0.22 ± 0.25 | 0.22 ± 0.22 | 0.21 ± 0.20 | 0.30 ± 0.31 | – |
| T4-C2 | 0.31 ± 0.12 | 0.07 ± 0.36 | 0.12 ± 0.16 | 0.25 ± 0.12 | -0.21 ± 0.31 | – |
| T5-C2 | 0.03 ± 0.22 | -0.34 ± 0.48 | 0.08 ± 0.21 | 0.03 ± 0.26 | -0.27 ± 0.41 | – |
| T1-C3 | -0.40 ± 0.25 | 0.10 ± 0.59 | -0.27 ± 0.27 | -0.39 ± 0.27 | 0.32 ± 0.51 | – |
| T2-C3 | -0.23 ± 0.23 | 0.39 ± 0.26 | -0.25 ± 0.26 | -0.19 ± 0.19 | 0.07 ± 0.28 | – |
| T3-C3 | 0.19 ± 0.12 | 0.28 ± 0.15 | 0.06 ± 0.09 | 0.19 ± 0.13 | 0.08 ± 0.19 | – |
| T4-C3 | 0.27 ± 0.25 | 0.22 ± 0.20 | 0.33 ± 0.30 | 0.26 ± 0.19 | 0.40 ± 0.16 | – |
| T5-C3 | -0.09 ± 0.24 | -0.31 ± 0.30 | 0.06 ± 0.24 | -0.03 ± 0.26 | -0.18 ± 0.32 | – |
| T1-C4 | -0.42 ± 0.40 | -0.28 ± 0.50 | -0.28 ± 0.45 | -0.25 ± 0.37 | -0.29 ± 0.57 | – |
| T2-C4 | -0.31 ± 0.38 | -0.08 ± 0.29 | -0.23 ± 0.33 | -0.34 ± 0.36 | 0.02 ± 0.30 | – |
| T3-C4 | -0.05 ± 0.14 | 0.03 ± 0.14 | -0.21 ± 0.18 | -0.32 ± 0.18 | 0.05 ± 0.14 | – |
| T4-C4 | -0.08 ± 0.07 | 0.12 ± 0.13 | -0.11 ± 0.06 | 0.05 ± 0.07 | 0.04 ± 0.13 | – |
| T5-C4 | -0.01 ± 0.03 | 0.01 ± 0.13 | -0.01 ± 0.03 | 0.04 ± 0.09 | -0.03 ± 0.16 | – |
| Avg ± SEM | 0.42 ± 0.06 | 0.33 ± 0.06 | 0.43 ± 0.06 | 0.43 ± 0.06 | 0.32 ± 0.07 | 0.39 ± 0.06 |

Table 60: Room $\mathcal{W}$ for PM.

| | T1 | T2 | T3 | T4 | T5 | Avg ± SEM |
|---|---|---|---|---|---|---|
| T1-C1 | -0.33 ± 0.23 | -0.54 ± 0.20 | -0.27 ± 0.25 | -0.22 ± 0.09 | -0.49 ± 0.17 | -0.39 ± 0.14 |
| T2-C1 | 0.32 ± 0.12 | 0.35 ± 0.11 | 0.32 ± 0.14 | 0.41 ± 0.14 | 0.15 ± 0.10 | -0.11 ± 0.09 |
| T3-C1 | 0.60 ± 0.08 | 0.40 ± 0.07 | 0.74 ± 0.10 | 0.66 ± 0.10 | 0.52 ± 0.04 | 0.03 ± 0.01 |
| T4-C1 | 0.54 ± 0.12 | 0.26 ± 0.09 | 0.46 ± 0.10 | 0.55 ± 0.08 | 0.44 ± 0.10 | 0.03 ± 0.03 |
| T5-C1 | 0.12 ± 0.12 | -0.33 ± 0.25 | 0.06 ± 0.11 | 0.12 ± 0.15 | -0.27 ± 0.15 | -0.24 ± 0.07 |
| T1-C2 | -0.25 ± 0.16 | -0.21 ± 0.11 | -0.10 ± 0.12 | -0.32 ± 0.09 | -0.30 ± 0.14 | – |
| T2-C2 | -0.14 ± 0.09 | 0.02 ± 0.22 | -0.14 ± 0.08 | -0.13 ± 0.12 | 0.09 ± 0.15 | – |
| T3-C2 | -0.13 ± 0.11 | -0.27 ± 0.12 | -0.20 ± 0.13 | -0.18 ± 0.12 | -0.17 ± 0.20 | – |
| T4-C2 | -0.04 ± 0.07 | -0.32 ± 0.21 | -0.15 ± 0.12 | -0.05 ± 0.08 | -0.52 ± 0.19 | – |
| T5-C2 | -0.17 ± 0.17 | -0.77 ± 0.38 | -0.15 ± 0.15 | -0.28 ± 0.19 | -0.67 ± 0.31 | – |
| T1-C3 | -0.42 ± 0.25 | -0.68 ± 0.44 | -0.38 ± 0.25 | -0.43 ± 0.25 | -0.33 ± 0.36 | – |
| T2-C3 | -0.35 ± 0.20 | -0.19 ± 0.16 | -0.39 ± 0.26 | -0.33 ± 0.18 | -0.30 ± 0.19 | – |
| T3-C3 | -0.04 ± 0.08 | -0.05 ± 0.10 | -0.13 ± 0.05 | 0.03 ± 0.07 | -0.26 ± 0.12 | – |
| T4-C3 | -0.02 ± 0.07 | -0.05 ± 0.15 | -0.02 ± 0.06 | 0.02 ± 0.06 | 0.02 ± 0.10 | – |
| T5-C3 | -0.24 ± 0.21 | -0.56 ± 0.26 | -0.11 ± 0.17 | -0.30 ± 0.18 | -0.44 ± 0.28 | – |
| T1-C4 | -0.50 ± 0.39 | -0.63 ± 0.45 | -0.45 ± 0.42 | -0.39 ± 0.35 | -0.64 ± 0.52 | – |
| T2-C4 | -0.46 ± 0.35 | -0.27 ± 0.23 | -0.30 ± 0.31 | -0.47 ± 0.34 | -0.30 ± 0.23 | – |
| T3-C4 | -0.12 ± 0.12 | -0.06 ± 0.13 | -0.24 ± 0.17 | -0.34 ± 0.18 | -0.15 ± 0.07 | – |
| T4-C4 | -0.10 ± 0.07 | -0.13 ± 0.08 | -0.12 ± 0.06 | -0.05 ± 0.04 | -0.11 ± 0.08 | – |
| T5-C4 | -0.11 ± 0.06 | -0.25 ± 0.07 | -0.10 ± 0.06 | -0.13 ± 0.05 | -0.31 ± 0.11 | – |
| Avg ± SEM | -0.09 ± 0.06 | -0.21 ± 0.05 | -0.08 ± 0.06 | -0.09 ± 0.06 | -0.20 ± 0.05 | -0.14 ± 0.06 |

Table 61: Flappy $\mathcal{F}$ for PM.

| | T1 | T2 | T3 | T4 | T5 | Avg ± SEM |
|---|---|---|---|---|---|---|
| T1-C1 | 3.68 ± 0.11 | 2.69 ± 0.08 | 2.21 ± 0.16 | 2.44 ± 0.19 | 1.97 ± 0.10 | 0.82 ± 0.05 |
| T2-C1 | 1.91 ± 0.15 | 1.77 ± 0.14 | 1.71 ± 0.16 | 1.03 ± 0.18 | 0.76 ± 0.09 | 0.41 ± 0.08 |
| T3-C1 | 2.39 ± 0.17 | 2.14 ± 0.17 | 1.95 ± 0.27 | 1.17 ± 0.27 | 1.14 ± 0.21 | 0.36 ± 0.12 |
| T4-C1 | 0.41 ± 0.14 | 0.43 ± 0.13 | 0.35 ± 0.30 | 0.23 ± 0.22 | 0.41 ± 0.14 | 0.03 ± 0.08 |
| T5-C1 | 0.33 ± 0.19 | 0.39 ± 0.18 | 0.60 ± 0.31 | 1.09 ± 0.41 | 0.66 ± 0.22 | 0.23 ± 0.09 |
| T1-C2 | 0.19 ± 0.11 | 0.15 ± 0.12 | 0.70 ± 0.43 | 1.36 ± 0.26 | 1.13 ± 0.21 | – |
| T2-C2 | 0.04 ± 0.16 | 0.18 ± 0.17 | 0.45 ± 0.25 | 0.79 ± 0.26 | 1.01 ± 0.18 | – |
| T3-C2 | -0.07 ± 0.17 | 0.08 ± 0.21 | -0.06 ± 0.41 | 0.02 ± 0.26 | 0.31 ± 0.18 | – |
| T4-C2 | -0.39 ± 0.20 | -0.17 ± 0.20 | -0.41 ± 0.18 | -0.07 ± 0.30 | 0.31 ± 0.40 | – |
| T5-C2 | -0.22 ± 0.24 | -0.08 ± 0.18 | 0.37 ± 0.35 | -0.22 ± 0.15 | 0.77 ± 0.28 | – |
| T1-C3 | -0.17 ± 0.26 | -0.37 ± 0.15 | 0.33 ± 0.23 | -0.25 ± 0.44 | -0.23 ± 0.26 | – |
| T2-C3 | -0.01 ± 0.09 | -0.07 ± 0.14 | -0.59 ± 0.42 | 0.37 ± 0.29 | -0.51 ± 0.30 | – |
| T3-C3 | 0.05 ± 0.29 | -0.05 ± 0.18 | -0.42 ± 0.22 | -0.18 ± 0.30 | -0.16 ± 0.32 | – |
| T4-C3 | -0.26 ± 0.20 | -0.22 ± 0.22 | 0.40 ± 0.26 | -0.23 ± 0.33 | -0.08 ± 0.28 | – |
| T5-C3 | -0.12 ± 0.26 | 0.20 ± 0.23 | 0.63 ± 0.20 | -0.27 ± 0.19 | 0.34 ± 0.32 | – |
| T1-C4 | 0.22 ± 0.19 | 0.04 ± 0.18 | -0.12 ± 0.32 | 0.23 ± 0.34 | 0.15 ± 0.18 | – |
| T2-C4 | 0.07 ± 0.21 | -0.08 ± 0.22 | -0.72 ± 0.29 | 0.32 ± 0.19 | -0.19 ± 0.37 | – |
| T3-C4 | -0.24 ± 0.28 | -0.18 ± 0.24 | -0.39 ± 0.41 | -0.16 ± 0.43 | -0.09 ± 0.23 | – |
| T4-C4 | -0.33 ± 0.25 | -0.09 ± 0.24 | 0.35 ± 0.38 | 0.19 ± 0.27 | -0.24 ± 0.32 | – |
| T5-C4 | 0.16 ± 0.29 | 0.04 ± 0.27 | 0.01 ± 0.42 | -0.11 ± 0.30 | 0.07 ± 0.18 | – |
| Avg ± SEM | 0.38 ± 0.01 | 0.34 ± 0.01 | 0.37 ± 0.01 | 0.39 ± 0.02 | 0.38 ± 0.01 | 0.37 ± 0.01 |

Table 62: Flappy $\mathcal{W}$ for PM.

| | T1 | T2 | T3 | T4 | T5 | Avg ± SEM |
|---|---|---|---|---|---|---|
| T1-C1 | 0.00 ± 0.00 | 0.00 ± 0.00 | 0.00 ± 0.00 | 0.00 ± 0.00 | 0.00 ± 0.00 | -0.21 ± 0.02 |
| T2-C1 | 0.57 ± 0.08 | 0.49 ± 0.08 | 0.53 ± 0.08 | 0.25 ± 0.07 | 0.17 ± 0.03 | -0.18 ± 0.05 |
| T3-C1 | 0.86 ± 0.08 | 0.54 ± 0.06 | 0.45 ± 0.05 | 0.46 ± 0.08 | 0.44 ± 0.04 | -0.24 ± 0.07 |
| T4-C1 | -0.05 ± 0.10 | -0.01 ± 0.10 | -0.21 ± 0.13 | -0.26 ± 0.14 | -0.08 ± 0.07 | -0.36 ± 0.04 |
| T5-C1 | -0.10 ± 0.11 | 0.16 ± 0.13 | 0.08 ± 0.17 | 0.07 ± 0.28 | -0.06 ± 0.12 | -0.24 ± 0.06 |
| T1-C2 | -0.09 ± 0.09 | -0.19 ± 0.10 | -0.19 ± 0.24 | 0.30 ± 0.09 | 0.37 ± 0.12 | – |
| T2-C2 | -0.25 ± 0.10 | -0.33 ± 0.13 | -0.30 ± 0.19 | 0.03 ± 0.13 | 0.16 ± 0.10 | – |
| T3-C2 | -0.32 ± 0.13 | -0.32 ± 0.15 | -0.63 ± 0.32 | -0.47 ± 0.16 | -0.30 ± 0.07 | – |
| T4-C2 | -0.62 ± 0.16 | -0.51 ± 0.11 | -0.72 ± 0.13 | -0.47 ± 0.24 | -0.51 ± 0.24 | – |
| T5-C2 | -0.38 ± 0.20 | -0.34 ± 0.12 | -0.23 ± 0.21 | -0.44 ± 0.10 | -0.21 ± 0.12 | – |
| T1-C3 | -0.41 ± 0.18 | -0.54 ± 0.12 | 0.00 ± 0.21 | -0.78 ± 0.28 | -0.40 ± 0.22 | – |
| T2-C3 | -0.28 ± 0.07 | -0.25 ± 0.12 | -1.02 ± 0.29 | -0.12 ± 0.20 | -0.78 ± 0.25 | – |
| T3-C3 | -0.32 ± 0.22 | -0.33 ± 0.16 | -0.80 ± 0.17 | -0.59 ± 0.20 | -0.58 ± 0.19 | – |
| T4-C3 | -0.47 ± 0.15 | -0.44 ± 0.19 | -0.08 ± 0.16 | -0.46 ± 0.30 | -0.44 ± 0.22 | – |
| T5-C3 | -0.44 ± 0.18 | -0.14 ± 0.13 | -0.03 ± 0.14 | -0.58 ± 0.11 | -0.27 ± 0.21 | – |
| T1-C4 | -0.22 ± 0.14 | -0.23 ± 0.16 | -0.63 ± 0.20 | -0.68 ± 0.18 | -0.48 ± 0.17 | – |
| T2-C4 | -0.28 ± 0.11 | -0.35 ± 0.18 | -1.02 ± 0.27 | -0.29 ± 0.15 | -0.56 ± 0.27 | – |
| T3-C4 | -0.49 ± 0.24 | -0.43 ± 0.19 | -0.76 ± 0.30 | -0.62 ± 0.32 | -0.62 ± 0.18 | – |
| T4-C4 | -0.49 ± 0.22 | -0.36 ± 0.18 | -0.15 ± 0.23 | -0.34 ± 0.16 | -0.52 ± 0.20 | – |
| T5-C4 | -0.18 ± 0.18 | -0.38 ± 0.20 | -0.59 ± 0.31 | -0.50 ± 0.25 | -0.22 ± 0.13 | – |
| Avg ± SEM | -0.20 ± 0.02 | -0.20 ± 0.02 | -0.32 ± 0.02 | -0.27 ± 0.02 | -0.25 ± 0.02 | -0.25 ± 0.01 |

Table 63: Catcher $\mathcal{F}$ for PM.

| | T1 | T2 | T3 | T4 | T5 | Avg ± SEM |
|---|---|---|---|---|---|---|
| T1-C1 | 7.12 ± 0.26 | 4.47 ± 0.29 | 0.60 ± 0.06 | 0.30 ± 0.03 | 0.13 ± 0.01 | 0.85 ± 0.06 |
| T2-C1 | 1.31 ± 0.12 | 1.60 ± 0.19 | 0.42 ± 0.11 | 0.19 ± 0.06 | 0.04 ± 0.01 | 0.31 ± 0.05 |
| T3-C1 | 1.41 ± 0.13 | 2.48 ± 0.14 | 2.34 ± 0.15 | 1.08 ± 0.09 | 0.21 ± 0.02 | 0.40 ± 0.03 |
| T4-C1 | 0.13 ± 0.05 | 1.13 ± 0.14 | 4.01 ± 0.13 | 3.62 ± 0.19 | 0.47 ± 0.07 | 0.43 ± 0.04 |
| T5-C1 | -0.09 ± 0.05 | 0.20 ± 0.10 | 2.17 ± 0.19 | 3.61 ± 0.13 | 1.89 ± 0.15 | 0.35 ± 0.06 |
| T1-C2 | -0.12 ± 0.05 | -0.11 ± 0.07 | 0.16 ± 0.09 | 0.87 ± 0.17 | 3.67 ± 0.16 | – |
| T2-C2 | 0.07 ± 0.05 | -0.02 ± 0.05 | 0.09 ± 0.09 | 0.11 ± 0.07 | 2.45 ± 0.22 | – |
| T3-C2 | -0.13 ± 0.13 | -0.09 ± 0.15 | -0.13 ± 0.15 | -0.08 ± 0.14 | 0.28 ± 0.20 | – |
| T4-C2 | -0.09 ± 0.07 | -0.10 ± 0.09 | -0.05 ± 0.05 | -0.05 ± 0.06 | 0.08 ± 0.09 | – |
| T5-C2 | 0.16 ± 0.15 | 0.00 ± 0.20 | 0.08 ± 0.18 | 0.12 ± 0.18 | 0.07 ± 0.23 | – |
| T1-C3 | 0.03 ± 0.06 | -0.09 ± 0.14 | -0.06 ± 0.06 | 0.02 ± 0.07 | -0.04 ± 0.14 | – |
| T2-C3 | -0.16 ± 0.09 | -0.04 ± 0.15 | -0.10 ± 0.17 | -0.11 ± 0.16 | -0.17 ± 0.21 | – |
| T3-C3 | 0.02 ± 0.06 | 0.18 ± 0.12 | 0.08 ± 0.10 | 0.00 ± 0.05 | 0.03 ± 0.15 | – |
| T4-C3 | 0.08 ± 0.07 | -0.06 ± 0.11 | -0.07 ± 0.12 | 0.02 ± 0.16 | 0.06 ± 0.19 | – |
| T5-C3 | -0.03 ± 0.07 | -0.10 ± 0.09 | -0.09 ± 0.06 | -0.13 ± 0.10 | -0.18 ± 0.18 | – |
| T1-C4 | 0.04 ± 0.12 | 0.02 ± 0.21 | 0.03 ± 0.21 | -0.08 ± 0.22 | 0.02 ± 0.19 | – |
| T2-C4 | 0.07 ± 0.06 | 0.05 ± 0.06 | 0.14 ± 0.04 | 0.07 ± 0.09 | 0.22 ± 0.09 | – |
| T3-C4 | -0.01 ± 0.06 | 0.13 ± 0.13 | 0.05 ± 0.19 | 0.08 ± 0.16 | 0.08 ± 0.11 | – |
| T4-C4 | -0.02 ± 0.05 | -0.09 ± 0.11 | -0.28 ± 0.07 | -0.11 ± 0.05 | -0.20 ± 0.11 | – |
| T5-C4 | -0.11 ± 0.10 | -0.20 ± 0.13 | -0.11 ± 0.16 | -0.07 ± 0.16 | -0.18 ± 0.20 | – |
| Avg ± SEM | 0.48 ± 0.00 | 0.47 ± 0.01 | 0.46 ± 0.01 | 0.47 ± 0.01 | 0.45 ± 0.01 | 0.47 ± 0.01 |

Table 64: Catcher $\mathcal{W}$ for PM.

| | T1 | T2 | T3 | T4 | T5 | Avg ± SEM |
|---|---|---|---|---|---|---|
| T1-C1 | 0.00 ± 0.00 | 0.00 ± 0.00 | 0.00 ± 0.00 | 0.00 ± 0.00 | 0.00 ± 0.00 | -0.05 ± 0.04 |
| T2-C1 | 0.45 ± 0.04 | 0.23 ± 0.18 | 0.03 ± 0.02 | 0.01 ± 0.02 | 0.00 ± 0.00 | -0.00 ± 0.04 |
| T3-C1 | 0.97 ± 0.02 | 0.88 ± 0.05 | 0.18 ± 0.03 | 0.07 ± 0.01 | 0.02 ± 0.01 | 0.01 ± 0.03 |
| T4-C1 | 0.00 ± 0.01 | 0.37 ± 0.09 | 0.81 ± 0.05 | 0.60 ± 0.09 | 0.09 ± 0.03 | -0.02 ± 0.03 |
| T5-C1 | -0.11 ± 0.05 | 0.05 ± 0.06 | 0.81 ± 0.05 | 0.90 ± 0.01 | 0.19 ± 0.04 | -0.06 ± 0.03 |
| T1-C2 | -0.12 ± 0.05 | -0.14 ± 0.06 | 0.02 ± 0.06 | 0.32 ± 0.09 | 0.67 ± 0.08 | – |
| T2-C2 | -0.05 ± 0.02 | -0.08 ± 0.03 | -0.07 ± 0.02 | -0.01 ± 0.03 | 0.71 ± 0.06 | – |
| T3-C2 | -0.20 ± 0.11 | -0.18 ± 0.13 | -0.17 ± 0.14 | -0.19 ± 0.13 | -0.12 ± 0.10 | – |
| T4-C2 | -0.12 ± 0.06 | -0.15 ± 0.08 | -0.12 ± 0.04 | -0.09 ± 0.04 | -0.14 ± 0.09 | – |
| T5-C2 | -0.05 ± 0.04 | -0.23 ± 0.09 | -0.12 ± 0.08 | -0.10 ± 0.06 | -0.14 ± 0.17 | – |
| T1-C3 | -0.08 ± 0.06 | -0.26 ± 0.10 | -0.11 ± 0.06 | -0.07 ± 0.04 | -0.19 ± 0.10 | – |
| T2-C3 | -0.20 ± 0.08 | -0.20 ± 0.11 | -0.23 ± 0.13 | -0.25 ± 0.13 | -0.31 ± 0.17 | – |
| T3-C3 | -0.00 ± 0.05 | 0.02 ± 0.06 | -0.05 ± 0.07 | -0.06 ± 0.04 | -0.20 ± 0.10 | – |
| T4-C3 | -0.00 ± 0.03 | -0.17 ± 0.07 | -0.13 ± 0.10 | -0.09 ± 0.11 | -0.12 ± 0.13 | – |
| T5-C3 | -0.10 ± 0.06 | -0.16 ± 0.08 | -0.18 ± 0.06 | -0.14 ± 0.10 | -0.32 ± 0.14 | – |
| T1-C4 | -0.10 ± 0.10 | -0.19 ± 0.16 | -0.16 ± 0.17 | -0.25 ± 0.17 | -0.27 ± 0.13 | – |
| T2-C4 | -0.01 ± 0.04 | -0.06 ± 0.04 | 0.01 ± 0.02 | -0.05 ± 0.02 | -0.00 ± 0.06 | – |
| T3-C4 | -0.10 ± 0.03 | -0.09 ± 0.05 | -0.18 ± 0.09 | -0.15 ± 0.05 | -0.15 ± 0.05 | – |
| T4-C4 | -0.07 ± 0.04 | -0.18 ± 0.07 | -0.33 ± 0.07 | -0.20 ± 0.07 | -0.32 ± 0.11 | – |
| T5-C4 | -0.17 ± 0.09 | -0.29 ± 0.10 | -0.30 ± 0.09 | -0.24 ± 0.12 | -0.48 ± 0.11 | – |
| Avg ± SEM | -0.00 ± 0.02 | -0.04 ± 0.02 | -0.01 ± 0.01 | 0.00 ± 0.02 | -0.05 ± 0.01 | -0.02 ± 0.01 |

Table 65: Room $\mathcal{F}$ for Qreg.

| | T1 | T2 | T3 | T4 | T5 | Avg ± SEM |
|---|---|---|---|---|---|---|
| T1-C1 | 0.19 ± 0.28 | -0.30 ± 0.26 | -0.03 ± 0.34 | 0.08 ± 0.24 | -0.13 ± 0.16 | 0.14 ± 0.05 |
| T2-C1 | 0.69 ± 0.44 | 1.54 ± 0.35 | 0.77 ± 0.38 | 0.95 ± 0.45 | 1.57 ± 0.44 | 0.40 ± 0.08 |
| T3-C1 | 1.76 ± 0.54 | 3.66 ± 0.46 | 1.93 ± 0.56 | 1.82 ± 0.55 | 3.46 ± 0.50 | 0.80 ± 0.08 |
| T4-C1 | 1.84 ± 0.56 | 1.87 ± 0.27 | 2.19 ± 0.58 | 1.84 ± 0.56 | 1.67 ± 0.33 | 0.63 ± 0.11 |
| T5-C1 | 0.63 ± 0.25 | 0.46 ± 0.18 | 1.02 ± 0.32 | 0.83 ± 0.32 | 0.77 ± 0.24 | 0.23 ± 0.04 |
| T1-C2 | 0.24 ± 0.14 | 0.96 ± 0.25 | 0.13 ± 0.12 | 0.12 ± 0.06 | 1.12 ± 0.27 | – |
| T2-C2 | 0.16 ± 0.15 | 0.68 ± 0.25 | 0.09 ± 0.12 | 0.19 ± 0.11 | 0.59 ± 0.26 | – |
| T3-C2 | 0.26 ± 0.17 | 0.42 ± 0.24 | 0.22 ± 0.14 | 0.24 ± 0.15 | 0.39 ± 0.22 | – |
| T4-C2 | 0.69 ± 0.24 | -0.09 ± 0.11 | 0.54 ± 0.23 | 0.60 ± 0.23 | -0.13 ± 0.15 | – |
| T5-C2 | -0.07 ± 0.10 | 0.20 ± 0.13 | 0.23 ± 0.07 | 0.06 ± 0.14 | 0.25 ± 0.11 | – |
| T1-C3 | -0.03 ± 0.06 | 0.15 ± 0.11 | -0.03 ± 0.09 | 0.02 ± 0.06 | 0.26 ± 0.12 | – |
| T2-C3 | 0.34 ± 0.10 | 0.05 ± 0.05 | 0.08 ± 0.07 | 0.06 ± 0.16 | -0.02 ± 0.06 | – |
| T3-C3 | 0.37 ± 0.12 | 0.18 ± 0.14 | 0.34 ± 0.17 | 0.11 ± 0.15 | 0.06 ± 0.14 | – |
| T4-C3 | 0.19 ± 0.22 | -0.03 ± 0.06 | 0.39 ± 0.20 | 0.46 ± 0.21 | 0.06 ± 0.07 | – |
| T5-C3 | 0.06 ± 0.07 | 0.02 ± 0.06 | -0.06 ± 0.06 | 0.17 ± 0.17 | -0.10 ± 0.10 | – |
| T1-C4 | 0.05 ± 0.09 | -0.00 ± 0.05 | -0.15 ± 0.15 | 0.04 ± 0.11 | 0.04 ± 0.06 | – |
| T2-C4 | -0.05 ± 0.11 | -0.01 ± 0.04 | 0.03 ± 0.05 | 0.10 ± 0.16 | 0.17 ± 0.06 | – |
| T3-C4 | 0.23 ± 0.13 | 0.08 ± 0.09 | 0.10 ± 0.08 | 0.23 ± 0.15 | 0.05 ± 0.05 | – |
| T4-C4 | 0.26 ± 0.10 | 0.02 ± 0.07 | 0.26 ± 0.12 | 0.00 ± 0.05 | -0.01 ± 0.04 | – |
| T5-C4 | 0.03 ± 0.09 | 0.04 ± 0.04 | 0.07 ± 0.07 | 0.04 ± 0.05 | 0.01 ± 0.03 | – |
| Avg ± SEM | 0.39 ± 0.05 | 0.50 ± 0.01 | 0.41 ± 0.05 | 0.40 ± 0.06 | 0.51 ± 0.01 | 0.44 ± 0.03 |

Table 66: Room $\mathcal{W}$ for Qreg.

| | T1 | T2 | T3 | T4 | T5 | Avg ± SEM |
|---|---|---|---|---|---|---|
| T1-C1 | -0.27 ± 0.16 | -0.53 ± 0.18 | -0.40 ± 0.25 | -0.28 ± 0.10 | -0.37 ± 0.11 | -0.13 ± 0.03 |
| T2-C1 | 0.18 ± 0.17 | 0.10 ± 0.05 | 0.19 ± 0.17 | 0.25 ± 0.16 | 0.14 ± 0.09 | 0.01 ± 0.03 |
| T3-C1 | 0.09 ± 0.10 | 0.66 ± 0.09 | 0.19 ± 0.12 | 0.23 ± 0.11 | 0.66 ± 0.11 | 0.07 ± 0.03 |
| T4-C1 | 0.28 ± 0.10 | 0.58 ± 0.10 | 0.47 ± 0.14 | 0.34 ± 0.10 | 0.53 ± 0.14 | 0.07 ± 0.04 |
| T5-C1 | 0.39 ± 0.15 | -0.13 ± 0.06 | 0.41 ± 0.10 | 0.29 ± 0.14 | -0.03 ± 0.10 | 0.01 ± 0.02 |
| T1-C2 | -0.02 ± 0.06 | 0.26 ± 0.09 | -0.12 ± 0.07 | -0.16 ± 0.05 | 0.24 ± 0.11 | – |
| T2-C2 | -0.08 ± 0.09 | 0.15 ± 0.08 | -0.09 ± 0.08 | -0.05 ± 0.04 | 0.19 ± 0.08 | – |
| T3-C2 | 0.02 ± 0.08 | 0.01 ± 0.05 | -0.09 ± 0.06 | -0.05 ± 0.04 | 0.00 ± 0.09 | – |
| T4-C2 | 0.00 ± 0.06 | -0.15 ± 0.10 | 0.02 ± 0.05 | 0.03 ± 0.04 | -0.22 ± 0.13 | – |
| T5-C2 | -0.15 ± 0.07 | -0.08 ± 0.05 | 0.13 ± 0.06 | -0.01 ± 0.13 | -0.10 ± 0.05 | – |
| T1-C3 | -0.17 ± 0.10 | -0.01 ± 0.06 | -0.16 ± 0.09 | -0.13 ± 0.06 | 0.06 ± 0.07 | – |
| T2-C3 | 0.02 ± 0.02 | -0.02 ± 0.04 | -0.05 ± 0.07 | -0.20 ± 0.10 | -0.07 ± 0.06 | – |
| T3-C3 | 0.02 ± 0.03 | -0.02 ± 0.03 | 0.01 ± 0.05 | -0.11 ± 0.10 | -0.08 ± 0.04 | – |
| T4-C3 | -0.18 ± 0.11 | -0.09 ± 0.04 | 0.05 ± 0.07 | 0.05 ± 0.02 | -0.06 ± 0.04 | – |
| T5-C3 | -0.01 ± 0.05 | -0.06 ± 0.03 | -0.13 ± 0.05 | 0.00 ± 0.07 | -0.20 ± 0.08 | – |
| T1-C4 | -0.08 ± 0.07 | -0.06 ± 0.04 | -0.22 ± 0.14 | -0.14 ± 0.08 | -0.05 ± 0.03 | – |
| T2-C4 | -0.15 ± 0.08 | -0.08 ± 0.04 | -0.11 ± 0.05 | -0.05 ± 0.09 | 0.02 ± 0.03 | – |
| T3-C4 | 0.05 ± 0.05 | -0.02 ± 0.05 | -0.09 ± 0.08 | -0.04 ± 0.07 | -0.04 ± 0.04 | – |
| T4-C4 | -0.05 ± 0.04 | -0.04 ± 0.05 | -0.01 ± 0.04 | -0.14 ± 0.05 | -0.06 ± 0.03 | – |
| T5-C4 | -0.11 ± 0.05 | -0.03 ± 0.03 | 0.01 ± 0.06 | -0.02 ± 0.04 | -0.06 ± 0.02 | – |
| Avg ± SEM | -0.01 ± 0.02 | 0.02 ± 0.01 | 0.00 ± 0.03 | -0.01 ± 0.03 | 0.02 ± 0.01 | 0.00 ± 0.02 |

Table 67: Flappy $\mathcal{F}$ for Qreg.

| | T1 | T2 | T3 | T4 | T5 | Avg ± SEM |
|---|---|---|---|---|---|---|
| T1-C1 | 2.71 ± 0.19 | 1.79 ± 0.12 | 1.11 ± 0.07 | 1.03 ± 0.08 | 1.21 ± 0.09 | 0.58 ± 0.05 |
| T2-C1 | 2.05 ± 0.16 | 2.13 ± 0.15 | 1.11 ± 0.09 | 0.58 ± 0.09 | 0.43 ± 0.10 | 0.41 ± 0.05 |
| T3-C1 | 1.85 ± 0.17 | 2.56 ± 0.13 | 2.00 ± 0.13 | 1.30 ± 0.11 | 0.78 ± 0.14 | 0.48 ± 0.05 |
| T4-C1 | -0.66 ± 0.14 | 0.37 ± 0.14 | 1.42 ± 0.16 | 1.89 ± 0.19 | 1.35 ± 0.25 | 0.25 ± 0.06 |
| T5-C1 | -0.79 ± 0.16 | -0.19 ± 0.15 | 0.58 ± 0.25 | 1.53 ± 0.25 | 2.38 ± 0.29 | 0.23 ± 0.06 |
| T1-C2 | 0.29 ± 0.17 | 0.23 ± 0.20 | 0.13 ± 0.27 | 0.25 ± 0.22 | 1.14 ± 0.26 | – |
| T2-C2 | 1.04 ± 0.17 | 0.63 ± 0.26 | 0.17 ± 0.23 | -0.42 ± 0.23 | -0.41 ± 0.30 | – |
| T3-C2 | 0.32 ± 0.17 | 0.59 ± 0.21 | 0.28 ± 0.20 | -0.17 ± 0.19 | -0.32 ± 0.25 | – |
| T4-C2 | -0.34 ± 0.21 | -0.17 ± 0.21 | 0.31 ± 0.31 | 0.27 ± 0.22 | -0.03 ± 0.29 | – |
| T5-C2 | -0.23 ± 0.11 | -0.37 ± 0.22 | 0.31 ± 0.14 | 0.37 ± 0.17 | 0.51 ± 0.31 | – |
| T1-C3 | 0.08 ± 0.15 | 0.19 ± 0.14 | 0.27 ± 0.23 | 0.20 ± 0.25 | 0.25 ± 0.33 | – |
| T2-C3 | 0.61 ± 0.19 | -0.03 ± 0.19 | -0.07 ± 0.19 | -0.22 ± 0.14 | -0.11 ± 0.33 | – |
| T3-C3 | 0.39 ± 0.11 | 0.30 ± 0.27 | -0.15 ± 0.28 | 0.03 ± 0.31 | -0.01 ± 0.35 | – |
| T4-C3 | -0.34 ± 0.18 | 0.34 ± 0.20 | -0.29 ± 0.24 | 0.30 ± 0.27 | -0.06 ± 0.32 | – |
| T5-C3 | -0.19 ± 0.18 | -0.07 ± 0.27 | -0.07 ± 0.31 | 0.09 ± 0.21 | 0.14 ± 0.33 | – |
| T1-C4 | 0.16 ± 0.18 | 0.06 ± 0.25 | 0.28 ± 0.29 | 0.05 ± 0.36 | 0.21 ± 0.36 | – |
| T2-C4 | 0.22 ± 0.19 | -0.08 ± 0.25 | 0.15 ± 0.24 | 0.10 ± 0.31 | 0.25 ± 0.32 | – |
| T3-C4 | 0.21 ± 0.22 | -0.01 ± 0.27 | -0.10 ± 0.26 | -0.16 ± 0.33 | -0.16 ± 0.28 | – |
| T4-C4 | 0.33 ± 0.27 | 0.14 ± 0.32 | 0.19 ± 0.27 | -0.18 ± 0.41 | 0.20 ± 0.34 | – |
| T5-C4 | -0.40 ± 0.15 | -0.25 ± 0.18 | 0.19 ± 0.27 | 0.34 ± 0.26 | 0.73 ± 0.22 | – |
| Avg ± SEM | 0.37 ± 0.02 | 0.41 ± 0.01 | 0.39 ± 0.01 | 0.36 ± 0.01 | 0.42 ± 0.02 | 0.39 ± 0.01 |

Table 68: Flappy $\mathcal{W}$ for Qreg.

| | T1 | T2 | T3 | T4 | T5 | Avg ± SEM |
|---|---|---|---|---|---|---|
| T1-C1 | -0.00 ± 0.00 | 0.00 ± 0.00 | 0.00 ± 0.00 | 0.00 ± 0.00 | -0.00 ± 0.00 | -0.15 ± 0.03 |
| T2-C1 | 0.58 ± 0.10 | 0.45 ± 0.10 | 0.27 ± 0.07 | 0.13 ± 0.03 | 0.15 ± 0.06 | -0.16 ± 0.04 |
| T3-C1 | 0.71 ± 0.07 | 0.72 ± 0.02 | 0.40 ± 0.05 | 0.21 ± 0.03 | 0.14 ± 0.02 | -0.12 ± 0.04 |
| T4-C1 | -0.69 ± 0.14 | 0.05 ± 0.09 | 0.30 ± 0.08 | 0.28 ± 0.05 | 0.10 ± 0.07 | -0.25 ± 0.04 |
| T5-C1 | -0.83 ± 0.15 | -0.43 ± 0.10 | -0.05 ± 0.13 | 0.47 ± 0.07 | 0.60 ± 0.07 | -0.23 ± 0.03 |
| T1-C2 | -0.21 ± 0.04 | -0.22 ± 0.12 | -0.32 ± 0.19 | -0.02 ± 0.15 | 0.32 ± 0.15 | – |
| T2-C2 | 0.12 ± 0.04 | -0.03 ± 0.10 | -0.32 ± 0.12 | -0.65 ± 0.16 | -0.73 ± 0.20 | – |
| T3-C2 | -0.01 ± 0.12 | 0.08 ± 0.15 | -0.22 ± 0.17 | -0.35 ± 0.16 | -0.65 ± 0.16 | – |
| T4-C2 | -0.47 ± 0.18 | -0.45 ± 0.14 | -0.24 ± 0.18 | -0.11 ± 0.17 | -0.55 ± 0.20 | – |
| T5-C2 | -0.41 ± 0.07 | -0.56 ± 0.15 | -0.12 ± 0.18 | -0.06 ± 0.12 | 0.10 ± 0.08 | – |
| T1-C3 | -0.25 ± 0.12 | -0.13 ± 0.09 | -0.09 ± 0.13 | -0.32 ± 0.17 | -0.17 ± 0.22 | – |
| T2-C3 | -0.08 ± 0.08 | -0.22 ± 0.11 | -0.53 ± 0.14 | -0.44 ± 0.16 | -0.58 ± 0.28 | – |
| T3-C3 | 0.13 ± 0.08 | -0.16 ± 0.14 | -0.47 ± 0.25 | -0.50 ± 0.15 | -0.43 ± 0.23 | – |
| T4-C3 | -0.40 ± 0.15 | -0.08 ± 0.12 | -0.49 ± 0.19 | -0.31 ± 0.12 | -0.35 ± 0.21 | – |
| T5-C3 | -0.37 ± 0.14 | -0.41 ± 0.14 | -0.42 ± 0.19 | -0.34 ± 0.18 | -0.29 ± 0.21 | – |
| T1-C4 | -0.27 ± 0.09 | -0.24 ± 0.20 | -0.23 ± 0.19 | -0.35 ± 0.22 | -0.46 ± 0.23 | – |
| T2-C4 | -0.15 ± 0.11 | -0.31 ± 0.18 | -0.41 ± 0.15 | -0.28 ± 0.15 | -0.28 ± 0.19 | – |
| T3-C4 | -0.23 ± 0.10 | -0.38 ± 0.17 | -0.43 ± 0.20 | -0.58 ± 0.24 | -0.37 ± 0.24 | – |
| T4-C4 | -0.07 ± 0.16 | -0.36 ± 0.22 | -0.27 ± 0.15 | -0.56 ± 0.31 | -0.39 ± 0.20 | – |
| T5-C4 | -0.51 ± 0.14 | -0.53 ± 0.13 | -0.26 ± 0.16 | -0.14 ± 0.16 | -0.04 ± 0.13 | – |
| Avg ± SEM | -0.17 ± 0.01 | -0.16 ± 0.01 | -0.20 ± 0.02 | -0.20 ± 0.02 | -0.19 ± 0.02 | -0.18 ± 0.01 |

Table 69: Catcher $\mathcal{F}$ for Qreg.

| | T1 | T2 | T3 | T4 | T5 | Avg ± SEM |
|---|---|---|---|---|---|---|
| T1-C1 | 7.06 ± 0.32 | 4.37 ± 0.47 | 0.70 ± 0.10 | 0.28 ± 0.04 | 0.11 ± 0.02 | 0.84 ± 0.04 |
| T2-C1 | 1.37 ± 0.15 | 2.37 ± 0.25 | 1.06 ± 0.30 | 0.33 ± 0.12 | 0.04 ± 0.02 | 0.38 ± 0.02 |
| T3-C1 | 0.86 ± 0.19 | 2.64 ± 0.25 | 3.85 ± 0.08 | 3.17 ± 0.20 | 0.33 ± 0.06 | 0.57 ± 0.02 |
| T4-C1 | -1.28 ± 0.21 | 0.34 ± 0.10 | 3.58 ± 0.31 | 4.37 ± 0.16 | 1.66 ± 0.29 | 0.44 ± 0.03 |
| T5-C1 | -1.18 ± 0.26 | 0.07 ± 0.10 | 0.78 ± 0.10 | 1.69 ± 0.21 | 3.91 ± 0.07 | 0.28 ± 0.02 |
| T1-C2 | 0.92 ± 0.18 | -0.05 ± 0.08 | -0.05 ± 0.02 | 0.14 ± 0.05 | 3.16 ± 0.31 | – |
| T2-C2 | 1.78 ± 0.25 | 0.05 ± 0.07 | -0.12 ± 0.08 | -0.04 ± 0.02 | 0.65 ± 0.06 | – |
| T3-C2 | 0.40 ± 0.08 | 0.14 ± 0.09 | 0.00 ± 0.04 | -0.03 ± 0.03 | -0.08 ± 0.04 | – |
| T4-C2 | -0.08 ± 0.07 | 0.05 ± 0.04 | 0.12 ± 0.05 | 0.03 ± 0.02 | -0.07 ± 0.04 | – |
| T5-C2 | 0.01 ± 0.02 | -0.04 ± 0.03 | 0.03 ± 0.02 | 0.05 ± 0.03 | 0.12 ± 0.05 | – |
| T1-C3 | 0.07 ± 0.08 | -0.08 ± 0.08 | 0.03 ± 0.03 | 0.01 ± 0.00 | 0.11 ± 0.06 | – |
| T2-C3 | 0.06 ± 0.04 | 0.04 ± 0.03 | 0.01 ± 0.02 | 0.01 ± 0.01 | 0.03 ± 0.02 | – |
| T3-C3 | 0.02 ± 0.01 | 0.10 ± 0.09 | -0.02 ± 0.02 | -0.03 ± 0.02 | -0.03 ± 0.03 | – |
| T4-C3 | -0.00 ± 0.00 | 0.01 ± 0.01 | 0.02 ± 0.01 | 0.00 ± 0.00 | -0.01 ± 0.01 | – |
| T5-C3 | -0.01 ± 0.01 | -0.01 ± 0.01 | 0.02 ± 0.02 | 0.03 ± 0.02 | 0.05 ± 0.02 | – |
| T1-C4 | 0.00 ± 0.00 | -0.00 ± 0.00 | 0.00 ± 0.00 | 0.00 ± 0.00 | 0.01 ± 0.01 | – |
| T2-C4 | 0.01 ± 0.01 | 0.00 ± 0.01 | 0.00 ± 0.01 | -0.00 ± 0.00 | -0.01 ± 0.01 | – |
| T3-C4 | 0.00 ± 0.00 | 0.00 ± 0.00 | -0.01 ± 0.01 | -0.00 ± 0.00 | 0.01 ± 0.01 | – |
| T4-C4 | 0.00 ± 0.00 | 0.01 ± 0.01 | 0.01 ± 0.00 | 0.00 ± 0.00 | 0.00 ± 0.01 | – |
| T5-C4 | -0.00 ± 0.00 | -0.01 ± 0.01 | 0.01 ± 0.01 | 0.01 ± 0.00 | -0.01 ± 0.01 | – |
| Avg ± SEM | 0.50 ± 0.00 | 0.50 ± 0.00 | 0.50 ± 0.00 | 0.50 ± 0.00 | 0.50 ± 0.00 | 0.50 ± 0.00 |

Table 70: Catcher $\mathcal{W}$ for Qreg.

| | T1 | T2 | T3 | T4 | T5 | Avg ± SEM |
|---|---|---|---|---|---|---|
| T1-C1 | 0.00 ± 0.00 | 0.00 ± 0.00 | 0.00 ± 0.00 | 0.00 ± 0.00 | 0.00 ± 0.00 | 0.03 ± 0.01 |
| T2-C1 | 0.40 ± 0.08 | 0.22 ± 0.16 | 0.06 ± 0.06 | 0.06 ± 0.04 | 0.01 ± 0.01 | 0.07 ± 0.01 |
| T3-C1 | 0.69 ± 0.12 | 0.95 ± 0.05 | 0.26 ± 0.08 | 0.07 ± 0.04 | 0.01 ± 0.00 | 0.10 ± 0.01 |
| T4-C1 | -1.28 ± 0.21 | 0.34 ± 0.10 | 0.89 ± 0.03 | 0.80 ± 0.06 | 0.06 ± 0.01 | 0.03 ± 0.01 |
| T5-C1 | -1.23 ± 0.24 | -0.07 ± 0.06 | 0.71 ± 0.04 | 0.93 ± 0.02 | 0.29 ± 0.05 | 0.03 ± 0.01 |
| T1-C2 | -0.26 ± 0.08 | -0.10 ± 0.06 | -0.05 ± 0.02 | 0.11 ± 0.05 | 0.91 ± 0.02 | – |
| T2-C2 | 0.24 ± 0.06 | -0.04 ± 0.04 | -0.15 ± 0.06 | -0.04 ± 0.02 | 0.65 ± 0.06 | – |
| T3-C2 | 0.39 ± 0.08 | 0.03 ± 0.02 | -0.07 ± 0.04 | -0.06 ± 0.03 | -0.10 ± 0.04 | – |
| T4-C2 | -0.08 ± 0.07 | 0.02 ± 0.02 | 0.03 ± 0.02 | -0.00 ± 0.00 | -0.09 ± 0.04 | – |
| T5-C2 | -0.02 ± 0.01 | -0.05 ± 0.04 | 0.00 ± 0.00 | 0.01 ± 0.01 | -0.01 ± 0.03 | – |
| T1-C3 | 0.02 ± 0.03 | -0.09 ± 0.08 | -0.00 ± 0.00 | 0.00 ± 0.00 | 0.02 ± 0.03 | – |
| T2-C3 | 0.01 ± 0.00 | 0.01 ± 0.01 | -0.00 ± 0.00 | -0.00 ± 0.00 | 0.02 ± 0.02 | – |
| T3-C3 | 0.02 ± 0.01 | 0.01 ± 0.01 | -0.03 ± 0.02 | -0.03 ± 0.02 | -0.05 ± 0.02 | – |
| T4-C3 | -0.00 ± 0.00 | 0.01 ± 0.01 | 0.01 ± 0.01 | -0.00 ± 0.00 | -0.01 ± 0.01 | – |
| T5-C3 | -0.01 ± 0.01 | -0.01 ± 0.01 | 0.02 ± 0.02 | 0.00 ± 0.00 | -0.00 ± 0.00 | – |
| T1-C4 | -0.00 ± 0.00 | -0.00 ± 0.00 | -0.00 ± 0.00 | -0.00 ± 0.00 | -0.00 ± 0.01 | – |
| T2-C4 | -0.00 ± 0.00 | -0.00 ± 0.00 | -0.00 ± 0.00 | -0.00 ± 0.00 | -0.01 ± 0.01 | – |
| T3-C4 | -0.00 ± 0.00 | -0.00 ± 0.00 | -0.01 ± 0.01 | -0.01 ± 0.00 | -0.00 ± 0.00 | – |
| T4-C4 | -0.00 ± 0.00 | 0.01 ± 0.01 | 0.00 ± 0.00 | -0.00 ± 0.00 | -0.01 ± 0.01 | – |
| T5-C4 | -0.00 ± 0.00 | -0.01 ± 0.01 | 0.01 ± 0.01 | 0.00 ± 0.00 | -0.01 ± 0.01 | – |
| Avg ± SEM | -0.06 ± 0.02 | 0.06 ± 0.01 | 0.08 ± 0.01 | 0.09 ± 0.00 | 0.08 ± 0.00 | 0.05 ± 0.01 |

Table 71: Room $\mathcal{F}$ for Qreg+NWLU.

| | T1 | T2 | T3 | T4 | T5 | Avg ± SEM |
|---|---|---|---|---|---|---|
| T1-C1 | 5.94 ± 0.51 | 2.75 ± 0.24 | 5.85 ± 0.42 | 5.52 ± 0.43 | 2.80 ± 0.40 | 1.21 ± 0.08 |
| T2-C1 | 1.57 ± 0.18 | 1.63 ± 0.17 | 1.57 ± 0.20 | 1.81 ± 0.19 | 1.78 ± 0.18 | 0.45 ± 0.04 |
| T3-C1 | 1.90 ± 0.17 | 3.31 ± 0.14 | 1.98 ± 0.17 | 2.26 ± 0.23 | 3.19 ± 0.18 | 0.63 ± 0.04 |
| T4-C1 | 0.52 ± 0.13 | 1.54 ± 0.33 | 0.51 ± 0.14 | 0.62 ± 0.18 | 1.19 ± 0.25 | 0.24 ± 0.05 |
| T5-C1 | 0.06 ± 0.04 | 0.25 ± 0.11 | 0.06 ± 0.06 | 0.07 ± 0.07 | 0.25 ± 0.11 | 0.04 ± 0.01 |
| T1-C2 | -0.01 ± 0.01 | 0.54 ± 0.23 | -0.04 ± 0.06 | 0.02 ± 0.02 | 0.71 ± 0.21 | – |
| T2-C2 | 0.15 ± 0.05 | 0.07 ± 0.10 | 0.02 ± 0.04 | 0.17 ± 0.04 | 0.18 ± 0.14 | – |
| T3-C2 | 0.04 ± 0.02 | -0.00 ± 0.07 | 0.08 ± 0.04 | 0.00 ± 0.04 | -0.06 ± 0.04 | – |
| T4-C2 | -0.01 ± 0.02 | 0.15 ± 0.08 | 0.06 ± 0.05 | -0.00 ± 0.02 | 0.03 ± 0.04 | – |
| T5-C2 | -0.00 ± 0.02 | 0.07 ± 0.05 | -0.04 ± 0.05 | 0.02 ± 0.02 | 0.05 ± 0.05 | – |
| T1-C3 | 0.03 ± 0.04 | -0.01 ± 0.04 | -0.01 ± 0.04 | 0.01 ± 0.02 | 0.07 ± 0.05 | – |
| T2-C3 | -0.01 ± 0.02 | -0.08 ± 0.07 | 0.04 ± 0.08 | 0.00 ± 0.04 | 0.00 ± 0.03 | – |
| T3-C3 | 0.00 ± 0.02 | 0.05 ± 0.03 | 0.01 ± 0.04 | -0.01 ± 0.02 | -0.07 ± 0.03 | – |
| T4-C3 | 0.02 ± 0.01 | 0.07 ± 0.04 | 0.04 ± 0.03 | 0.03 ± 0.03 | 0.03 ± 0.05 | – |
| T5-C3 | -0.01 ± 0.01 | -0.00 ± 0.05 | 0.02 ± 0.04 | -0.01 ± 0.02 | 0.07 ± 0.04 | – |
| T1-C4 | 0.00 ± 0.00 | 0.09 ± 0.04 | 0.01 ± 0.03 | -0.02 ± 0.01 | 0.01 ± 0.04 | – |
| T2-C4 | 0.02 ± 0.02 | 0.02 ± 0.02 | 0.02 ± 0.01 | 0.03 ± 0.02 | 0.03 ± 0.04 | – |
| T3-C4 | -0.01 ± 0.01 | -0.02 ± 0.02 | 0.01 ± 0.02 | 0.03 ± 0.02 | 0.01 ± 0.04 | – |
| T4-C4 | -0.02 ± 0.01 | 0.02 ± 0.05 | -0.01 ± 0.03 | -0.00 ± 0.00 | -0.02 ± 0.03 | – |
| T5-C4 | -0.00 ± 0.02 | -0.05 ± 0.03 | -0.03 ± 0.03 | -0.01 ± 0.01 | -0.05 ± 0.05 | – |
| Avg ± SEM | 0.51 ± 0.01 | 0.52 ± 0.01 | 0.51 ± 0.01 | 0.53 ± 0.01 | 0.51 ± 0.01 | 0.51 ± 0.01 |

Table 72: Room $\mathcal{W}$ for Qreg+NWLU.

| | T1 | T2 | T3 | T4 | T5 | Avg ± SEM |
|---|---|---|---|---|---|---|
| T1-C1 | 0.00 ± 0.00 | 0.00 ± 0.00 | 0.00 ± 0.00 | 0.00 ± 0.00 | 0.00 ± 0.00 | -0.01 ± 0.01 |
| T2-C1 | 0.55 ± 0.09 | 0.23 ± 0.11 | 0.50 ± 0.09 | 0.54 ± 0.09 | 0.10 ± 0.12 | 0.08 ± 0.02 |
| T3-C1 | 0.86 ± 0.04 | 0.97 ± 0.03 | 0.89 ± 0.06 | 0.95 ± 0.05 | 0.92 ± 0.05 | 0.20 ± 0.01 |
| T4-C1 | 0.06 ± 0.03 | 0.35 ± 0.20 | 0.07 ± 0.04 | 0.15 ± 0.07 | 0.46 ± 0.15 | 0.05 ± 0.02 |
| T5-C1 | 0.05 ± 0.04 | -0.09 ± 0.06 | 0.01 ± 0.05 | -0.00 ± 0.05 | -0.13 ± 0.06 | -0.03 ± 0.01 |
| T1-C2 | -0.02 ± 0.01 | 0.11 ± 0.07 | -0.10 ± 0.03 | -0.00 ± 0.02 | 0.11 ± 0.05 | – |
| T2-C2 | 0.01 ± 0.01 | -0.06 ± 0.09 | -0.01 ± 0.03 | 0.01 ± 0.02 | 0.05 ± 0.10 | – |
| T3-C2 | 0.01 ± 0.01 | -0.14 ± 0.05 | -0.01 ± 0.04 | -0.03 ± 0.03 | -0.08 ± 0.04 | – |
| T4-C2 | -0.02 ± 0.01 | 0.02 ± 0.04 | -0.02 ± 0.03 | -0.00 ± 0.02 | -0.02 ± 0.03 | – |
| T5-C2 | -0.01 ± 0.01 | -0.02 ± 0.03 | -0.09 ± 0.04 | -0.01 ± 0.01 | -0.05 ± 0.04 | – |
| T1-C3 | 0.01 ± 0.02 | -0.08 ± 0.03 | -0.03 ± 0.03 | 0.00 ± 0.02 | -0.07 ± 0.03 | – |
| T2-C3 | -0.02 ± 0.01 | -0.13 ± 0.06 | -0.02 ± 0.05 | -0.03 ± 0.02 | -0.05 ± 0.02 | – |
| T3-C3 | -0.01 ± 0.01 | -0.01 ± 0.02 | -0.03 ± 0.02 | -0.02 ± 0.01 | -0.11 ± 0.03 | – |
| T4-C3 | -0.00 ± 0.00 | 0.03 ± 0.03 | -0.01 ± 0.02 | -0.00 ± 0.02 | -0.01 ± 0.04 | – |
| T5-C3 | -0.01 ± 0.01 | -0.05 ± 0.04 | 0.01 ± 0.03 | -0.02 ± 0.01 | 0.00 ± 0.02 | – |
| T1-C4 | -0.00 ± 0.00 | 0.03 ± 0.03 | -0.01 ± 0.02 | -0.02 ± 0.01 | -0.05 ± 0.03 | – |
| T2-C4 | -0.00 ± 0.00 | -0.02 ± 0.03 | 0.00 ± 0.00 | 0.01 ± 0.01 | -0.03 ± 0.02 | – |
| T3-C4 | -0.01 ± 0.01 | -0.02 ± 0.02 | -0.00 ± 0.02 | 0.01 ± 0.01 | -0.04 ± 0.03 | – |
| T4-C4 | -0.02 ± 0.01 | -0.03 ± 0.02 | -0.02 ± 0.02 | -0.00 ± 0.00 | -0.04 ± 0.02 | – |
| T5-C4 | -0.00 ± 0.02 | -0.08 ± 0.03 | -0.04 ± 0.02 | -0.01 ± 0.01 | -0.08 ± 0.05 | – |
| Avg ± SEM | 0.07 ± 0.01 | 0.05 ± 0.01 | 0.05 ± 0.00 | 0.08 ± 0.00 | 0.04 ± 0.00 | 0.06 ± 0.00 |

Table 73: Flappy $\mathcal{F}$ for Qreg+NWLU.

| | T1 | T2 | T3 | T4 | T5 | Avg ± SEM |
|---|---|---|---|---|---|---|
| T1-C1 | 2.57 ± 0.16 | 1.95 ± 0.13 | 1.25 ± 0.05 | 1.09 ± 0.07 | 1.17 ± 0.09 | 0.57 ± 0.04 |
| T2-C1 | 1.79 ± 0.14 | 2.06 ± 0.17 | 1.26 ± 0.13 | 0.78 ± 0.11 | 0.53 ± 0.07 | 0.48 ± 0.05 |
| T3-C1 | 1.99 ± 0.12 | 2.36 ± 0.13 | 2.17 ± 0.22 | 1.45 ± 0.12 | 1.10 ± 0.13 | 0.56 ± 0.05 |
| T4-C1 | -0.27 ± 0.12 | 0.05 ± 0.13 | 1.19 ± 0.23 | 1.16 ± 0.14 | 1.34 ± 0.19 | 0.24 ± 0.05 |
| T5-C1 | -0.74 ± 0.13 | -0.31 ± 0.21 | 0.18 ± 0.25 | 0.95 ± 0.14 | 1.26 ± 0.19 | 0.17 ± 0.05 |
| T1-C2 | 0.41 ± 0.07 | 0.13 ± 0.18 | -0.06 ± 0.25 | 0.44 ± 0.15 | 0.67 ± 0.29 | – |
| T2-C2 | 1.11 ± 0.17 | 0.43 ± 0.22 | 0.27 ± 0.20 | 0.44 ± 0.25 | 0.10 ± 0.24 | – |
| T3-C2 | 0.38 ± 0.13 | 0.52 ± 0.22 | 0.47 ± 0.20 | 0.26 ± 0.15 | 0.09 ± 0.31 | – |
| T4-C2 | 0.15 ± 0.19 | 0.26 ± 0.15 | 0.26 ± 0.21 | -0.18 ± 0.33 | 0.16 ± 0.13 | – |
| T5-C2 | -0.19 ± 0.18 | -0.09 ± 0.20 | 0.33 ± 0.20 | 0.23 ± 0.19 | 0.48 ± 0.16 | – |
| T1-C3 | -0.25 ± 0.14 | -0.20 ± 0.16 | 0.06 ± 0.19 | 0.16 ± 0.26 | 0.54 ± 0.27 | – |
| T2-C3 | 0.56 ± 0.15 | 0.03 ± 0.18 | -0.37 ± 0.31 | -0.13 ± 0.14 | 0.32 ± 0.25 | – |
| T3-C3 | 0.41 ± 0.22 | 0.32 ± 0.12 | -0.22 ± 0.17 | 0.04 ± 0.26 | 0.25 ± 0.40 | – |
| T4-C3 | 0.02 ± 0.17 | 0.06 ± 0.21 | 0.52 ± 0.26 | 0.50 ± 0.27 | -0.47 ± 0.29 | – |
| T5-C3 | 0.01 ± 0.19 | -0.01 ± 0.16 | 0.33 ± 0.18 | 0.28 ± 0.24 | 0.34 ± 0.24 | – |
| T1-C4 | 0.32 ± 0.19 | 0.23 ± 0.22 | 0.22 ± 0.28 | 0.09 ± 0.24 | 0.64 ± 0.19 | – |
| T2-C4 | 0.11 ± 0.27 | 0.23 ± 0.22 | 0.52 ± 0.13 | -0.08 ± 0.14 | -0.36 ± 0.27 | – |
| T3-C4 | 0.02 ± 0.14 | 0.15 ± 0.17 | -0.04 ± 0.22 | -0.18 ± 0.48 | -0.28 ± 0.29 | – |
| T4-C4 | 0.08 ± 0.32 | -0.05 ± 0.22 | -0.23 ± 0.23 | 0.37 ± 0.25 | -0.15 ± 0.46 | – |
| T5-C4 | -0.41 ± 0.21 | 0.33 ± 0.15 | 0.04 ± 0.22 | 0.35 ± 0.27 | -0.04 ± 0.23 | – |
| Avg ± SEM | 0.40 ± 0.01 | 0.42 ± 0.01 | 0.41 ± 0.01 | 0.40 ± 0.02 | 0.38 ± 0.02 | 0.40 ± 0.00 |

Table 74: Flappy $\mathcal{W}$ for Qreg+NWLU.

| | T1 | T2 | T3 | T4 | T5 | Avg ± SEM |
|---|---|---|---|---|---|---|
| T1-C1 | 0.00 ± 0.00 | 0.00 ± 0.00 | 0.00 ± 0.00 | 0.00 ± 0.00 | -0.00 ± 0.00 | -0.16 ± 0.03 |
| T2-C1 | 0.49 ± 0.08 | 0.52 ± 0.09 | 0.29 ± 0.06 | 0.26 ± 0.05 | 0.22 ± 0.04 | -0.12 ± 0.03 |
| T3-C1 | 0.67 ± 0.03 | 0.70 ± 0.06 | 0.38 ± 0.03 | 0.27 ± 0.04 | 0.23 ± 0.02 | -0.11 ± 0.03 |
| T4-C1 | -0.36 ± 0.08 | -0.14 ± 0.11 | 0.39 ± 0.08 | 0.28 ± 0.05 | 0.12 ± 0.06 | -0.21 ± 0.03 |
| T5-C1 | -0.78 ± 0.12 | -0.42 ± 0.16 | -0.24 ± 0.19 | 0.38 ± 0.08 | 0.18 ± 0.09 | -0.22 ± 0.03 |
| T1-C2 | -0.13 ± 0.05 | -0.24 ± 0.11 | -0.36 ± 0.14 | -0.13 ± 0.05 | -0.03 ± 0.11 | – |
| T2-C2 | 0.07 ± 0.06 | -0.20 ± 0.11 | -0.34 ± 0.14 | -0.27 ± 0.13 | -0.20 ± 0.18 | – |
| T3-C2 | 0.16 ± 0.08 | 0.00 ± 0.10 | -0.20 ± 0.09 | -0.07 ± 0.10 | -0.46 ± 0.20 | – |
| T4-C2 | -0.16 ± 0.13 | -0.05 ± 0.08 | -0.06 ± 0.16 | -0.48 ± 0.26 | -0.22 ± 0.08 | – |
| T5-C2 | -0.43 ± 0.15 | -0.43 ± 0.12 | -0.03 ± 0.15 | -0.11 ± 0.13 | -0.15 ± 0.14 | – |
| T1-C3 | -0.39 ± 0.14 | -0.36 ± 0.13 | -0.35 ± 0.14 | -0.27 ± 0.16 | -0.11 ± 0.12 | – |
| T2-C3 | -0.08 ± 0.07 | -0.24 ± 0.11 | -0.72 ± 0.21 | -0.33 ± 0.10 | -0.15 ± 0.17 | – |
| T3-C3 | -0.09 ± 0.08 | -0.08 ± 0.08 | -0.44 ± 0.17 | -0.41 ± 0.16 | -0.73 ± 0.19 | – |
| T4-C3 | -0.20 ± 0.15 | -0.25 ± 0.14 | -0.12 ± 0.17 | -0.10 ± 0.18 | -0.92 ± 0.19 | – |
| T5-C3 | -0.23 ± 0.12 | -0.31 ± 0.10 | -0.12 ± 0.13 | 0.00 ± 0.22 | -0.02 ± 0.17 | – |
| T1-C4 | -0.08 ± 0.12 | -0.12 ± 0.12 | -0.24 ± 0.14 | -0.46 ± 0.19 | 0.04 ± 0.11 | – |
| T2-C4 | -0.27 ± 0.17 | -0.17 ± 0.14 | -0.07 ± 0.14 | -0.60 ± 0.11 | -0.61 ± 0.24 | – |
| T3-C4 | -0.20 ± 0.13 | -0.09 ± 0.13 | -0.41 ± 0.16 | -0.66 ± 0.33 | -0.75 ± 0.18 | – |
| T4-C4 | -0.32 ± 0.19 | -0.31 ± 0.19 | -0.46 ± 0.16 | -0.21 ± 0.17 | -0.69 ± 0.31 | – |
| T5-C4 | -0.59 ± 0.16 | -0.25 ± 0.09 | -0.22 ± 0.16 | -0.07 ± 0.14 | -0.48 ± 0.13 | – |
| Avg ± SEM | -0.15 ± 0.01 | -0.12 ± 0.01 | -0.17 ± 0.02 | -0.15 ± 0.02 | -0.24 ± 0.02 | -0.16 ± 0.01 |

Table 75: Catcher $\mathcal{F}$ for Qreg+NWLU.

| | T1 | T2 | T3 | T4 | T5 | Avg ± SEM |
|---|---|---|---|---|---|---|
| T1-C1 | 7.43 ± 0.15 | 4.34 ± 0.48 | 0.87 ± 0.20 | 0.35 ± 0.06 | 0.12 ± 0.02 | 0.86 ± 0.05 |
| T2-C1 | 1.30 ± 0.07 | 2.57 ± 0.21 | 0.55 ± 0.08 | 0.10 ± 0.02 | 0.03 ± 0.01 | 0.29 ± 0.01 |
| T3-C1 | 1.25 ± 0.08 | 2.84 ± 0.24 | 3.75 ± 0.10 | 2.11 ± 0.22 | 0.23 ± 0.04 | 0.51 ± 0.03 |
| T4-C1 | -0.40 ± 0.14 | 0.27 ± 0.08 | 4.02 ± 0.14 | 4.31 ± 0.09 | 1.63 ± 0.25 | 0.49 ± 0.02 |
| T5-C1 | -0.44 ± 0.14 | -0.03 ± 0.01 | 0.80 ± 0.04 | 2.69 ± 0.21 | 3.94 ± 0.08 | 0.36 ± 0.02 |
| T1-C2 | 0.31 ± 0.12 | -0.08 ± 0.04 | -0.03 ± 0.02 | 0.41 ± 0.08 | 3.19 ± 0.25 | – |
| T2-C2 | 0.48 ± 0.15 | 0.01 ± 0.02 | 0.01 ± 0.02 | 0.00 ± 0.02 | 0.66 ± 0.09 | – |
| T3-C2 | 0.08 ± 0.04 | 0.08 ± 0.04 | 0.03 ± 0.02 | -0.01 ± 0.02 | -0.19 ± 0.10 | – |
| T4-C2 | 0.00 ± 0.01 | 0.02 ± 0.02 | 0.01 ± 0.01 | 0.03 ± 0.02 | -0.06 ± 0.09 | – |
| T5-C2 | -0.00 ± 0.00 | 0.00 ± 0.00 | -0.00 ± 0.00 | 0.02 ± 0.02 | 0.24 ± 0.10 | – |
| T1-C3 | 0.00 ± 0.01 | -0.00 ± 0.00 | -0.00 ± 0.00 | 0.00 ± 0.00 | 0.17 ± 0.11 | – |
| T2-C3 | -0.00 ± 0.00 | 0.00 ± 0.00 | -0.00 ± 0.00 | -0.02 ± 0.02 | -0.00 ± 0.03 | – |
| T3-C3 | 0.00 ± 0.00 | 0.00 ± 0.00 | 0.00 ± 0.00 | -0.00 ± 0.00 | -0.01 ± 0.01 | – |
| T4-C3 | 0.00 ± 0.00 | 0.00 ± 0.00 | 0.00 ± 0.00 | 0.02 ± 0.02 | -0.01 ± 0.02 | – |
| T5-C3 | -0.00 ± 0.00 | -0.00 ± 0.00 | -0.00 ± 0.00 | 0.00 ± 0.00 | -0.01 ± 0.02 | – |
| T1-C4 | -0.00 ± 0.00 | -0.00 ± 0.00 | -0.00 ± 0.00 | -0.00 ± 0.00 | 0.05 ± 0.02 | – |
| T2-C4 | -0.00 ± 0.00 | 0.00 ± 0.00 | 0.00 ± 0.00 | 0.00 ± 0.00 | 0.02 ± 0.02 | – |
| T3-C4 | -0.00 ± 0.00 | 0.00 ± 0.00 | 0.00 ± 0.00 | -0.00 ± 0.00 | 0.00 ± 0.00 | – |
| T4-C4 | 0.00 ± 0.00 | 0.00 ± 0.00 | 0.00 ± 0.00 | 0.00 ± 0.00 | 0.00 ± 0.00 | – |
| T5-C4 | -0.00 ± 0.00 | -0.00 ± 0.00 | 0.00 ± 0.00 | 0.00 ± 0.00 | -0.02 ± 0.02 | – |
| Avg ± SEM | 0.50 ± 0.00 | 0.50 ± 0.00 | 0.50 ± 0.00 | 0.50 ± 0.00 | 0.50 ± 0.00 | 0.50 ± 0.00 |

Table 76: Catcher $\mathcal{W}$ for Qreg+NWLU.

| | T1 | T2 | T3 | T4 | T5 | Avg ± SEM |
|---|---|---|---|---|---|---|
| T1-C1 | 0.00 ± 0.00 | 0.00 ± 0.00 | 0.00 ± 0.00 | 0.00 ± 0.00 | 0.00 ± 0.00 | 0.06 ± 0.00 |
| T2-C1 | 0.43 ± 0.02 | 0.56 ± 0.11 | 0.02 ± 0.02 | 0.00 ± 0.01 | 0.01 ± 0.00 | 0.08 ± 0.00 |
| T3-C1 | 0.97 ± 0.02 | 1.00 ± 0.00 | 0.27 ± 0.04 | 0.06 ± 0.01 | 0.02 ± 0.00 | 0.11 ± 0.01 |
| T4-C1 | -0.40 ± 0.14 | 0.23 ± 0.07 | 0.90 ± 0.02 | 0.60 ± 0.08 | 0.06 ± 0.02 | 0.06 ± 0.01 |
| T5-C1 | -0.45 ± 0.14 | -0.03 ± 0.01 | 0.71 ± 0.04 | 0.92 ± 0.01 | 0.38 ± 0.07 | 0.07 ± 0.01 |
| T1-C2 | -0.04 ± 0.03 | -0.08 ± 0.04 | -0.03 ± 0.02 | 0.36 ± 0.08 | 0.92 ± 0.02 | – |
| T2-C2 | 0.11 ± 0.05 | -0.02 ± 0.01 | -0.00 ± 0.01 | -0.01 ± 0.02 | 0.58 ± 0.07 | – |
| T3-C2 | 0.06 ± 0.03 | 0.00 ± 0.00 | 0.00 ± 0.00 | -0.01 ± 0.02 | -0.22 ± 0.09 | – |
| T4-C2 | -0.01 ± 0.01 | 0.01 ± 0.02 | 0.01 ± 0.01 | 0.00 ± 0.00 | -0.11 ± 0.08 | – |
| T5-C2 | -0.00 ± 0.00 | -0.00 ± 0.00 | -0.00 ± 0.00 | 0.00 ± 0.00 | 0.00 ± 0.03 | – |
| T1-C3 | -0.00 ± 0.00 | -0.00 ± 0.00 | -0.00 ± 0.00 | 0.00 ± 0.00 | 0.05 ± 0.05 | – |
| T2-C3 | -0.00 ± 0.00 | -0.00 ± 0.00 | -0.00 ± 0.00 | -0.02 ± 0.02 | -0.01 ± 0.03 | – |
| T3-C3 | -0.00 ± 0.00 | 0.00 ± 0.00 | -0.00 ± 0.00 | -0.00 ± 0.00 | -0.02 ± 0.01 | – |
| T4-C3 | -0.00 ± 0.00 | 0.00 ± 0.00 | 0.00 ± 0.00 | -0.00 ± 0.00 | -0.03 ± 0.02 | – |
| T5-C3 | -0.00 ± 0.00 | -0.00 ± 0.00 | -0.00 ± 0.00 | -0.00 ± 0.00 | -0.03 ± 0.02 | – |
| T1-C4 | -0.00 ± 0.00 | -0.00 ± 0.00 | -0.00 ± 0.00 | -0.00 ± 0.00 | -0.00 ± 0.00 | – |
| T2-C4 | -0.00 ± 0.00 | -0.00 ± 0.00 | -0.00 ± 0.00 | -0.00 ± 0.00 | 0.01 ± 0.02 | – |
| T3-C4 | -0.00 ± 0.00 | 0.00 ± 0.00 | -0.00 ± 0.00 | -0.00 ± 0.00 | -0.00 ± 0.00 | – |
| T4-C4 | -0.00 ± 0.00 | -0.00 ± 0.00 | 0.00 ± 0.00 | -0.00 ± 0.00 | -0.00 ± 0.00 | – |
| T5-C4 | -0.00 ± 0.00 | -0.00 ± 0.00 | -0.00 ± 0.00 | 0.00 ± 0.00 | -0.02 ± 0.02 | – |
| Avg ± SEM | 0.03 ± 0.01 | 0.08 ± 0.00 | 0.09 ± 0.00 | 0.09 ± 0.00 | 0.08 ± 0.01 | 0.08 ± 0.00 |

