# OpenReview forum: "Don't Forget the Critic: Value-Based Data Rehearsal for Multi-Cyclic Continual Reinforcement Learning"
_TMLR — Under review for TMLR_

### Review · Reviewer_AeQS · 2026-06-12

**Summary Of Contributions:**

**Summary**

This paper studies value-based data rehearsal for continual reinforcement learning in multi-cyclic task sequences. The authors investigate Q-value regularization for DQN and propose Qreg+NWLU, which improves vanilla Qreg by continuously collecting rehearsal samples, periodically updating stored Q-values, and applying regularization once samples are available. Experiments show that Qreg+NWLU generally improves return, reduces cyclic forgetting, and stabilizes learning compared with vanilla Qreg and several CRL baselines.

**Strength**

1. The paper studies an underexplored direction: value-based rehearsal in continual RL, rather than the more common actor-centric rehearsal setting.
2. The proposed method is simple and easy to implement. Its components directly address stale Q-values and delayed rehearsal regularization.
3. The experiments include multiple baselines, three environments, transfer and forgetting metrics, and useful ablations.

**Weakness**: The empirical scope is limited. All experiments use DQN, fixed task orders, and relatively modest environments, so the broader generality of the method is unclear.

**Audience:**

Yes

**Audience Explanation:**

Its focus on value-based rehearsal is relatively underexplored, and the multi-cyclic evaluation setting is relevant. Researchers working on continual reinforcement learning, replay-based methods, and value-based RL should be of interest.

**Broader Impact Concerns:**

No broader concerns.

**Claims And Evidence:**

Yes

**Claims Explanation:**

The experiments show that Qreg+NWLU outperforms vanilla Qreg and several CRL baselines on the tested multi-cyclic DQN tasks, and the ablations support the usefulness of its components.

**Requested Changes:**

**Critical change**: The code link currently appears to be empty.

**Change that would strengthen the paper**: Abbreviations such as DQN, DDQN, and EWC are not defined in full.

Beyond these points, I do not have additional requested changes.

---

> ### Author Response · Authors · 2026-07-16
>
> We thank the reviewer for their feedback. We address the requested changes below and have updated the paper to reflect these changes.
>
> 1. We have now posted the core code to the repository. The code is not currently runnable as it depends on forked repositories that would violate the double-blind review process. However, upon publication we plan to release the entire code with directions on how to setup the environment and run the code.
>
> 2. We have made sure to define and explain these abbreviations in the Introduction, Related Work, and Implementation sections.

---

### Review · Reviewer_XrL1 · 2026-07-06

**Summary Of Contributions:**

This paper investigates the problem of catastrophic forgetting in multi-cyclic continuous reinforcement learning (Multi-Cyclic CRL) when using the value-based deep reinforcement learning algorithm (DQN). In existing CRL approaches, data rehearsal is primarily used in the actor-critic method and only regularizes the actor, neglecting the critic’s regularization. This paper first attempted direct Q-value regularization (Qreg) but found it to be unstable and susceptible to “primacy bias.” To address this, the authors propose the Qreg+NWLU method, which includes two improvements: (1) Live-Updates: Continuously collect replay samples during training and dynamically update the stored Q-target values; (2) No-Wait Regularization: Do not wait until the first task is completed to initiate regularization; instead, apply it immediately as soon as samples become available.

**Audience:**

Yes

**Audience Explanation:**

1. Current work on CRL data replay has almost exclusively focused on the Policy Gradient/Actor-Critic framework and has deliberately avoided Critic regularization (as it is considered harmful). This paper systematically explores the combined application of Data Rehearsal and Q-value Regularization to DQN, a foundational value-based algorithm, and identifies the causes of instability in the original Qreg (outdated Q-objectives and sampling bias), making it a significant pioneering contribution.

2. The two improvements to Qreg+NWLU—Live Sampling + Dynamic Q-Value Updates and Early Regularization—are simple in design and easy to implement, requiring no changes to the network architecture or the introduction of complex meta-learning techniques. The paper uses Q-norm trajectory plots (Fig. 4) to visually demonstrate how the method smooths out Q-value jumps between tasks, providing a reasonable explanation for the source of the improved stability.

**Broader Impact Concerns:**

While the paper studies relatively simple game/minigame environments (Minihack, Flappy Bird, Catcher), the proposed method (multi-cycle continual RL with data rehearsal) could be directly transferred to more sensitive domains such as autonomous driving, robotic manipulation in dynamic factories, or adaptive recommendation systems where forgetting past behaviors has real-world consequences.

**Claims And Evidence:**

Yes

**Claims Explanation:**

The paper evaluates the model across three environments (including the challenging Minihack Room sequence), multiple cycles (4 cycles), and multiple random seeds, and provides detailed ablation experiments (removing No-Wait, Live, and Updates individually) that clearly quantify the role of each component. The introduction of the dual metrics Worst Transfer (W) and Final Transfer (F) better reflects forgetting and positive transfer in CRL than a single average return metric.

**Requested Changes:**

1.  All tables report Mean ± SEM(Standard Error of the Mean), but the full text does not include results of significance tests such as pairwise t-tests, Wilcoxon tests, or Bonferroni corrections. We recommend adding Task-pair significance tests and a summary of the significance matrix

2. In Table 69 (Catcher ℱ for Qreg), the last row, 0.50±0.00, appears repeatedly in each column. What is the reason for this?

3. Qreg+NWLU appears only in Room / Flappy / Catcher (Tables 71–76), and no systematic ablation was performed with NWLU alone. It is recommended to include an NWLU-only condition or to discuss why NWLU must be used in conjunction with Qreg.

4. EWC / L2 / PackNet perform relatively poorly, and the paper lacks search curves or sensitivity analyses for λ_EWC / λ_L2 / sparsity (PackNet).

---

> ### Author Response · Authors · 2026-07-16
>
> We thank the reviewer for their feedback and address the requested changes below.
>
> 1. Thank you for the suggestion to further strengthen the evaluation. We did not include significance testing for the Final Transfer and Worst Transfer metrics because, in our view, such comparisons would not be meaningful. Both metrics are normalized relative to an individual algorithm's own performance throughout training. Consequently, an algorithm with poor overall performance can still show little or no forgetting, appearing on par with an algorithm that achieved stronger performance with minimal or no forgetting. Direct statistical comparisons between algorithms on these metrics can therefore be misleading. For this reason, we use the red highlighting in the tables to emphasize the occurrence of notable forgetting rather than direct comparison of the metrics across methods. While these metrics are imperfect for cross-algorithm comparison, they remain useful for identifying whether forgetting or transfer is occurring. With that said, we performed pairwise t-tests comparing total return performance between Qreg+NWLU and all other algorithms across the main results, using a significance threshold of p < 0.05, provided in the tables below. For Room, we only see significant differences. For Catcher, we see two non-significant differences arise, with Qreg in Task 3 and Task 4. In Flappy, we see two non-significant differences arise, with PM in Task 1 and Qreg in Task 4.
>
> 2. After double-checking, the computation of these tables is correct. What we are seeing is likely due to rounding. All values are near 0.5 with correspondingly small standard errors, so cumulative rounding across many cells produces column averages that deviate only slightly from 0.50.
>
> 3. We would appreciate clarification on the reviewer's request. NWLU is not intended to function as a standalone learning mechanism. Rather, it is designed to work in conjunction with an underlying data rehearsal regularization method. In the context of value-function learning, this underlying method is Qreg. NWLU refers to strategies for managing the rehearsal replay buffer (RRB) or long-term memory, and for determining when regularization begins. As a result, evaluating NWLU in isolation would not be meaningful. However, in Section 6.1, we did conduct ablation studies testing each component of NWLU, assessing the contribution of its memory-management and regularization-triggering mechanisms separately.
>
> 4. As these methods serve as baselines, and given the substantial computational cost associated with multi-cycle experiments, we did not conduct an exhaustive hyperparameter search. Instead, we searched over hyperparameter ranges recommended in the original papers for each algorithm and selected the best-performing configuration from those candidate settings. This approach allowed us to maintain consistency with prior work while keeping the computational requirements of the study tractable.
>
> **Main Results: Room P-Values**
> | Task | DQN | PM | PackNet | EWC | L2 | MER | Qreg |
> |---|---|---|---|---|---|---|---|
> | T1 | 1.86e-07 | 2.30e-06 | 6.68e-03 | 7.44e-03 | 1.49e-07 | 1.43e-09 | 0.00e+00 |
> | T2 | 0.00e+00 | 0.00e+00 | 0.00e+00 | 2.71e-10 | 0.00e+00 | 7.27e-10 | 4.74e-04 |
> | T3 | 3.62e-07 | 3.93e-06 | 4.02e-02 | 1.02e-02 | 7.97e-07 | 9.67e-09 | 6.70e-10 |
> | T4 | 6.23e-08 | 3.96e-07 | 2.95e-02 | 5.57e-03 | 1.75e-08 | 2.64e-10 | 0.00e+00 |
> | T5 | 0.00e+00 | 0.00e+00 | 0.00e+00 | 0.00e+00 | 0.00e+00 | 4.13e-10 | 5.46e-04 |
>
> **Main Results: Flappy P-Values**
> | Task | DQN | PM | PackNet | EWC | L2 | MER | Qreg |
> |---|---|---|---|---|---|---|---|
> | T1 | 5.00e-05 | **1.39e-01** | 4.37e-06 | 2.08e-07 | 7.71e-08 | 9.19e-07 | 1.72e-09 |
> | T2 | 1.27e-05 | 1.72e-03 | 6.27e-06 | 2.14e-09 | 1.76e-10 | 1.98e-08 | 5.55e-08 |
> | T3 | 2.44e-05 | 1.22e-04 | 9.04e-06 | 1.04e-08 | 2.12e-09 | 4.29e-08 | 1.37e-03 |
> | T4 | 1.56e-05 | 1.35e-05 | 4.23e-05 | 2.25e-08 | 9.88e-09 | 1.40e-07 | **2.79e-01** |
> | T5 | 2.67e-06 | 2.98e-06 | 2.47e-05 | 9.58e-08 | 6.04e-08 | 4.82e-07 | 4.30e-03 |
>
> **Main Results: Catcher P-Values**
> | Task | DQN | PM | PackNet | EWC | L2 | MER | Qreg |
> |---|---|---|---|---|---|---|---|
> | T1 | 2.17e-06 | 2.39e-02 | 3.18e-06 | 1.05e-09 | 1.79e-06 | 3.12e-06 | 1.52e-02 |
> | T2 | 1.45e-06 | 3.33e-06 | 5.39e-05 | 1.88e-10 | 1.17e-07 | 1.45e-05 | 1.18e-02 |
> | T3 | 1.47e-05 | 8.10e-06 | 9.03e-04 | 7.45e-09 | 3.87e-09 | 6.84e-05 | **9.81e-01** |
> | T4 | 1.24e-04 | 1.10e-05 | 7.13e-03 | 1.55e-09 | 7.18e-09 | 4.99e-04 | **1.15e-01** |
> | T5 | 3.67e-06 | 4.29e-05 | 3.83e-05 | 6.87e-08 | 8.55e-08 | 2.91e-06 | 3.17e-03 |

---

### Review · Reviewer_6gi1 · 2026-07-07

**Summary Of Contributions:**

This paper introduces a method for continual reinforcement learning (CRL). Continual reinfocement learning has been known to suffer from catastrophic overfitting and existing evaluations overlook the scenario that multi-cyclic environments with repeated task sequences. The authors propose data rehearsal strategy Qreg+NWLU which allows continuous data rehearsal and adapts the Q-Values throughout training. The authors also adopts no-wait regularization specifically for the online task nature. Experiment results on various RL environments show that the proposed method Qreg+NWLU is able to address catastrophic forgetting and increase transferability among environments.

**Additional Comments:**

What would be the real-world environments to test/deploy continual reinforcement learning beyond simple Atari games? It would be interesting to see some discussions.

**Audience:**

Yes

**Audience Explanation:**

I think the problem this paper aims to address should be of interest to TMLR community. The authors investigates a practical and challenging problem of how to do continual reinforcement learning effectively. While the experimental results are not tested a wide range of RL environments, the initial results presented in this paper are promising and marks a solid step towards RL on the fly. Therefore I feel this paper should be considered for TMLR's acceptance.

**Claims And Evidence:**

Yes

**Claims Explanation:**

1. The authors have conducted thorough and careful ablation experiments to demonstrate the effectiveness of the proposed method. The experimental setup in 5.1 and 5.2 is also explained in great details.
2. The authors have discussed the limitations of the method, which is beneficial for understanding this approach better.

**Requested Changes:**

1. Can the authors evaluate on more RL environments other than Room, Flappy Bird and Catcher? for example, MuJoCo for continuous control, and Procgen to test generalization.
2. It would be good to show discussions on how the results or proposed method could  generalize to other RL domains, e.g., language models.
3. For Section 4, i think it would be make the presentation better if the authors could spend some effort on explaining the motivation for the approaches, instead of just introducing them up-front. For instance, for no-wait regularization, it would help to add discussions on why and how this is necessary for continual reinforcement learning, and how the proposal arises from the failure cases observed from previous approaches.

---

> ### Author Response · Authors · 2026-07-16
>
> We thank the reviewers for their time and feedback and try to address some of the requested changes and questions below.
>
> 1.  We selected these environments as initial testbeds for two main reasons. First, computational cost is a major consideration in continual RL (CRL). CRL experiments require conducting multiple learning cycles, making them substantially more expensive than many standard RL environments. Therefore, running a wide variety of RL benchmark task sequences is often computationally intractable. Second, these environments provide relatively softer transitions between tasks. Environments such as Atari typically involve hard transitions to entirely different tasks with little overlap between them, making it difficult to observe and evaluate knowledge transfer and retention. Therefore, the environments we chose offer a more controlled setting for analyzing the strengths and weaknesses of CRL algorithms. This is why these environments have been widely adopted in prior CRL research, which facilitates direct comparison with existing methods and ensures consistency with established evaluation protocols. Although the environments we use are relatively simple, they still clearly expose significant issues with current baselines, validating existing and proposed approaches. Procgen offers a promising benchmark with computational efficiency and softer transitions. However, even CRL algorithms built on more advanced RL methods, such as CLEAR using IMPALA, tend to struggle on it without a large number of steps [1]. We believe it acts as a good environment to explore in the future and is a strong candidate for more challenging CRL evaluations.
>
> 2. Thank you for the interesting suggestion. This paper focuses specifically on continual RL, and accordingly, the proposed method centers on value-function-based learning, which does not readily transfer to other learning domains, although it could potentially transfer to other value-function-based continual learning methods. That said, the core ideas underlying our method, such as frequent sampling of long-term memories, regularization over these memories, and continual updating of stored memories, are general in nature and could be extended to other domains (including the proposed language models), which could be an interesting direction for future research.
>
> 3. We attempted to motivate each strategy at the point it was introduced, as well as in our initial discussion of Qreg. For example, in the last paragraph of Section 4.1, we note that traditional implementations of Qreg only draws samples for the rehearsal replay buffer (RRB) at the end of a task. If the task is inadequately learned, this risks storing suboptimal or inaccurate Q-values and introduces sampling bias. In Section 4.2, we discuss how "Live" sampling mitigates this selection bias. Similarly, for "Updates," we explain that inaccurate Q-values can still be stored even with Live sampling, motivating the need to periodically update RRB samples throughout training. Finally, for "No-Wait," we discuss how it helps stabilize early learning, an issue we observe with Qreg on the Room task sequence. We also touch on these motivations at several points in Section 6, though these mentions are brief due to page limitations. That said, we are happy to expand these discussions given page space if the reviewer feels this would strengthen the paper.
>
> 4. Regarding real-world environments, this area/aspect remains largely underexplored in continual RL. Most CRL research is still conducted in highly controlled and simulated environments, due to the high cost, safety, and practical challenges associated with deploying and evaluating CRL agents in physical systems. Therefore, there is not yet a widely established real-world benchmark. However, we believe the most promising applications of CRL at the current stage lie in robotics, where agents must continuously adapt to changing environments and tasks in confined spaces (e.g., warehouses, factories, and homes). While demonstrating effectiveness in these real-world settings is an important long-term goal, we believe it remains a broader challenge for the CRL community and is beyond the scope of this paper.
>
> [1] Sam Powers, Eliot Xing, Eric Kolve, Roozbeh Mottaghi, and Abhinav Gupta. Cora: Benchmarks, baselines, and metrics as a platform for continual reinforcement learning agents. In Conference on Lifelong Learning Agents, Proceedings of Machine Learning Research, 2022.